# Dissecting the cellular specificity of smoking effects and reconstructing lineages in the human airway epithelium

Katherine C. Goldfarbmuren [1,6], Nathan D. Jackson[1,6], Satria P. Sajuthi[1], Nathan Dyjack[1], Katie S. Li[1], Cydney L. Rios[1], Elizabeth G. Plender[1], Michael T. Montgomery [1], Jamie L. Everman [1], Preston E. Bratcher [2,3], Eszter K. Vladar [4,5] & Max A. Seibold [1✉]

Cigarette smoke first interacts with the lung through the cellularly diverse airway epithelium and goes on to drive development of most chronic lung diseases. Here, through single cell RNA-sequencing analysis of the tracheal epithelium from smokers and non-smokers, we generate a comprehensive atlas of epithelial cell types and states, connect these into lineages, and define cell-specific responses to smoking. Our analysis infers multi-state lineages that develop into surface mucus secretory and ciliated cells and then contrasts these to the unique specification of submucosal gland (SMG) cells. Accompanying knockout studies reveal that tuft-like cells are the likely progenitor of both pulmonary neuroendocrine cells and CFTR-rich ionocytes. Our smoking analysis finds that all cell types, including protected stem and SMG populations, are affected by smoking through both pan-epithelial smoking response networks and hundreds of cell-specific response genes, redefining the penetrance and cellular specificity of smoking effects on the human airway epithelium.

[1] Center for Genes, Environment, and Health, National Jewish Health, Denver, CO 80206, USA. [2] Department of Pediatrics, National Jewish Health, Denver, CO 80206, USA. [3] Department of Pediatrics, University of Colorado-AMC, Aurora, CO 80045, USA. [4] Division of Pulmonary Sciences and Critical Care Medicine, University of Colorado-AMC, Aurora, CO 80045, USA. [5] Department of Cell and Developmental Biology, University of Colorado-AMC, Aurora, CO 80045, USA. [6] These authors contributed equally: Katherine C. Goldfarbmuren, Nathan D. Jackson. ✉email: seiboldm@njhealth.org

The human airway epithelium is a complex, cellularly diverse tissue that has a critical role in respiratory health by facilitating air transport, barrier function, mucociliary clearance, and the regulation of lung immune responses. These airway functions are accomplished through interactions among a functionally diverse set of both abundant (ciliated, mucus secretory, and basal stem) and rare cell types (tuft, pulmonary neuroendocrine, and ionocyte), which compose the airway surface epithelium. This remarkable cell diversity derives from the basal airway stem cell by way of multiple branching lineages[1,2], yet, the nature of these lineages, their transcriptional regulation, and the functional heterogeneity to which they lead, remain incompletely defined in humans. Equally important to airway function, if even more poorly understood on both a molecular and cellular level, is the epithelium of the submucosal glands (SMG), a network that is contiguous with the surface epithelium and a critical source of airway mucus and defensive secretions.

Gene expression and histological studies of the airway epithelium have demonstrated that both molecular dysfunction and cellular imbalance due to shifting cell composition in the epithelium are common features of most chronic lung diseases, including asthma[3] and chronic obstructive pulmonary disease[4] (COPD). This cellular remodeling is largely mediated by interaction of the epithelium with inhaled agents such as cigarette smoke, air pollution, and allergens, which are risk factors for these diseases. Among these exposures, cigarette smoke is the most detrimental and, as the primary driver of COPD[5] and a common trigger of asthma exacerbations[6], constitutes the leading cause of preventable death in the US[7]. Smoking is known to induce mucus metaplasia[8], and gene expression studies based on bulk RNA-sequencing have established the marked influence of this exposure on airway epithelial gene expression[9–11]. However, these bulk expression changes are a composite of all cell type gene expression changes, cell frequency shifts, and emergent metaplastic cell states, making it impossible to determine the precise cellular and molecular changes induced by smoke exposure using this type of expression data.

Here, we use single-cell RNA-sequencing (scRNA-seq) to define the transcriptional cell types and states of the tracheal airway epithelium in smokers and non-smokers, infer the lineage relationships among these cells, and determine the influence of cigarette smoke on individual surface and SMG airway epithelial cell types with single-cell resolution.

## Results

**Cellular diversity in the human tracheal epithelium**. To interrogate cellular diversity within the human tracheal epithelium, we enzymatically dissociated tracheal specimens from fifteen donors and then subjected these cells to scRNA-seq (Fig. 1a). These donors included six never-smokers (hereafter, non-smokers) and six heavy smokers (hereafter, smokers; ≥15 pack years) (Supplementary Table 1), allowing us to evaluate the transcriptional effects of smoking habit on each epithelial cell type.

Analysis of expression profiles from 36,248 epithelial cells identified ten cell clusters, each containing the full range of donors and smoking habits (Fig. 1b, Supplementary Fig. 1a, b). Between 243 and 2220, differentially expressed genes (DEGs) distinguished these clusters from one another (Supplementary Fig. 1c), facilitating assignment of cell types or states (Fig. 1c). Three KRT5-expressing basal cell populations were distinguished from one another by expression of genes involved in proliferation (Supplementary Fig. 1c), differentiation (IL33 and TP63), or a squamous metaplastic response to injury or stress (e.g. KRT14 and KRT13[12,13], Supplementary Fig. 2). This KRT14[high] basal cell population exhibited high proteasomal/ubiquitination activity,

consistent with a stressed cell[14]. These heterogeneous basal cell expression profiles were confirmed by immunofluorescence (IF) labeling in tracheal tissue (Fig. 1d). Besides the expected mucus secretory and ciliated cell populations, which were identified by canonical markers, we also identified a cluster characterized by high KRT8 expression (Fig. 1c, e, f). Consistent with KRT8 being a differentiating epithelial cell marker[15], KRT8[+] cells localized to the mid-to-upper epithelium, above KRT5[+] basal cells and often reaching the airway surface as shown by IF (Fig. 1f). Gene expression across KRT8[high] cells was highly heterogeneous, with a wide range of expression for both basal (KRT5 and TP63) and early secretory cell (SCGB1A1 and WFDC2) markers. We also identified low abundance clusters containing cells expressing markers diagnostic for pulmonary neuroendocrine cells (PNECs and ASCL1)[16], ionocytes (FOXI1)[17,18], or tuft cells (POU2F3)[19]. In total, these rare cells comprised only 0.8% of all epithelial cells (Supplementary Fig. 1d, Fig. 1g).

In addition to surface epithelial populations, two clusters highly expressed known glandular genes[20] (Fig. 1c, Supplementary Fig. 1e), suggesting that our digest isolated SMG epithelial cells. One of these SMG clusters highly expressed basal cell markers (e.g. KRT5 and KRT14), whereas the other exhibited mucus secretory character, including high MUC5B expression, which we confirmed with IF (Fig. 1h, Supplementary Fig. 1f).

**Smoking decreases functional diversity of the epithelium**. To understand the effect of smoking habit, we first examined whether genes previously reported to be differentially expressed between current and never-smokers, based on bulk RNA-seq from bronchial airway epithelial brushings[9], were similarly affected in each cell type independently. In smokers relative to non-smokers, all cell types exhibited higher mean expression of reported smoking-upregulated genes (Supplementary Fig. 3a). Similarly, five of eight cell types in smokers exhibited reduced expression of reported smoking-downregulated genes.

Unbiased transcriptome-wide differential expression analysis identified over 100 DEGs between smokers and non-smokers in each cell type (Supplementary Fig. 3b). Importantly, 4–54% of the smoking DEGs for each cell type were unique to that population, revealing a cell-type-specific aspect to the smoking response, discussed below (Fig. 2a, Supplementary Fig. 3b). In addition, we identified a core response to smoking that encompassed genes upregulated or downregulated in at least five cell types (Fig. 2a). Among this core response were polycyclic aromatic hydrocarbon metabolizing genes (e.g., CYP1B1), S100 family genes (known to regulate immune homeostasis and tissue repair), markers of squamous metaplasia (e.g., KRT14 and KRT17), and interferon and chemokine inflammatory signaling (Fig. 2b, Supplementary Fig. 3c). These results suggest that previously reported responses to smoking, which include toxin metabolism, macrophage recruitment, and squamous metaplasia, are a joint effort conducted across epithelial cell types.

The downregulated core response centered on deactivation of innate immune function characteristic of secretory cells, which included genes such as secretoglobin 1A1 (SCGB1A1), BPIFA1, SAA1, and LCN2 (Fig. 2b). Consistent with this, BPIFA1 and SCGB1A1, as well as SCGB1A1 protein, were often less abundant in the tracheal epithelium of smokers (Supplementary Fig. 4a, b). Notably, the downregulated core response contained HLA type II genes (Fig. 2b), possibly signaling an underappreciated role of antigen presentation by the epithelium (Supplementary Fig. 4c, d), which is suppressed by smoking.

Although underpowered with our sample size, we also explored whether relative proportions of epithelial cell types were altered in smokers (Fig. 2c, Supplementary Fig. 3d). In support of

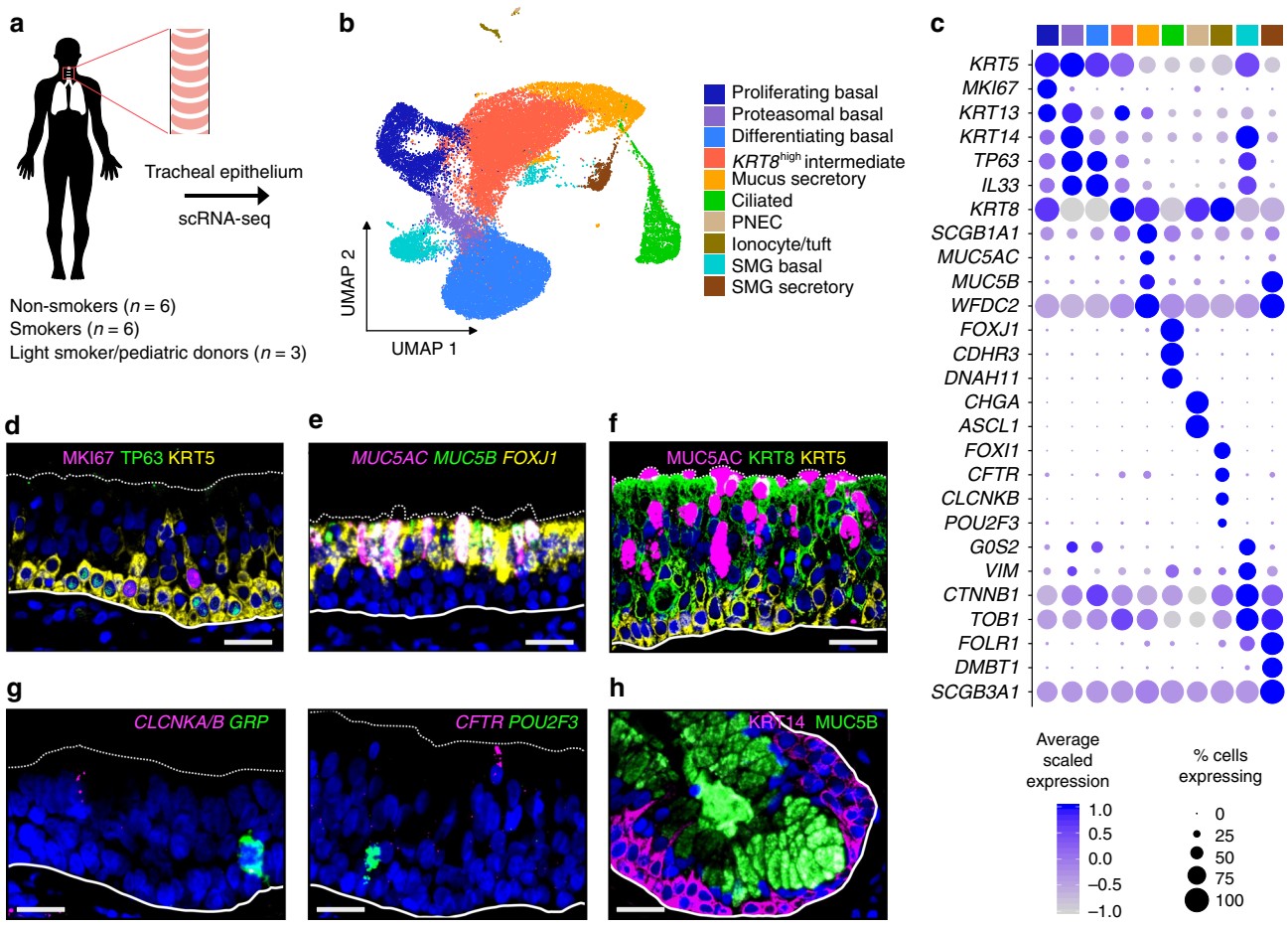

**Fig. 1 Single-cell RNA-seq reveals broad cellular diversity within the human tracheal epithelium. a** Study design for scRNA-seq of human tracheal epithelium. **b** UMAP visualization of cells in the human trachea depicts unsupervised clusters defining broad cell types present. **c** Dot plot highlights markers distinguishing broad cell categories based on average expression level (color) and ubiquity (size). Colored bar corresponds to the cell types/states in **b**. **d** Immunofluorescence (IF) labeling of human tracheal sections shows MKI67 (magenta) and TP63 (green) in a subset of KRT5high (yellow) basal cells. Scale bar = 25 µm. DAPI labeling of nuclei (blue). Dashed and solid lines represent the apical edge and basement membrane of the epithelium, respectively. **e** Fluorescence in situ hybridization (FISH) co-localizes (seen as white) MUC5B (green) and MUC5AC (magenta) mRNA to non-ciliated cells. FOXJ1 (ciliated marker, yellow). Dashed, solid lines and scale bar as in **d**. **f** IF labeling localizes KRT8 (green) to mid-upper epithelium, MUC5AC (magenta), KRT5 (yellow). Dashed, solid lines, and scale bar as in **d**. **g** FISH distinguishes rare cell mRNA markers: left, PNECs (GRP, green) and ionocytes (CLCNKA/B, magenta); right, ionocytes (CFTR, magenta) and tuft-like cells (POU2F3, green). Dashed, solid lines, and scale bars as in **d**. **h** IF labeling localizes MUC5B (green) and KRT14 (magenta) to distinct cells in the SMGs. Dashed, solid lines, and scale bar as in **d**. See also Supplementary Figs. 1 and 2.

well-established mucus overproduction in response to smoking[8], we observed clear trends toward increased frequencies of both surface and SMG mucus secretory cells, as well as a decreased frequency of ciliated cells among smokers. In addition, smokers displayed a trend toward increased frequency of proteasomal basal cells, which exhibit a stressed/squamous gene profile consistent with smoke exposure (Fig. 2c).

Transcriptional reprogramming in smokers mirrored these shifts in cell-type proportions (Fig. 2d). Specifically, genes upregulated by smoking in all basal cell populations were enriched for markers of both proliferating and proteasomal basal cells, suggesting that smoking shifts basal cells toward a more stressed state, less regulated by proliferation checks (Fig. 2d, top). In addition, genes upregulated by smoking among apical epithelial cell types (KRT8high, mucus secretory, and ciliated) were enriched for mucus secretory cell markers, supporting an increase in the secretory activity of these cells (Fig. 2d, top). In contrast, for each cell type, genes downregulated by smoking were enriched for the defining markers of that cell type, suggesting that the specialized functions of each cell type are systematically dampened by smoking (Fig. 2d, bottom). These results

demonstrate how smoking may shift overall epithelial function away from a diversity of cell types with specialized functions, toward a consensus increase in mucus secretion, proliferation, and response to stress.

**Mucus and other secretory cells form a continuous lineage.** Airway secretory cells include both club cells, which produce SCGB1A1-laden defensive secretions and mucus secretory cells. Although NOTCH signaling drives secretory cell fate in the differentiating airway epithelium[21,22], and the transcription factor (TF), SPDEF[23,24], specifically drives inflammation-induced mucus metaplasia, converting club cells into mucus cells in mice[25], little else is known regarding regulation of human secretory cell development. Therefore, we reconstructed a secretory cell lineage using pseudotime trajectory analysis[26] of the mucus secretory cells combined with the differentiating KRT8high population (Supplementary Fig. 5a, b). This analysis aligned most cells along a single lineage (Supplementary Fig. 5b) depicting basal-like cells transitioning into mucus secretory cells via three sequential phases of gene expression (Fig. 3a). The first of these

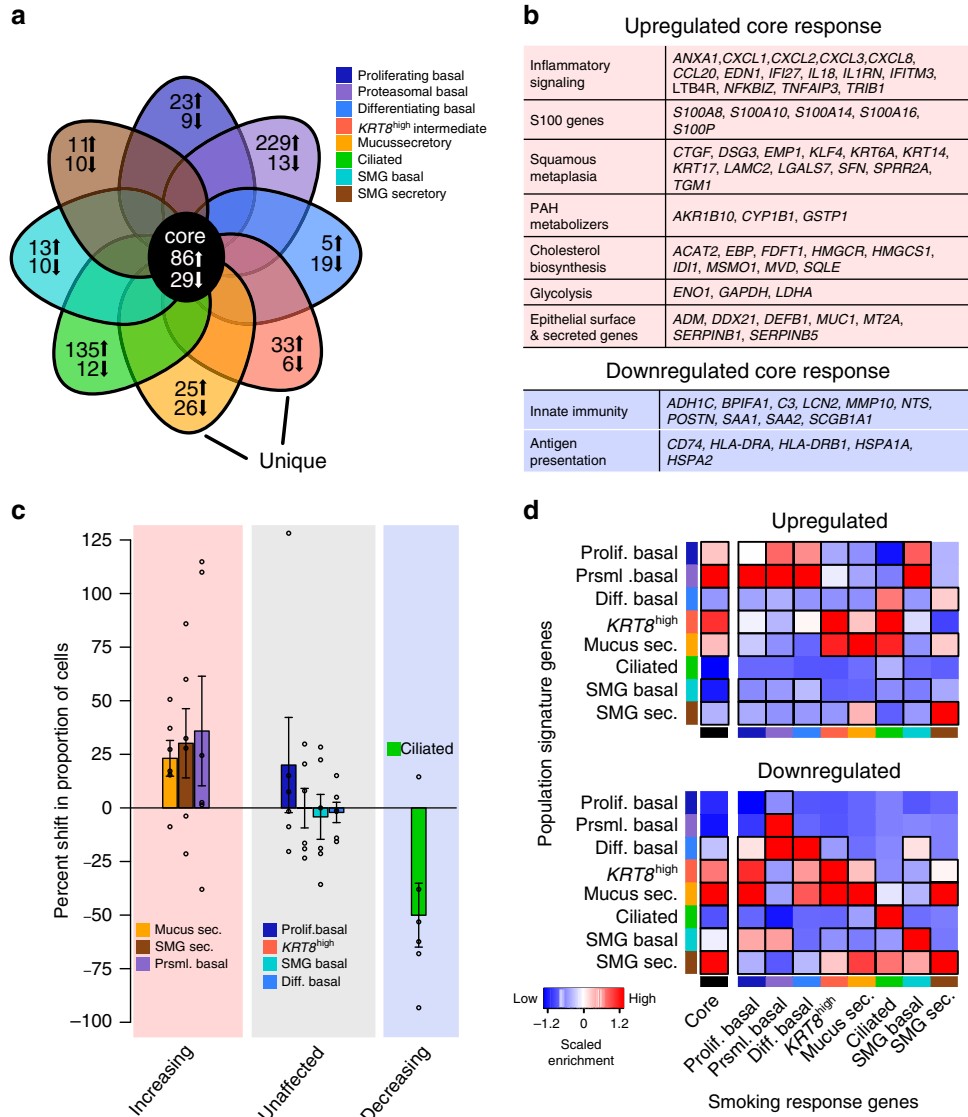

**Fig. 2 Smoking induces both shared and unique responses across cell types that decrease functional diversity of the epithelium. a** Schematic Venn diagram summarizes core and unique smoking responses across eight broad cell populations, with number of upregulated and downregulated genes unique to populations given in the tips and number of core genes affected in ≥5 populations in the center. Note degree of overlap in the diagram is not proportional to gene overlap for readability. Detailed percentages in Supplementary Fig. 3b. **b** Denoted key pathways and genes comprising the core upregulated and downregulated smoking response. **c** Relative proportions of broad cell populations shift with prolonged smoke exposure. Average and standard error of percent shift in proportion are shown for smokers ($n = 6$) relative to non-smokers ($n = 6$). Points show percent shifts for each individual smoker donor relative to the mean proportion across non-smokers. **d** Enrichment of cell-type signature genes from non-smokers (rows) in core and non-core smoking response genes for each of the broad cell populations (columns), based on hypergeometric tests. Color of box depicts scaled level of enrichment (−log10 of FDR-adjusted p-value), black outline of box indicates significant enrichment at FDR < 0.05. See also Supplementary Figs. 3 and 4.

phases (secretory preparation) was highly enriched for genes involved in ATP production and protein translation elongation, likely reflecting preparation for the high-energy demands of secretory protein production (Fig. 3a). Secretory preparation genes were enriched for WNT and NOTCH signaling and included the *NOTCH3* receptor, and TF, *KLF3* (Fig. 3a, b).

The second expression phase, characteristic of club cells, was enriched for O-linked glycosylation of mucins and xenobiotic metabolism, and contained airway transmembrane mucin genes. As potential regulators of these functions, both *NOTCH2* and *NKX3.1* reached peak expression during this pseudotime phase. Moreover, an array of TFs increased in expression during the club secretory pseudotime phase, eventually reaching a crescendo in the third and final phase of secretory cell development. These TFs

exhibited expression patterns mirroring that of *SCGB1A1* and included the driver of mucus metaplasia, *SPDEF*, as well as two previously unreported TFs, *MESP1* and *CREB3L1*. Also among this TF set was *XBP1*, which is likely driving a cellular stress response to the production of secreted proteins[27,28] (Fig. 3b).

The terminal mucus secretory phase contained both *MUC5AC* and *MUC5B*, along with the TF, *FOXA3*. This phase was also highly enriched for genes involved in N-linked glycosylation, vesicle coating, and unfolded protein response, consistent with these cells actively producing and secreting mucus (Fig. 3a,b). In summary, our in-silico trajectory analysis suggests a developmental lineage of human secretory cells, driven by sequentially activated TFs, which transitions through functional intermediates (club cells) to culminate in a multi-functional mucus secretory

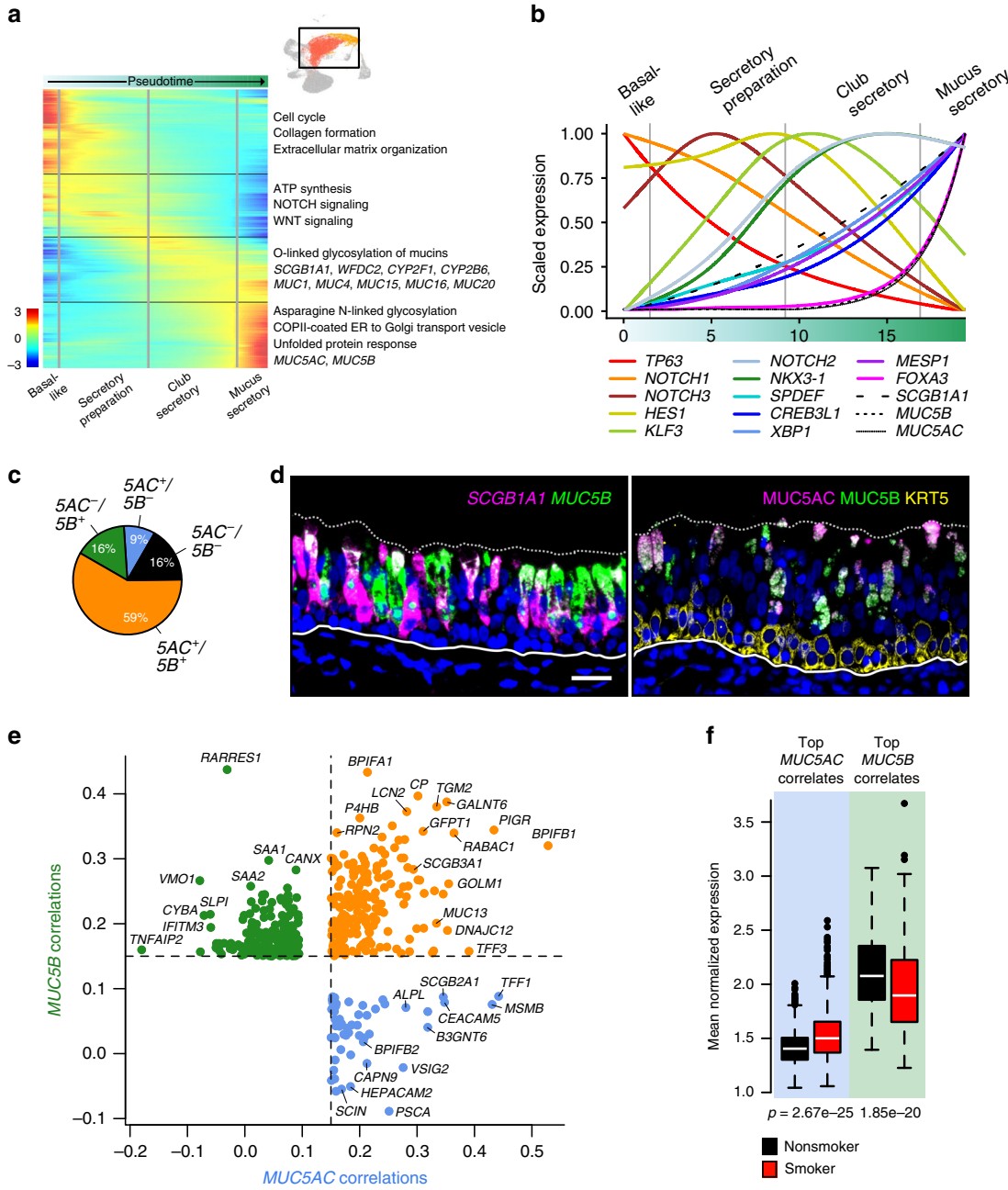

**Fig. 3 In vivo secretory cells form a continuous lineage and exhibit MUC5AC-correlated smoking effects. a** Heat map of smoothed expression across a Monocle-inferred lineage trajectory shows transitions in transcriptional programs that underlie differentiation in the in vivo human airway epithelium, from basal-like pre-secretory (*KRT8*^high) cells into mucus secretory cells. Select genes that represent these programs are shown, all significantly correlated with pseudotime. Key enrichment pathways and genes belonging to each block are indicated at right. **b** Scaled, smoothed expression of select transcriptional regulators (colored lines) and canonical markers (black dashed/solid lines) across pseudotime differentiation of human tracheal secretory cells in vivo. The *x* axis corresponds to the *x*-axis in **a**. **c** Pie chart depicts proportions of mature mucus secretory cells exhibiting different *MUC5AC* and *MUC5B* mucin co-expression profiles. **d** Co-expression of common secretory markers at the mRNA level (left, FISH with *SCGB1A1* in magenta, *MUC5B* in green) and protein level (right, IF labeling with MUC5AC in magenta, MUC5B in green and KRT5 in yellow). In both images, overlaid magenta/green appears as white. Dashed and solid lines represent the apical edge and basement membrane of the epithelium, respectively. Scale bar = 25 μm. **e** Smoking-independent correlation coefficients of *MUC5B*-correlated and *MUC5AC*-correlated genes. Genes are colored based on whether they were significantly correlated with only *MUC5B* (green), only *MUC5AC* (blue), or both (orange). Only the strongest correlations are plotted (correlations > 0.15), select genes are labeled. **f** Box plots illustrate the converse effects of smoking on the mean expression of the top 25 *MUC5B*- and *MUC5AC*-specific correlated genes in mature mucus secretory cells (n = 713 non-smoker cells, n = 886 smoker cells). Box centers give the median, upper and lower box bounds correspond to first and third quartiles, and the upper/lower whiskers extend from the upper/lower bounds up to/down from the largest/smallest value, no further than 1.5× IQR from the upper/lower bound (where IQR is the inter-quartile range). Data beyond the end of whiskers are plotted individually. *p*-values are from one-sided Wilcoxon tests. See also Supplementary Fig. 5.

cell[29–32], although all cells in this lineage need not reach this mucus secretory endpoint, depending on internal or external differentiation cues. However, lineage tracing studies will be needed to confirm this hypothesis.

**Smoking drives a MUC5AC co-expression profile in mucus cells.** Next, we investigated whether mature mucus secretory cells, as identified in our trajectory analysis (Fig. 3a, Supplementary Fig. 5b), comprise transcriptionally distinct subsets that carry out mucociliary and airway defense functions. Subclustering analysis did not yield novel or previously reported[17] mucus secretory cell subtypes (Supplementary Fig. 5c, d). When classified by co-expression patterns of the canonical airway secretory genes, *SCGB1A1*, *MUC5AC*, and *MUC5B*, most cells expressed *SCGB1A1* (83%) and/or at least one mucin (84%). Moreover, 59% of cells expressed both *MUC5AC* and *MUC5B*, whereas 16% and 9% were only positive for *MUC5B* and *MUC5AC*, respectively (Fig. 3c, Supplementary Fig. 5c), findings confirmed by tissue labeling (Figs. 1e and 3d). These patterns suggest that these canonical secretory genes all reach peak expression together in this mature mucus secretory state[29].

Examining smoking effects on these mucins, mean expression of *MUC5AC* was increased, whereas *MUC5B* trended downward. Similarly, the frequency of *MUC5AC*+/*MUC5B*− cells increased with smoking, whereas the frequency of *MUC5AC*−/*MUC5B*+ cells decreased (Supplementary Fig. 5e, f). Further, hundreds of genes were correlated with either *MUC5AC* or *MUC5B* (but not both) within mature mucus secretory cells, suggesting that these mucins are associated with distinct functional programs (Fig. 3e). Smoking shifted the transcriptional balance toward the *MUC5AC* program, as evidenced by increased and decreased mean expression of *MUC5AC*- and *MUC5B*-correlated genes with smoking habit, respectively (Fig. 3f). For example, mature mucus secretory cells of smokers exhibited depletion of *MUC5B*-specific correlated genes encoding known secretory defense proteins (e.g. *SLPI*, *SAA1*, and CYBA) and a suite of class II HLA genes, and at the same time exhibited enhancement of *MUC5AC*-specific correlated genes encoding a different set of secreted proteins (e.g. *MSMB*, *TFF1*, and *BPIFB2*) and genes involved in both mucin production and its associated ER stress response (Fig. 3e). Together, these data further support a continuous secretory cell lineage, demonstrating how smoking may drive mucin-balanced secretory cells toward an expression state dominated by a *MUC5AC* co-expression program.

**Functional differences in surface and SMG secretory cells.** Human airway mucus is a composite of secretions produced by both surface and SMG mucus secretory cells[33]. To identify shared and unique aspects of these related cell types, we compared the expression profiles of these two populations. This analysis identified 256 genes that defined both SMG and surface mucus secretory cell types, which were enriched for transmembrane transport and mucosal defense (Supplementary Fig. 6a). Despite these similarities, hundreds of genes were uniquely characteristic of either surface or SMG mucus secretory cells (Fig. 4a, Supplementary Fig. 6b). Namely, the SMG population specifically expressed a unique repertoire of secretory proteins with strong enrichment for innate immunity functions (Fig. 4a). Furthermore, although both populations highly expressed *MUC5B* (Fig. 4b), *MUC5AC* was much lower in SMG cells, both in terms of mean expression and number of expressing cells (Fig. 4c). Expression of genes involved in ER-to-Golgi vesicle-mediated transport, ER protein processing, and mucin glycosylation were also significantly reduced or absent in the SMG cells (Fig. 4a). Distinct TF profiles were observed in these two cell types, including high expression of CREB3L1 and SPDEF on the surface, whereas SMG cells uniquely expressed *SOX9*, as well as known lacrimal gland TFs, *FOXC1* and *BARX2*[34,35] (Fig. 4a). These data suggest that unique TFs in SMG cells drive the production of mucus secretions with novel defense proteins. Moreover, SMG secretory cells produce a *MUC5B*-dominated mucus that requires less post-translational processing and glycosylation than surface epithelial mucus production, consistent with recent studies finding that mucus produced from the SMGs has distinct physical properties from that of surface epithelia[36,37].

Similar to the surface epithelia, smoking suppressed SMG secretory cell *MUC5B* expression, while the proportion of *MUC5AC*-only cells increased (Supplementary Fig. 6c). Smoking also uniquely increased expression of interleukin-6 (*IL6*), *BPIFA2*, *BPIFB2*, *FCGBP*, and *STATH*, whereas *BPIFB1* was suppressed (Supplementary Fig. 6d), illustrating the altered state of these cells among smokers.

**Basal and myoepithelial cell states of the SMG.** Our broad clustering identified a population of cells expressing a signature characteristic of SMG basal cells (*KRT14*, *CAV2*, *IFITM3*, and *ACTN1*) (Fig. 1b, Supplementary Figs. 2b and 6e)[2,38–40]. Subclustering of this population revealed three SMG basal cell states, united by *KRT14* expression (Fig. 4d, e, Supplementary Fig. 6e, f). The smallest cluster uniquely expressed 34 muscle contraction/development genes (e.g. *ACTA2* and *MYLK*) (Supplementary Fig. 6f), reflecting murine myoepithelial cells[41,42]. Importantly, recent studies have established that the myoepithelial cell serves as the stem cell of the murine SMG, and can also regenerate surface epithelia in settings of severe injury[41,42]. Our ACTA2+ SMG population was *MKI67*− and poorly expressed *KRT5*, suggesting it represents a quiescent myoepithelial population. Compared to these myoepithelial cells, the other two SMG basal states exhibited expression more typical of surface basal cells, including higher *KRT5*. Basal state A expressed more *IL33*, a marker of surface differentiating basal cells in our dataset (Fig. 4e), whereas state B expressed more SMG secretory markers, potentially marking the former as a basal cell state initiating differentiation and the latter as a state committed to a secretory fate.

Supporting this, pseudotime trajectory[26] analysis suggested that basal state A leads to basal state B, and then into SMG mucus secretory cells (Supplementary Fig. 6g, Fig. 4f). Transitioning out of the basal state A involved reducing expression of *KRT14* and *IL33* while simultaneously gaining expression of SMG mucus cell TFs (*SOX9*, *FOXC1*, and *BARX2*), as well as TFs, *FOSL1* and *PTTG1* (Fig. 4f). IF labeling of ACTA2+ myoepithelial cells revealed a spectrum of co-localization with KRT5+ SMG basal cells (Fig. 4g), supporting the human myoepithelial cell as the initiation point for SMG differentiation.

Smoking upregulated expression of 162 genes in SMG basal cells, fewer than 6% of which encompassed baseline markers of these cells. Accordingly, smokers' SMG basal cells appear to have acquired novel functions characteristic of the proteasomal basal cell population, while downregulating extracellular matrix remodeling gene expression (Supplementary Fig. 6h).

**Sequential transcription programs drive motile ciliogenesis.** Upon cell fate acquisition, nascent ciliated cells activate expression of a vast transcriptional program, precipitating the generation of hundreds of cytoplasmic basal bodies which traffic to and dock with the apical membrane where they elongate motile axonemes[43]. As our in vivo scRNA-seq data did not wholly capture the heterogeneity reflective of this progression, we studied the process by culturing basal tracheal epithelial cells at air–liquid interface (ALI) and harvesting replicate cultures at 20 timepoints

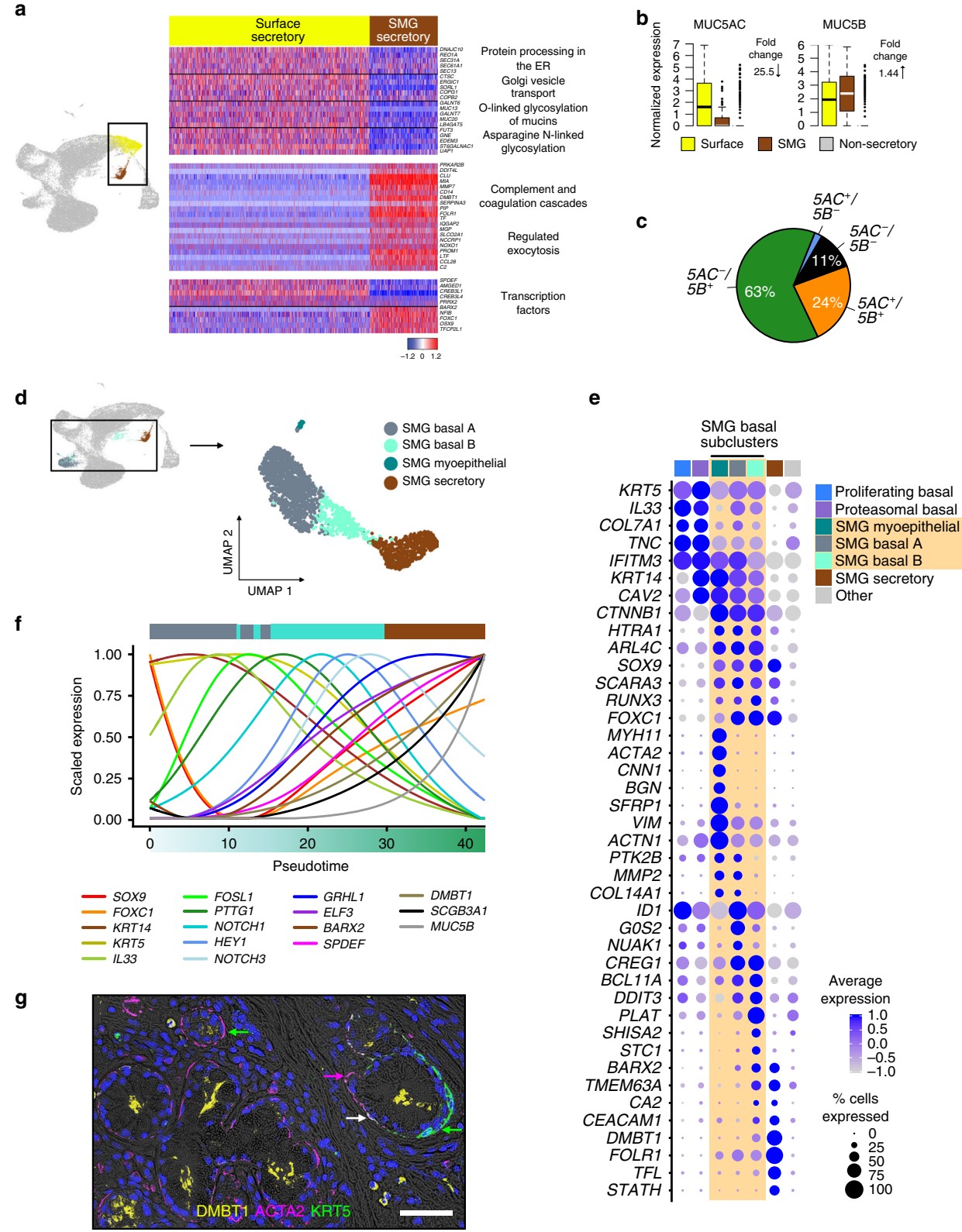

across mucociliary differentiation for scRNA-seq analysis (Fig. 5a–c, Supplementary Fig. 7a). Clustering yielded three populations uniquely expressing ciliary genes (Fig. 5d, Supplementary Fig. 7b). To determine how these populations fit into ciliated cell differentiation, we performed trajectory reconstruction[44] of these and the other cell populations. We identified two major lineages, one of which transitioned from basal cells through early secretory cells, culminating in a sequential ordering of the three ciliary populations (Fig. 5e) that matched the real-time appearance of these ciliary states across ALI differentiation (Supplementary Fig. 7c). The early ciliating state was enriched for genes involved in basal body assembly[45] (*DEUP1*, *STIL*, and *PLK4*) (Fig. 5d) and

**Fig. 4 Specialized SMG mucus secretory cells are predicted to derive from SMG basal stem cells and both are modified by smoking. a** Heat map depicts select genes, functional terms and TFs that distinguish surface and SMG secretory cells. Detailed heat map in Supplementary Fig. 6b. **b** Box plots of normalized mucin expression across surface mature mucus secretory (yellow, $n = 1858$ cells), SMG mucus secretory (brown, $n = 633$ cells) and non-secretory cells (gray, $n = 32,589$ cells). Box centers give the median, upper and lower box bounds correspond to first and third quartiles, and the upper/lower whiskers extend from the upper/lower bounds up to/down from the largest/smallest value, no further than 1.5× IQR from the upper/lower bound (where IQR is the inter-quartile range). Data beyond the end of whiskers are plotted individually. Median fold-change between surface and SMG secretory cells is indicated. **c** Pie chart depicts proportions of SMG secretory cells exhibiting different *MUC5AC* and *MUC5B* mucin co-expression profiles. **d** UMAP of SNN subclustering for SMG cells. **e** Dot plot showing the expression of markers that unite and distinguish SMG basal cell substates (peach underlay), relative to surface populations, SMG secretory cells, and each other. **f** Scaled, smoothed expression of key genes and regulators across a pseudotime trajectory that models the differentiation process of SMG cells. A minimum spanning tree of the trajectory can be found in Supplementary Fig. 6g. **g** IF labeling illustrates myoepithelial cells (ACTA2+, magenta) transitioning to SMG basal cells (KRT5+, green), where overlaid magenta/green appear white. Example myoepithelial (magenta arrows), SMG basal (green arrow), and transitioning (white arrow) cells are highlighted. DMBT1 is in yellow and scale bar = 50 μm. See also Supplementary Fig. 6.

also contained known early transcriptional drivers of ciliogenesis (*MCIDAS*, *MYB*, and *TP73*)[46,47] (Fig. 5f). The subsequent ciliary states were highly enriched for mature ciliated cell genes, but the first of these in pseudotime was distinguished by basal body docking (*CEP290* and *TTBK2*)[48] and axoneme assembly (*IFT52*) genes (Fig. 5d), and the highest expression of ciliogenesis TFs (*GRHL2*, *RFX2*, and *RFX3*)[49–51]. These TFs were downregulated in the final ciliary state, which displayed the highest expression of the ciliogenesis and ciliary maintenance TF, *FOXJ1*[52] (Fig. 5f). This third state also showed high expression of mitochondrial biogenesis and ATP synthesis genes, consistent with the significant energy requirements of axonemal motility[53] (Supplementary Fig. 7d). This ordering of the second and third states was supported by unspliced-to-spliced ratios[54] (Fig. 5g, Supplementary Fig. 7e), illustrating a potential role for mRNA processing to trigger the final stage of ciliogenesis.

**Smoking blocks early FOXN4-mediated ciliogenesis.** Similar to the recently reported deuterosomal ciliated cell state, our early ciliating state exclusively expressed the forkhead box N4 gene (*FOXN4*)[18,32]. As FOXN4 is a regulator of ciliogenesis in *Xenopus*[55], we explored whether it has a similar role in humans through our ALI culture model. We detected nuclear FOXN4 early (day 9) but not late in ALI mucociliary differentiation (day 21) (Fig. 6a). CRISPR-Cas9 knockout (KO) of *FOXN4* carried out in basal cells (Supplementary Fig. 8a) resulted in a partially penetrant block to ciliogenesis upon differentiation. At day 21, 76% of ciliated cells had no or short and sparse cilia compared to only 2% in the control (Fig. 6b, c). The abnormal KO cells retained basal bodies and deuterosomes[45] in the cytoplasm (Fig. 6d), indicating that the basal body generation machinery was intact, but basal body docking and deuterosome disassembly was blocked. Thus, our data are consistent with *FOXN4* regulating this step in early ciliogenesis.

In vivo, most ciliated cells were mature and only a small subcluster of ciliated cells resembled the early *FOXN4*+ ciliating state (Fig. 6e, f; Supplementary Fig. 8b). Notably, these earliest ciliating cells also expressed *SPDEF*, *MUC5AC*, and mature mucus secretory genes (Fig. 6f, right), in contrast to the non-mucus producing early secretory cells that gave rise to the *FOXN4*+ early ciliating state in vitro (Fig. 5e). To see whether this putative transdifferentiating state could be generated in vitro, we blocked Notch signaling by gamma-secretase inhibitor (DAPT) treatment to induce ciliated cell formation in mature ALI cultures[56,57]. This treatment generated mature mucus secretory cells with an early ciliogenesis phenotype, as judged by confocal IF microscopy (Fig. 6g, Supplementary Fig. 8c, d). These results suggest that during de novo epithelization, ciliated cells derive from early secretory cells, whereas in the homeostatic airway, mature mucus cells transdifferentiate into ciliated cells through

this hybrid state[32], possibly in response to external environmental stimuli.

Although both hybrid and mature ciliated cells increased *MUC5AC* expression with smoking, mature ciliated cells uniquely activated many genes involved in xenobiotic metabolism, interferon-gamma signaling, and response to oxidative stress (Fig. 6h, Supplementary Fig. 8e), likely reflecting an attempt by these cells to cope with smoking-induced molecular damage. In contrast, hybrid secretory/ciliating cells markedly dampened their characteristic, ciliogenesis gene expression, an effect not observed in mature ciliated cells (Fig. 6i, Supplementary Fig. 8e), suggesting that hybrid cells are uniquely vulnerable to losing their ciliogenesis function and thus their ability to regenerate ciliated cells in the airways of smokers.

**A lineage relationship among rare epithelial cell types.** Subclustering rare cells identified three distinct populations, expressing canonical markers of either PNECs[16], tuft cells[19,58], or ionocytes[17,18] (Fig. 7a). Differential expression analysis allowed us to infer function for each rare cell type (Fig. 7b). Human PNECs specifically expressed multiple secreted neurotransmitters, neurotransmitter processing genes, and voltage-gated cation channels (Fig. 7b). Human ionocytes highly expressed genes involved in energy production, cholesterol biosynthesis, and *CFTR*[17,18], in addition to several other chloride channel genes (Fig. 7b, c). Despite characteristic *POU2F3* and *ASCL2* expression in our human tuft cell population, many diagnostic markers in mice (e.g., *GNAT3* and *TRPM5*) were not detected in our tuft population, while other murine markers were observed (*HCK* and *LRMP*)[17] (Fig. 7b, Supplementary Fig. 9a). Therefore, we classified our *POU2F3*+/*ASCL2*+ cells as tuft-like, to signal their unique profile compared to previously described murine tuft cells.

Consistent with murine lineage tracing results[17], the appearance of these populations in our ALI cultures (Supplementary Fig. 9b–d) suggests that rare cells derive from basal cells. Yet, little is known about the differentiation process that produces these cells. The tendency for these populations to cluster together in the larger dataset suggested these cell types may develop from the same lineage. Supporting this, differential expression analysis identified 133 genes highly expressed in each of these three in vivo populations, compared to non-rare cells, as well as hundreds of genes uniquely shared between pairs of rare cell types (Supplementary Table 2). Of these pairs, ionocytes and tuft-like cells shared 107 unique genes, 16% of which were transcriptional regulators, such as reported ionocyte TF, *ASCL3*[17,18] (Supplementary Table 2). Moreover, ionocyte marker, *FOXI1*[59], reported to be sufficient to produce *CFTR*high ionocytes[17,18], was expressed by roughly half of *POU2F3*+ tuft-like cells (Fig. 7c). Despite *FOXI1* expression levels comparable to ionocytes, these tuft-like cells mostly lacked detectable *CFTR* expression. Fluorescence

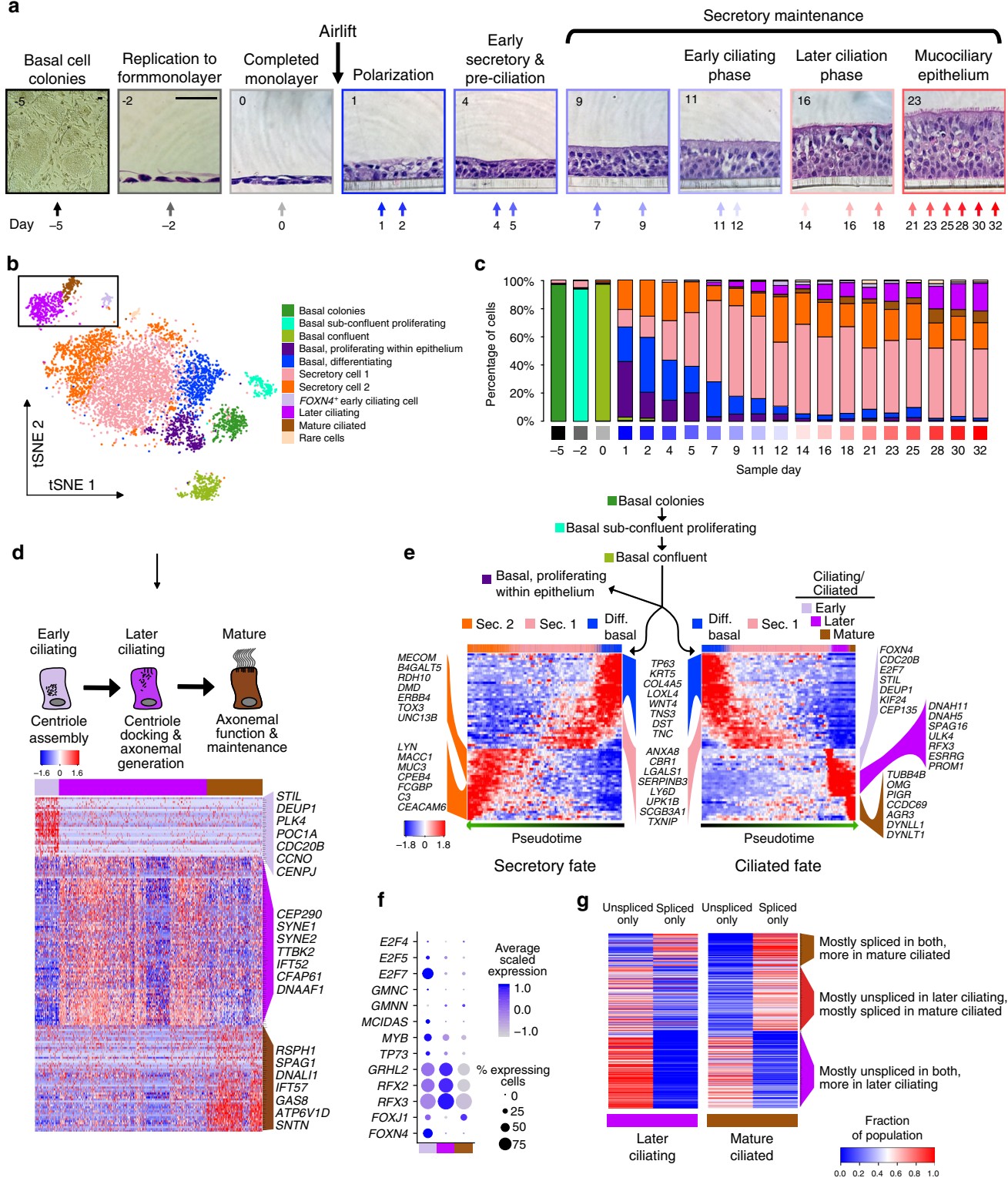

in situ hybridization (FISH) confirmed this co-expression pattern (Fig. 7d), where, on average, 48% of *FOXI1*+ cells exhibited an ionocyte pattern (*CFTR*+/*POU2F3*−), whereas 38% of *FOXI1*+ cells exhibited a tuft-like pattern (*CFTR*−/*POU2F3*+) (Fig. 7e, Supplementary Fig. 9e). This suggests a possible lineage where tuft-like cells give rise to ionocytes. Consistent with this hypothesis, tuft-like expression signatures begin and peak early in ALI differentiation, whereas signatures of both ionocytes and PNECs appear much later (Fig. 7f). Further, of the three in vivo

rare cell types, tuft-like expression was most similar to that of basal cells (Supplementary Fig. 9f). These data support a rare cell lineage that proceeds from tuft-like cells to ionocytes and possibly PNECs.

To investigate this putative lineage relationship, we targeted the *POU2F3* and *FOXI1* TF genes, required for tuft cell[19] and ionocyte specification, respectively[17,19], using CRISPR/Cas9 KO in human tracheal basal cells (*n* = 5 donors, Supplementary Fig. 8a). Hereafter, KO refers to the mosaic cultures produced by

**Fig. 5 Sequential transcriptional programs drive motile ciliogenesis. a** Histological overview of human basal cell ALI mucociliary differentiation. Image outlines: shades of black = submerged culture, shades of blue to red = polarized differentiation and maintenance of ALI human epithelial cell culture. Representative brightfield or H&E stained images of indicated timepoints are shown, scale bars in far left panels are both 50 μm. **b** tSNE plot depicts the distribution of inferred clusters of in vitro cells transcriptionally sampled from across the entire differentiation time course. Cluster identities based on expressed markers are shown at the right. Ciliated cell clusters are boxed. **c** Proportion of cells in each cell state (corresponding to clusters in **b**) present at each timepoint over differentiation. Time course black/blue/red gradient coloring at bottom corresponds to colors in **a**. **d** Heat map depicts gene signatures of three ciliated cell states (function summarized in schematic above) in human airway epithelial ALI cultures sampled across differentiation. Genes plotted were those with known ciliogenic function that were characteristic of one of the three states, as indicated. **e** Similarities and differences in transcriptional programs between distinct pseudotime lineages constructed with Slingshot[81] that lead to mature secretory and ciliated cells in vitro. Select markers or genes correlated with pseudotime are indicated. **f** Dot plots reveal TFs exhibiting expression associated with ciliated states in vitro. **g** Heat map illustrates differences in proportions of spliced or unspliced transcripts for a given gene (one per row) between later ciliating and mature ciliated cells that exhibit non-zero expression for the gene. mRNA splice status was inferred using the Velocyto[78] pipeline. Genes listed are cilia-related genes with non-zero expression (ignoring splicing) in at least 10% of cells for at least one of the later ciliating or mature ciliated cell populations. See Supplementary Fig. 7.

this technique, including wildtype and mono- or bi-allelic editing. ALI cultures derived from *POU2F3* KO cells significantly decreased expression of tuft-like, ionocyte, and PNEC marker genes (Fig. 7g). Moreover, *POU2F3* KO cultures exhibited strong depletion in the number of likely ionocytes (FOXI1+) and PNECs (*GRP*+) as assessed by wholemount IF or FISH microscopy, respectively (Fig. 7h, i). In contrast, *FOXI1* KO cultures exhibited depleted ionocyte marker expression and FOXI1+ cells while maintaining or increasing tuft-like and PNEC markers and cell counts (Fig. 7g–i). Together, these data support a branched lineage model (Fig. 7j) where both ionocytes and PNECs differentiate from *POU2F3*+ tuft-like cells, and whereby blockade of *FOXI1*-dependent ionocyte differentiation increases abundance of both tuft-like cells and PNECs.

**Rare cell types regulate normal epithelial electrophysiology**. As ionocytes and PNECs both express multiple specific ion channels (Fig. 7b), we hypothesized that *POU2F3* and *FOXI1* KO would alter the electrophysiological properties of their ALI cultures. Ussing chamber analysis (Supplementary Fig. 10a) revealed baseline hyperpolarization (increased transmembrane potential) in both KO cultures, along with decreased conductance and relatively unaffected current (Fig. 8a, Supplementary Fig. 11). These alterations are consistent with reduction in ion transport leading to buildup of chemical potential across ionocyte-depleted epithelia from both KOs, suggesting that ionocytes contribute to the maintenance of proper ion transport in these cultures, potentially at both paracellular[60] and transcellular levels (Supplementary Fig. 10b).

The hyperpolarization of *POU2F3* and *FOXI1* KO cultures could result from disruption of ionocyte-enriched CFTR, which would be consistent with CFTR mutant/KO data from mice and human cell cultures[61,62], as well as nasal potential difference measurements in individuals with cystic fibrosis[63]. Indeed, on a per cell basis, in vivo, ionocyte *CFTR* expression was between 19- and 547-fold higher than in other cell types (Fig. 8b). However, the low frequency of these cells (0.3% overall) means only 11% of total *CFTR* transcripts expressed by the epithelium were derived from ionocytes, whereas other more abundant cells contribute more total *CFTR* expression (e.g. *KRT8*high, 46%) (Fig. 8b). In addition, bulk *CFTR* expression began and peaked much earlier than other ionocyte marker genes over ALI differentiation (Fig. 8c), and was unaltered by *POU2F3* or *FOXI1* KO (Fig. 7g), further supporting a significant *CFTR* contribution from other epithelial cell types. This mRNA distribution was echoed at the protein level, where the highest concentration of apical CFTR signal localized to cells with FOXI1+ nuclei, but a considerable amount of CFTR signal resided in other non-FOXI1+ cells (Fig. 8d, Supplementary Movie 1). Finally, although *POU2F3* or

*FOXI1* KO severely dysregulated ion transport in ALIs, these cultures robustly retained CFTR activity, as measured by enhanced voltage response to forskolin/IBMX stimulation and CFTR(inh)−172 inhibition treatments in KO cultures relative to controls (Fig. 8e, Supplementary Fig. 10c, d). Altogether, these results suggest that FOXI1+ ionocytes, while contributing the densest CFTR and being critical for ion transport homeostasis, are not solely maintaining CFTR activity in the human tracheal epithelium.

We also found that during periods of intermittent short-circuit current measurement (Supplementary Fig. 10a), *POU2F3* KO cultures exhibited a reduced current response, whereas the response of *FOXI1* KO cultures was strongly enhanced in comparison to control cultures (Fig. 8f, Supplementary Fig. 10e). As neuronal cells highly express voltage-gated ion channels for the propagation of action potentials[64], the opposite direction of these responses between the two KOs is consistent with an effect caused by the under- and over-represented voltage-controlled PNECs that characterize *POU2F3* KO and *FOXI1* KO epithelia, respectively. As the measurement of short-circuit currents in this setting inhibits paracellular ion transport[60,65], these observations may also indicate a difference in the cultures' compensatory responses to this inhibition. Thus, while hyperpolarization takes place in both ionocyte-depleted KO cultures, the response to electrically mediated depolarization is dampened in PNEC-low *POU2F3* KO cultures and exaggerated in PNEC-high *FOXI1* KO cultures, revealing an important contribution of PNECs to airway epithelial ion transport function.

Although rare cells lacked most of the core smoking responses (Supplementary Fig. 3b), smoking induced remarkable transcriptional upregulation in tuft-like cells and downregulation in both ionocytes and PNECs (Fig. 8g). Downregulated DEGs in ionocytes and PNECs were enriched in defining markers of these cells, suggesting a loss of their specialized function in smokers (Fig. 8h, left). In contrast, upregulated DEGs in tuft-like cells were enriched in secretory and proteosomal basal cell markers, as well as in markers of ionocytes and PNECs (Fig. 8h, right). Smoking-induced shifts in the relative frequency of rare cell populations reflected these transcriptional responses. In non-smokers, tuft-like cell abundance tended to decrease with age, whereas PNECs increased and ionocytes or composite rare cells largely remained unchanged (Fig. 8i). This age-dependent shift in cellular composition away from tuft-like cells and toward PNECs tended to be enhanced in smokers, such that smokers exhibited rare cell ratios characteristic of older airways (Fig. 8i). Collectively, these responses in transcriptional programming and cellular composition are consistent with smoke exposure pushing tuft-like cells toward incomplete or compromised ionocyte or PNEC phenotypes, further supporting our proposed rare cell lineage.

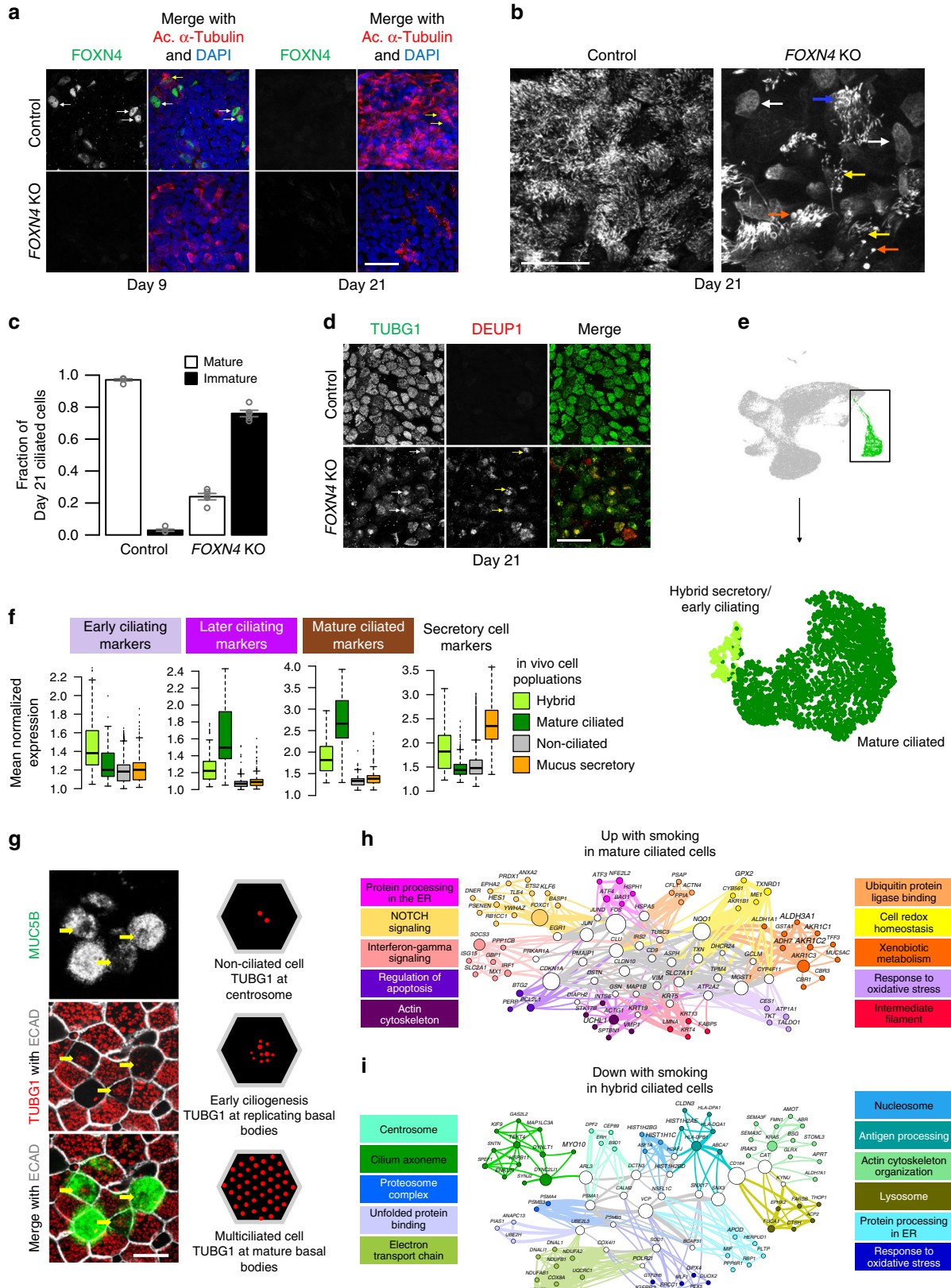

## Discussion

In this study, we have generated an agnostic atlas of the human in vivo tracheal airway epithelium among smokers and non-smokers, identifying and characterizing cell types, cell states, and lineage relationships among them. As such, our study expands on the mouse in vivo and human in vitro airway epithelial atlases published recently[17,18]. We also provide a much more densely sampled in vitro time course of human airway epithelial differentiation. Our data reveal that during both in vitro differentiation and in vivo homeostasis, ciliated cells derive from a secretory

**Fig. 6 Smoking inhibits the early ciliating state. a** Wholemount IF labeling of *FOXN4* KO in human tracheal ALI cultures. Arrows, immature (white) and mature ciliated cells (yellow). Scale bar = 25 μm. **b** Sample axonemal phenotypes by acetylated α-Tubulin IF labeling (white). Arrows, *FOXN4* KO cells displaying absent (white), short or sparse (yellow), or bulging (orange) axonemes were classified immature; dense, well-formed axonemes (blue) were classified mature. Scale bar = 25 μm. **c** Quantified fraction of mature and immature ciliated cells from control (*n* = 584) and *FOXN4* KO cultures (*n* = 854) in **b**. Bar plots: mean fractions ± standard error (*n* = 5 confocal fields); points: fraction for each field. **d** Wholemount IF labeling of *FOXN4* KO illustrates basal bodies are generated (γ-Tubulin, green), but fail to dock (white arrows) and deuterosomes are assembled (DEUP1, red), but retained (yellow arrows). Scale bar = 25 μm. **e** UMAP with subclustering of in vivo ciliated cells. Mature ciliated cells combine two subgroups (Supplementary Fig. 8b). **f** Average expression of markers from three in vitro ciliogenesis states and in vivo mature mucus secretory cells (far right). Box centers: median; upper/lower box bounds: first/third quartiles; upper/lower whiskers: extend from upper/lower bounds up to/down from the largest/smallest value, no further than 1.5× the inter-quartile range from upper/lower bounds; points: outliers. *N* (left to right) = 140; 2216; 30,866; 3026 cells. **g** (Left) Hybrid secretory/ciliated cells exhibit MUC5B labeling and early ciliogenesis phenotypes via TUBG1 morphology under confocal IF microscopy[82] (day 16 ALI cultures imaged 48 h post-DAPT). Scale bar = 10 μm. (Right) TUBG1 pattern schematic for non-ciliated, early ciliating, and mature multiciliated cells. **h** Functional gene network (FGN) of non-core upregulated genes in mature ciliated cells. Edges: connect genes annotated for the same enrichment terms; node colors: functional metagroups containing genes (exemplar terms, right); white nodes: hub genes (in multiple metagroups); node size: gene connectivity; label size: mean log fold-change between smokers and non-smokers; edge thickness: number of shared terms; edge color: metagroup membership of the connected node(s) if one or both are not hub genes; gray edges connect two hub genes. **i** FGN for genes downregulated by smoking in hybrid secretory/ciliated cells (network as described in **h**). See Supplementary Fig. 8.

---

progenitor through multiple, discrete, transcriptional states, regulated by a suite of TFs that include *FOXN4*, which we identify in humans as a regulator of this earliest ciliating state. Similarly, we show that the heterogeneity in secretory cells (club, mucus secretory cells expressing one or both of *MUC5B* and *MUC5AC*) is likely all part of a continuous secretory lineage that culminates in a multi-mucin producing mucus secretory cell.

Our atlas also produces the first transcriptional picture of human airway SMG cells, allowing us to identify a human equivalent to the recently described murine myoepithelial stem cell[41,42]. Although the stem cell function of the human myoepithelial will still need to be proven, our SMG lineage analysis and IF labeling is consistent with the human myoepithelial cell silencing its muscle expression program to assume both surface basal (*KRT5* and *KRT14*) and unique glandular expression (*SOX9*), similar to the mouse cell. This basal cell state can then differentiate into a mucus secretory cell, as orchestrated by TFs distinct from those involved in surface mucus secretory cell differentiation. The uniqueness of this program produces a vastly different secretory cell, with distinct mucin expression and processing and a specialized repertoire of secreted proteins. It remains unclear whether these SMG stem cells can repopulate the surface epithelium in humans as in mice[42,66].

Through a combination of scRNA-seq and CRISPR knockout studies we unveiled a tuft-like cell population, and we provide strong evidence that disease-relevant rare cell types are connected by a branched lineage which proceeds from tuft-like cells to both ionocytes and PNECs. This proposed lineage relationship between tuft-like cells, ionocytes, and PNECs may relate to our observation of an age-dependent balance of rare cell ratios and the recently reported tuft-like variants of small cell lung cancer, generally thought to be a PNEC-derived tumor[58]. We confirm that the homeostatic human airway epithelium does contain ionocytes and that they highly express *CFTR*. However, the large proportion of *CFTR* expression deriving from other epithelial cell types and our observation of *FOXI1*/*CFTR* decoupling by time course and knockout electrophysiology cautions against the simple model where cystic fibrosis is caused by loss of CFTR function in FOXI1$^+$ ionocytes. In fact, our data suggest that epithelial CFTR activity is not reduced by depletion in ionocyte numbers. Rather, our electrophysiology data define ionocytes as critical for homeostatic ion transport in the human airway epithelium in a fashion that involves other channels/transporters, but may still include CFTR. This critical role for ionocytes in epithelial ion transport may stem from their ion transport activity

alone and/or an ability to coordinate both paracellular and transcellular ion transport activity across the epithelium.

Importantly, we used scRNA-seq to deconstruct smoking effects on the epithelium to the cell-type level, which we have then reassembled into a comprehensive model of how smoking modifies epithelial function as a whole (Fig. 9). To summarize the composite epithelial responses to smoking, pan-epithelial effects of smoking reach the basal stem cells and include induction of chemokine signaling and xenobiotic metabolism at the expense of innate immune signaling. Surface secretory cells shift their mucin programs toward a *MUC5AC*-dominated inflammatory state while SMG secretory cells lose many of their distinctive defensive secretions. Hybrid secretory/early ciliating cells preferentially lose ciliogenic function, potentially hindering regeneration of ciliated cells upon injury, and tuft-like cells are pushed toward functionally impaired ionocyte- or PNEC-like states. We note these effects are based on sampling the tracheal airway. Future work will be necessary to extend our single-cell understanding of smoking effects to the small airways, which exhibit extensive pathology in smoking-related lung disease. Nevertheless, these data together paint a smoker epithelium that has been rendered more functionally monochromatic, collapsing on the secretory and proteosomal basal cell phenotypes at the expense of its normal defensive, interactive, and reparative roles essential to lung health and homeostasis.

## Methods
**Materials and correspondence**. Further information and requests for resources and reagents should be directed to and will be fulfilled by Max A. Seibold (seiboldm@njhealth.org)

## Experimental methods
**Key resources**. All reagents and resources referred to below are summarized with vendors and identifiers in Supplementary Data 1.

**Human trachea samples**. Fifteen human tracheal airway epithelia were isolated from de-identified donors whose lungs were not suitable for transplantation. Lung specimens were obtained from the International Institute for the Advancement of Medicine (Edison, NJ) and the Donor Alliance of Colorado. The National Jewish Health Institutional Review Board (IRB) approved the research under IRB protocols HS-3209 and HS-2240. We obtained informed consent from authorized family members of all donors. Smokers with at least 15 pack years were classified as heavy, whereas smokers with fewer than 5 pack years were classified as light. See Supplementary Table 1 for donor details. As only heavy smokers (*N* = 6) were used in our investigation of smoking effects, throughout the paper we simply refer to heavy smokers as smokers and to never-smokers (*N* = 6, excluding a pediatric donor) as non-smokers.

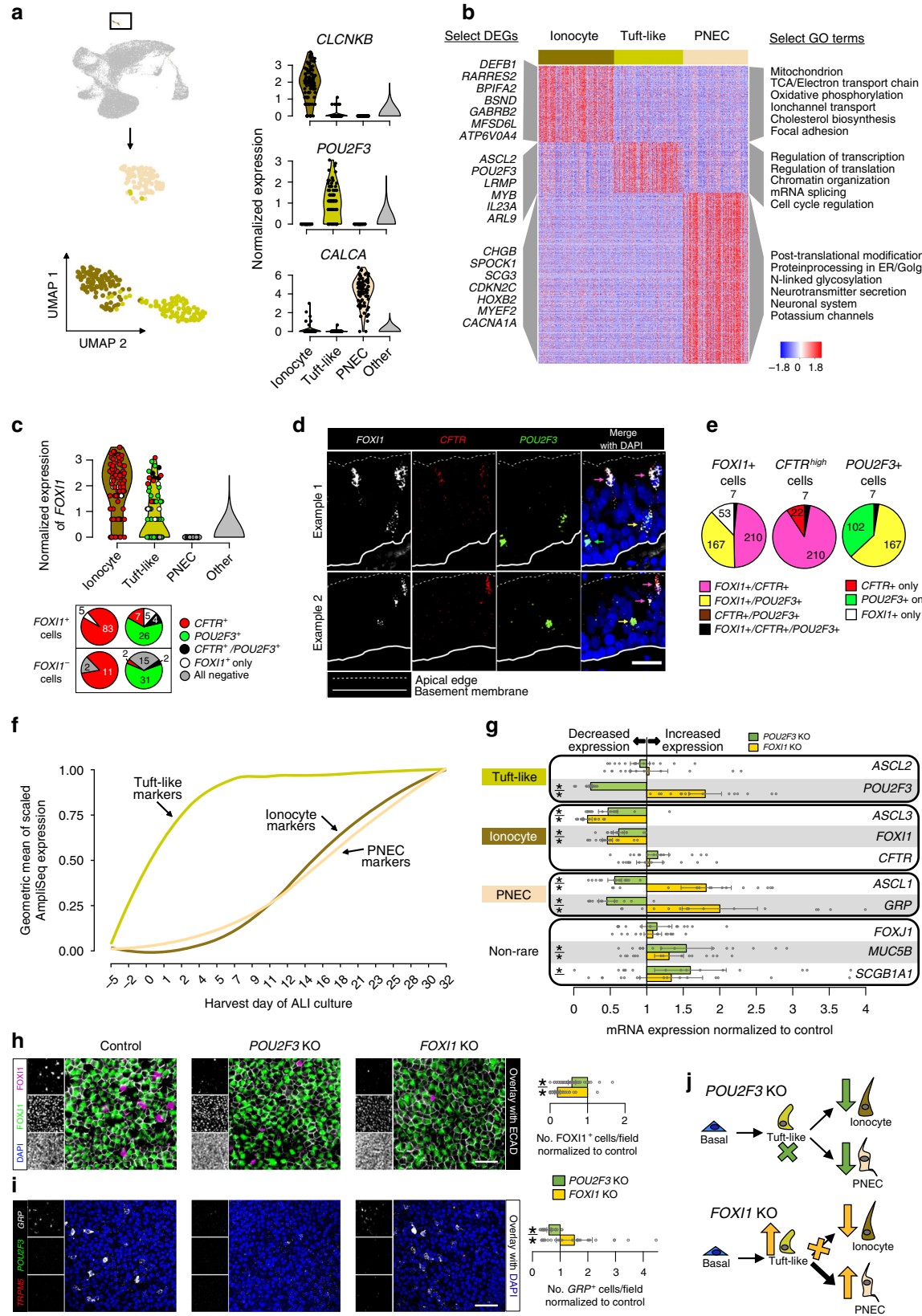

**In vivo harvest for scRNA-seq (10× Genomics).** All fifteen tracheal donors were used for single-cell sequencing. Human tracheas were wet in Stock solution (DMEM-F + 1× PSA), and fat and connective tissue were removed, before cutting into small sections. Sections were rinsed in Stock solution to remove mucus before proteolytic digest (0.2% Protease in Stock solution) overnight at 4 °C, with rocking.

Protease was neutralized with FBS, the supernatant was saved (tube 1), and tracheal sections washed (5 mM HEPES, 5 mM EDTA, and 150 mM NaCl) for 20 min at 37 °C. The supernatant was also saved (tube 2) and the loosened epithelium was then manually scraped off into stock solution with 10% FBS (tube 3), and all cells were collected by centrifugation for 10 min at 225×g, 4 °C (tubes 1, 2, and 3). Cell

**Fig. 7 A lineage relationship among rare epithelial cell types. a** Left, in vivo rare cell subclustering UMAP. Right, violin plots of rare cell markers. **b** Heat map of unique gene signatures across rare cell types. Right, select gene ontology terms enriched within signatures. **c** Top, Violin plots show *FOXI1* expression in tuft-like cells and ionocytes. Point color: co-expression of *FOXI1* with *CFTR* (red), *POU2F3* (green), or neither (white). Bottom, quantification of co-expression. **d** FISH illustrates *FOXI1* in ionocytes (*FOXI1*⁺/*CFTR*⁺, pink arrows) and tuft-like cells (*FOXI1*⁺/*POU2F3*⁺, yellow arrows) of human tracheal epithelium in vivo. Green arrow, *FOXI1*⁻/POU2F3⁺ tuft-like cell. Scale bar = 25 μm. **e** Average co-expression quantification of FISH in **d** (n = 561 total ionocytes/tuft-like cells imaged across 4 donors; number of cells indicated). **f** Geometric mean of scaled bulk mRNA expression from top 25 in vivo markers for each rare cell type. **g** qRT-PCR from ALI day 32 *POU2F3* or *FOXI1* KO cultures relative to controls. Bars: estimated normalized mRNA expression (n = 5 donors, three cultures each); estimates: coefficients from a linear model (donor set to random predictor); points: individual measures, normalized to control estimates; lines: estimate standard error; *p < 0.05, when compared to control with F-test, Satterthwaite approximation of degrees of freedom (p-values, top to bottom: 0.339, 0.750, 6.05e−8, 2.42e−8, 9.55e−5, 6.81e−8, 0.00189, 3.42e−5, 0.179, 0.728, 0.00268, 6.47e−7, 0.00158, 2.83e−7, 0.261, 0.503, 2.61e−4, 0.0291, 0.00259, 0.0782). **h** Left, Representative wholemount IF for FOXI1 (magenta, top inlay), FOXJ1 (green, middle inlay), DAPI (bottom inlay, excluded from overlay) from cultures in **g**; ECAD (white), cell boundaries. Scale bar = 25 μm. Right, IF quantification, bars: estimated number FOXI1⁺ cells (n = 5 donors, 10–13 fields each) based on linear model in **g**, normalized to controls, lines: standard error; *: as in **g** (top, p = 7.63e−5; bottom, p = 1.60e−30). **i** Left, Representative wholemount FISH for *GRP* (white, top inlay), *POU2F3* (green, middle inlay), *TRPM5* (red, bottom inlay) from cultures in **g**, overlay includes single DAPI slice (blue) for context. Scale bar = 50 μm. Right, FISH quantification, bars: estimated number *GRP*⁺ cells (n = 3 donors, 8 fields each) based on linear model in **g**, normalized to controls; lines: standard error; *: as in **g** (top, p = 7.67e−4; bottom, p = 0.0265). **j** Branched rare cell lineage schematic. See Supplementary Fig. 9 and Supplementary Table 2.

---

pellets resuspended in BEGM + 0.5× PSA were filtered using a 70 μm cell strainer, collected by centrifugation (5 min, 225×g, 4 °C) and cryopreserved in freeze media (F-media, 30% FBS, 10% DMSO). On the day of capture, cells were quick thawed, washed twice in 1× PBS/BSA (0.04%) and resuspended at 1200 cells/μL for capture on the 10× Genomics platform.

**Primary basal cell expansion**. Primary human basal airway epithelial cells from tracheal digests were expanded at 37 °C on NIH 3T3 fibroblast feeders in F-media (67.5% DMEM-F, 25% Ham's F-12, 7.5% FBS, 1.5 mM L-glutamine, 25 ng/mL hydrocortisone, 12 5 ng/mL EGF, 8.6 ng/mL cholera toxin, 24 μg/mL Adenine, 0.1% insulin, 75 U/mL pen/strep) with ROCK1 Inhibitor (RI, 16 μg/mL), and antibiotics (1.25 μg/mL amphotericin B, 2 μg/mL fluconazole, 50 μg/mL gentamicin)[67], and cryopreserved in freeze media upon initial passaging (P1).

**In vitro ALI culture (time course)**. For the in vitro time course (n = 20 timepoints), we used tracheal cells from three donors that included a heavy smoker, light smoker, and non-smoker used above for in vivo scRNA-seq. Tracheal cells (P1) were expanded a second time on feeders in F-media/RI and harvested by differential trypsinization with FBS neutralization. After washing in 1× PBS, cells were resuspended in 1× HBSS and subjected to DNase digest for 5 min at 37 °C. DNase was diluted 2-fold with HBSS, cells were centrifuged 225×g, 5 min, 4 °C, and seeded onto bovine collagen-coated 6.5 mm transwell inserts (2 × 10⁴ cells/insert) in ALI Expansion medium (50% BEBM, 50% DMEM-C, 0.5 mg/mL BSA, 80 μM ethanolamine, 10 ng/mL hEGF, 0.4 μM MgSO₄, 0.3 μM MgCl₂, 1 μM CaCl₂, 30 ng/mL retinoic acid, 0.8× insulin*, 0.5× transferrin*, 1× hydrocortisone*, 1× epinephrine*, 1× bovine pituitary extract*, 1× gentamicin/amphotericin*, *relative to BEGM Bullet Kit aliquot) with RI (day −5). RI was removed after 24 h, and ALI expansion medium was changed 48 h later (day −2). After another 48 h (day 0), apical medium was removed and basolateral medium was replaced with PneumaCult (PC)-ALI medium. Basolateral medium was exchanged for fresh PC-ALI every 48 or 72 h for the subsequent 11 days, and then daily for the following 22 days along with an apical wash of 20 μL PC-ALI.

**In vitro ALI culture (DAPT stimulation)**. Tracheal cells (P1) were expanded a second time on rattail collagen-coated dishes in PneumaCult Expansion Plus Medium (PEP)/RI and harvested by trypsinization with FBS neutralization. After washing twice in 1× PBS, cells were resuspended in ALI expansion medium with RI and seeded onto bovine collagen-coated 6.5 mm transwell inserts (2 × 10⁴ cells/insert) containing ALI Expansion medium with RI in the basolateral chamber (day −5). RI was removed after 24 h, and ALI expansion medium was changed 48 h later (day −2). After another 48 h (day 0), apical medium was removed and basolateral medium was replaced with PC-ALI medium. Basolateral medium was exchanged for fresh PC-ALI on day 2, 5, 7, 9, 11, and 13, with an apical wash of 20 μL PC-ALI on day 7, 9, 11, and 13. On day 14, all inserts were rinsed twice with PBS and DAPT (1μM final concentration in PC-ALI) was applied basolaterally. ALIs were harvested for wholemount IF labeling on day 16.

**In vitro harvest for scRNA-seq (WaferGen)**. For each timepoint, medium was removed from endpoint ALI cultures, and the apical chamber was washed with warm PBS/DTT (10 mM) for 5 min at 37 °C, followed by a warm PBS wash of both chambers. Cultures were dislodged from the insert with 200 μL apical dissociation solution (Accutase with 5 mM EDTA and 5 mM EGTA) for 30 min at 37 °C with occasional manual agitation[68]. Single-cell suspensions were diluted, centrifuged,

washed once with PBS/DTT and twice with PBS, before cryopreservation (F-media, 40% FBS, 10% DMSO). On the day of capture, cells were quick thawed, washed with 1× PBS (no BSA) and counted before proceeding with WaferGen capture according to the manufacturer's instructions with the following modifications: 5 or 10 × 10⁴ cells were stained per sample, single-cell candidates were confirmed by manual visual triage.

**CRISPR-Cas9 KO in human tracheal basal cells**. Two CRISPR RNA (crRNA) guides targeting human *FOXN4*, *FOXI1*, or *POU2F3* annealed with universal tracrRNA and complexed with Alt-R HiFi Cas9 nuclease (1:1.2 duplex:nuclease), were electroporated (3.1 μM RNP) into PEP/RI expanded human tracheal basal cells (5 × 10⁵ cells/transfection) with the Amaxa Nucleofector II (pulse code W-001). RNP-containing basal cells were then expanded on rat collagen-coated dishes in PEP medium with RI, and seeded onto transwell inserts (1 × 10⁵ cells/insert) in PEP medium (*FOXN4* KO) or ALI expansion medium (*FOXI1* and *POU2F3* KOs) with RI. RI was removed after 24 h, and 48 h later apical medium was removed and basolateral medium was replaced with PC-ALI medium (ALI day 0). Basolateral media was replaced with fresh PC-ALI every 48 or 72 h until harvest for wholemount IF labeling on the ALI day indicated (all KOs) or qRT-PCR, wholemount FISH, and Ussing analysis (*FOXI1* and *POU2F3* KOs). See Supplementary Fig. 8a for experimental schematic.

**IF microscopy (tissue and ALI section histological labeling)**. Adjacent cross-sections of human trachea were fixed in 10% neutral buffered formalin for >48 h at 4 °C. ALI cultures were washed with warm 1× PBS (5 min, 37 °C), fixed in PFA (1× PBS, 3.2% PFA, 3% sucrose) for 20 min on ice and washed twice with ice cold PBS. Tissue or ALI cultures were cut out of plastic supports, paraffin-embedded and sectioned onto microscope slides. Rehydration was performed with two 3 min washes in HistoChoice, followed by a standard ethanol dilution series, and antigen retrieval in Antigen Unmasking Solution for three 4-min boiling intervals. Slides were then cooled to room temperature on ice and washed three times with TBST (1× TBS, 0.5% Triton X-100) before blocking in Block-B buffer (1× TBS, 3% BSA, and 0.1% Triton X-100) for 30 min at room temperature. Double or triple primary antibody applications were performed in Block-B buffer overnight at 4 °C with dilutions as follows: KRT5 (1:500), TP63 (1:200), MKI67 (1:100), KRT8 (1:100), MUC5AC (1:500), MUC5B (1:200), KRT14 (1:200), DMBT1 (1:50), SCGB3A1 (1:20), ACTA2 (1:100), SCGB1A1 (1:500). Slides were washed three times in TBST before concurrent secondary application (1:500) in Block-B buffer with DAPI for 0 min and room temperature. Slides were again washed three times in TBST, mounted with ProLong Diamond Mount and imaged on an Echo Revolve R4 microscope.

**IF microscopy (ALI wholemount IF labeling)**. ALI cultures on transwell inserts were rinsed with 1× PBS, fixed for 15 min in 3.2% PFA (no sucrose) at room temperature or 10 min in methanol at −20 °C, rinsed twice more with 1× PBS and stored at 4 °C. Membranes were cut out of plastic supports and placed cell-side up on parafilm in a humid chamber. After three brief washes in TBST (*FOXN4* KO) or PBST (1× PBS, 0.1% TritonX-100; for DAPT, *FOXI1*, and *POU2F3* KOs), cells were blocked for 30 min at room temperature and primary antibodies were applied in Block-B for 2 h (*FOXN4* KO) or Block-H overnight (1× PBS, 10% Normal Horse Serum, 0.1% TritonX-100; for DAPT, *FOXI1*, and *POU2F3* KOs) at the following dilutions: FOXN4 (1:50), Acetylated α-Tubulin (1:1000), γ-Tubulin (1:500), DEUP1 (1:500), FOXJ1 (1:500), FOXI1 (1:50), ECAD (1:2500), CFTR (1:50), MUC5B (1:200). Membranes were washed three times in TBST (*FOXN4* KO) or PBST (for DAPT, *FOXI1*, and *POU2F3* KOs) before concurrent secondary application (1:500)

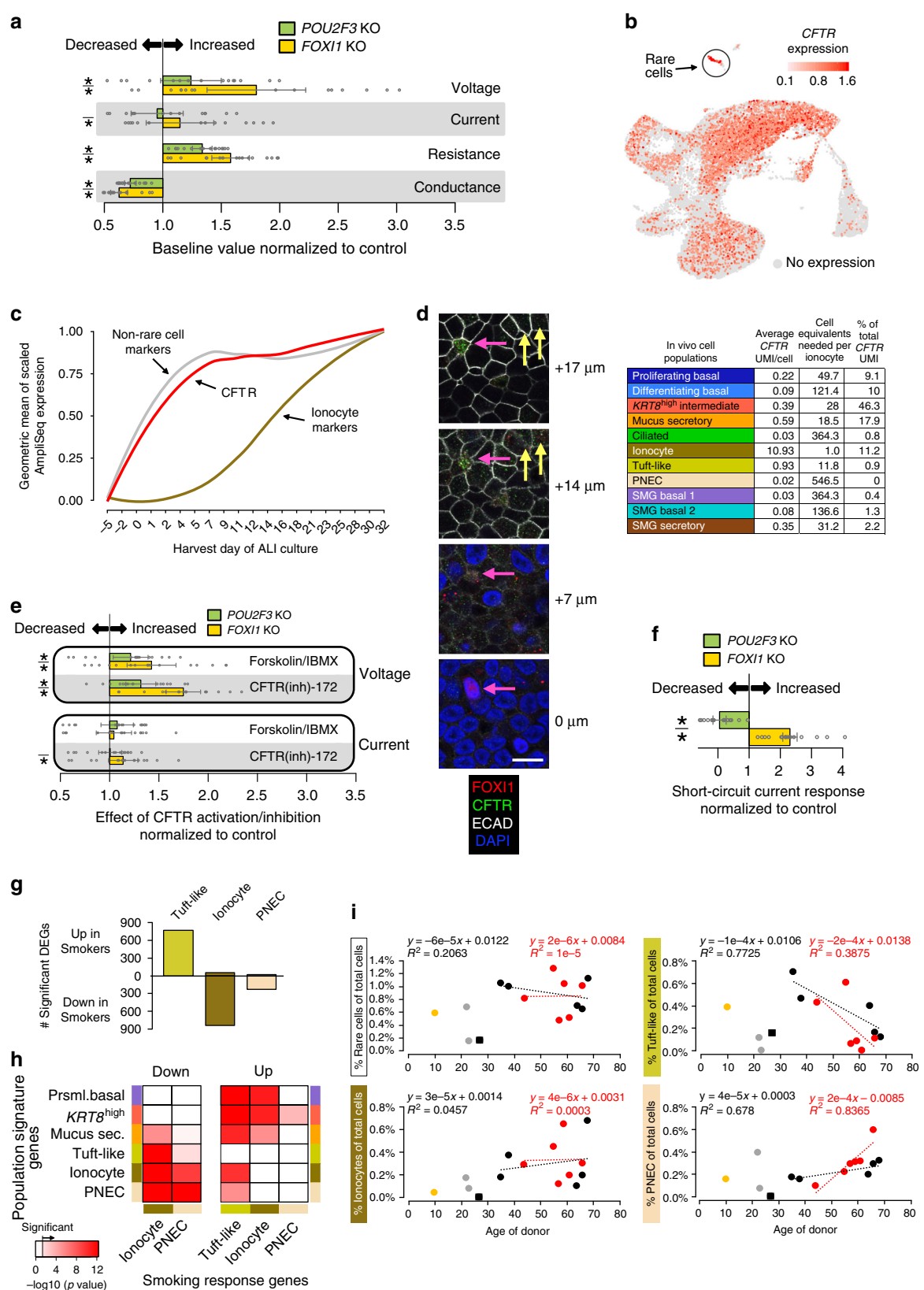

with DAPI for 30 min at room temperature. Membranes were again washed three times, mounted with ProLong Diamond (*FOXN4* KO) or Mowiol (10% Mowiol 4-88, 25% glycerol, 0.1 M Tris/pH8.5, 2% N-propyl gallate; for DAPT, *FOXI1*, and *POU2F3* KOs) and imaged on an Echo Revolve R4 (*FOXI1* and *POU2F3* KOs) or Leica TCS SP8 confocal (DAPT and *FOXN4* KO) microscopes.

**RNAScope FISH (tissue and ALI section histological labeling)**. Adjacent cross-sections of human trachea or PBS rinsed ALI cultures were fixed in 10% neutral buffered formalin for 24 ± 8 h at room temperature, washed with 1× PBS and paraffin-embedded immediately. Paraffin blocks were sectioned onto SuperFrost Plus slides, dried overnight, and baked for 1 h at 60 °C before immediately

**Fig. 8 Rare cells are critical for normal epithelial electrophysiology and compromised with smoking. a** Baseline electrophysiological parameters of KO cultures relative to controls (ALI day 32). Bars: estimated normalized parameter values, ($n = 4$ donors, four cultures each); estimates: coefficients from a linear model (donor set to random predictor); points: individual measures, normalized to control estimates; lines: estimate standard error; *$p < 0.05$, when compared to control with $F$-test, Satterthwaite approximation of degrees of freedom ($p$-values, top to bottom: 0.0157, 1.72e−10, 0.289, 0.00224, 9.63e−8, 5.38e−14, 1.66e−10, 2.91e−14). **b** Top, *CFTR* expression across UMAP of in vivo human tracheal epithelium. Bottom, Distribution of *CFTR* UMIs across major cell populations. **c** Geometric mean of scaled bulk RNA-seq expression for in vivo markers of non-rare cells, ionocytes, and *CFTR*, across 20 timepoints of ALI differentiation. **d** IF confocal microscopy of control day 32 human ALI cultures illustrates FOXI1 (red) and CFTR (green) cellular co-localization (four 1 μm slices of a 23-slice *z*-stack). Full *z*-stack in Supplementary Movie 1. ECAD (white), cell boundaries. Scale bar = 10 μm. **e** Electrophysiology of KOs treated with forskolin/IBMX, then CFTR(inh)−172, relative to controls. Plots are as in **a** ($p$-values, top to bottom: 4.98e−4, 2.50e−9, 3.85e−5, 8.47e−17, 0.0605, 0.282, 0.893, 0.00281). **f** Fluctuations in short-circuit current measurements of ATP-stimulated cultures. KOs are scaled to control cultures that were simultaneously clamped on the same Ussing apparatus. Plots are as in **a** (top, $p = 2.08e−6$; bottom $p = 8.44e−7$). **g** Number of DEGs was significantly altered in smokers vs non-smokers for rare cell types. **h** Enrichment of cell-type marker genes (rows) within rare cell smoker DEGs (columns) based on hypergeometric tests. Redness: enrichment level (−log10($p$-value)); white: non-significant. **i** Relationships between rare cell frequencies and donor age, by smoking status. See also Supplementary Figs. 10 and 11, and Supplementary Movie 1.

proceeding with the RNAScope Multiplex Fluorescent v2 assay according to the manufacturer's instructions with the following modifications: Target retrieval was performed for 15 min in a boiling beaker, Protease III was used for 30 min pre-treatment at 40 °C and hybridized slides were left overnight in 5× SSC before proceeding with the amplification and labeling steps. Opal fluors were applied at 1 in 1500 dilution, and slides were imaged on an Echo Revolve R4 microscope. Ionocytes and tuft-like cells were quantified by presence of grouped *FOXI1*, *POU2F3*, and/or *CFTR* puncta on triple labeled sections. Basolateral membrane length was quantified with the freehand line tool and measure functions in ImageJ.

**RNAScope FISH (ALI wholemount labeling).** ALI cultures on transwell inserts were rinsed with 1× PBS, fixed for 30 min in 3.2% PFA (no sucrose) at room temperature, rinsed twice more with 1× PBS and stored at 4 °C. Membranes were cut out of plastic supports and placed cell-side up on parafilm in a humid chamber for pre-treatment at room temperature: 70% ethanol for 1 min, 100% ethanol for 1 min, 1× PBS for 10 min, RNAScope Hydrogen Peroxide for 10 min. Membranes were rinsed twice with water, treated with Protease III (1:15 dilution) for 15 min at 40 °C and rinsed thrice with water before proceeding to at least 4 h of hybridization at 40 °C. Hybridized membranes were left overnight in 5× SSC before proceeding with the amplification and labeling steps in the humid chamber, but otherwise as specified by the manufacturer. Opal fluors were applied at 1 in 500 dilution, and slides were imaged on a Leica TCS SP8 confocal microscope.

**RNAScope FISH (tracheal digest cytospin labeling).** Tracheal cells prepared for 10X scRNA-seq were distributed on slides via centrifuge at 55.3×g, fixed for 15 min in 3.2% PFA/1.5% sucrose at room temperature, rinsed with 1× PBS and stored at 4 °C. Slides were subjected to pre-treatment at room temperature: 70% ethanol for 1 min, 100% ethanol for 1 min, 1× PBS for 10 min, RNAScope Hydrogen Peroxide for 10 min. Slides were rinsed twice with water, treated with Protease III for 15 min at 40 °C and rinsed thrice with water before proceeding to 4 h of hybridization at 40 °C. Hybridized slides were left overnight in 5× SSC before proceeding with the amplification and labeling steps, but otherwise as specified by the manufacturer. Opal fluors were applied at 1 in 500 dilution, and slides were imaged on an Echo Revolve R4 microscope.

**Wholemount IF/FISH quantitation.** Figure 6d: Ciliated cells were quantitated based on anti-acetylated α-Tubulin antibody labeling in maximum projections of confocal image stacks of control (584) or *FOXN4* KO (854) cells. "Mature" ciliated cells display numerous well-formed cilia, and "immature" cells display one of the indicated phenotypes: short, sparse, or bulging.
Figure 7h: FOXI1⁺/FOXJ1⁻ nuclei were quantitated from at least eight ×20 fields (Echo Revolve R4) for each KO treatment in four donors.
Figure 7i: PNECs were quantitated based on concentrated *GRP* labeling in maximum projections of confocal image stacks from eight ×40 fields (Leica TCS SP8 confocal) for each KO treatment in three donors.

**Quantitative RT-PCR.** On day 32 post-airlift, three replicate ALI cultures from each KO treatment (5 donors) were washed for 5 min with 1× PBS at 37 °C before direct lysis on the transwell insert. Lysates were thawed from −80 °C, and bulk RNA proceeded with the Zymo Quick-RNA Miniprep Kit including on-column DNase treatment. 200 ng of total RNA was reverse transcribed and duplicate reactions of 6 ng cDNA amplified the targets of interest via 5′ PrimeTime TaqMan assays. Raw Ct values were scaled to *GUSB* housekeeping loading control before downstream analysis.

**Ussing chamber analysis.** On day 32 post-airlift, four replicate ALI cultures from each KO treatment (4 donors) were subjected to electrophysiological analyses in an Ussing Chamber (Physiologic Instruments) under open-circuit conditions, where

intermittent short-circuit current measurements with pulsing (200 ms pulses at ±5 mV) was performed to obtain resistance and conductance values. Cells were symmetrically bathed in a modified Ringer's solution (120 mM NaCl, 10 mM D-Glucose, 3.3 mM $KH_2PO_4$, 0.83 mM $K_2HPO_4$, 1.2 mM $MgCl_2$, 1.2 mM $CaCl_2$, 25 mM $NaHCO_3$, pH 7.4). Cultures were treated acutely in the Ussing chamber with apical 100 μM amiloride, apical/basal 20 μM forskolin/100 μM IBMX, apical 10 μM CFTR(inh)−172 and apical 100 μM ATP.

**AmpliSeq of bulk RNA samples.** For each donor/timepoint, $0.5–3 \times 10^5$ cells were aliquoted from those harvested for scRNA-seq and bulk RNA was extracted with the Quick-RNA Microprep Kit including on-column DNase treatment. Isolated RNA was normalized to 3.5 ng input/sample for automated library preparation with the Ion AmpliSeq Transcriptome Human Gene Expression Kit, using 12 cycles of amplification. Libraries were sequenced with the Ion Proton System (ThermoFisher Scientific).

## Statistics and reproducibility

Figures 1d–h, 3d, Supplementary Figs. 1f, 6e: representative images from at least twenty ×20 fields from each of at least two donors.
Figure 4g: representative images from at least ten ×20 fields from each of two donors.
Figure 5a: representative images from at least ten ×20 fields from each of three donors for each timepoint.
Figure 6a, d: representative images from at least four fields from each of two independent experiments in cultures from a single donor.
Figure 6b: representative images from 584 control and 854 KO cells across five fields from two independent experiments in cells from a single donor.
Figure 6g: representative images from at least four fields from each of two donors' cultures.
Figure 7d: representative images from at least 2 cm of contiguous basolateral membrane from each of four donors (8.4 cm total).
Figure 8d, Supplementary Movie 1: representative images from at least four fields from each of four donors.
Supplementary Figure 2: representative images from at least eight fields from each of two non-smokers and two heavy smokers.
Supplementary Figure 9d: representative images from at least ten ×20 fields from the two timepoints indicated for cultures from each of two donors.

## Quantification and statistical analysis

**Pre-processing of in vivo scRNA-seq data.** Initial pre-processing of the 10× in vivo scRNA-seq data, including demultiplexing, alignment to the hg38 human genome, and UMI-based gene expression quantification, was performed using Cell Ranger (version 3.0, 10× Genomics).
We next carried out donor-specific filtering of cells to ensure that high quality single cells were used for downstream analysis. Although samples comprising multiple cells were removed during the cell selection stage, we safeguarded against doublets by removing all cells with either a gene count or UMI count over the 99th percentile. Furthermore, we removed cells exhibiting fewer than 1500 genes or >40% of mapped reads originating from the mitochondrial genome. Before downstream analysis, select mitochondrial and ribosomal genes (genes beginning with MTAT, MT-, MTCO, MTCY, MTERF, MTND, MTRF, MTRN, MRPL, MRPS, RPL, or RPS), or very lowly expressed genes (expressed in <0.1% of cells) were also removed. The final quality-controlled dataset consisted of 40,929 cells. After initial clustering and visualization, which allowed for identification of 12 major cell populations (Supplementary Fig. 1a), 2461 cells were removed that we characterized as non-epithelial (immune, endothelial, and mesenchymal cells). We then repeated the same QC filtering above for this culled dataset (except further

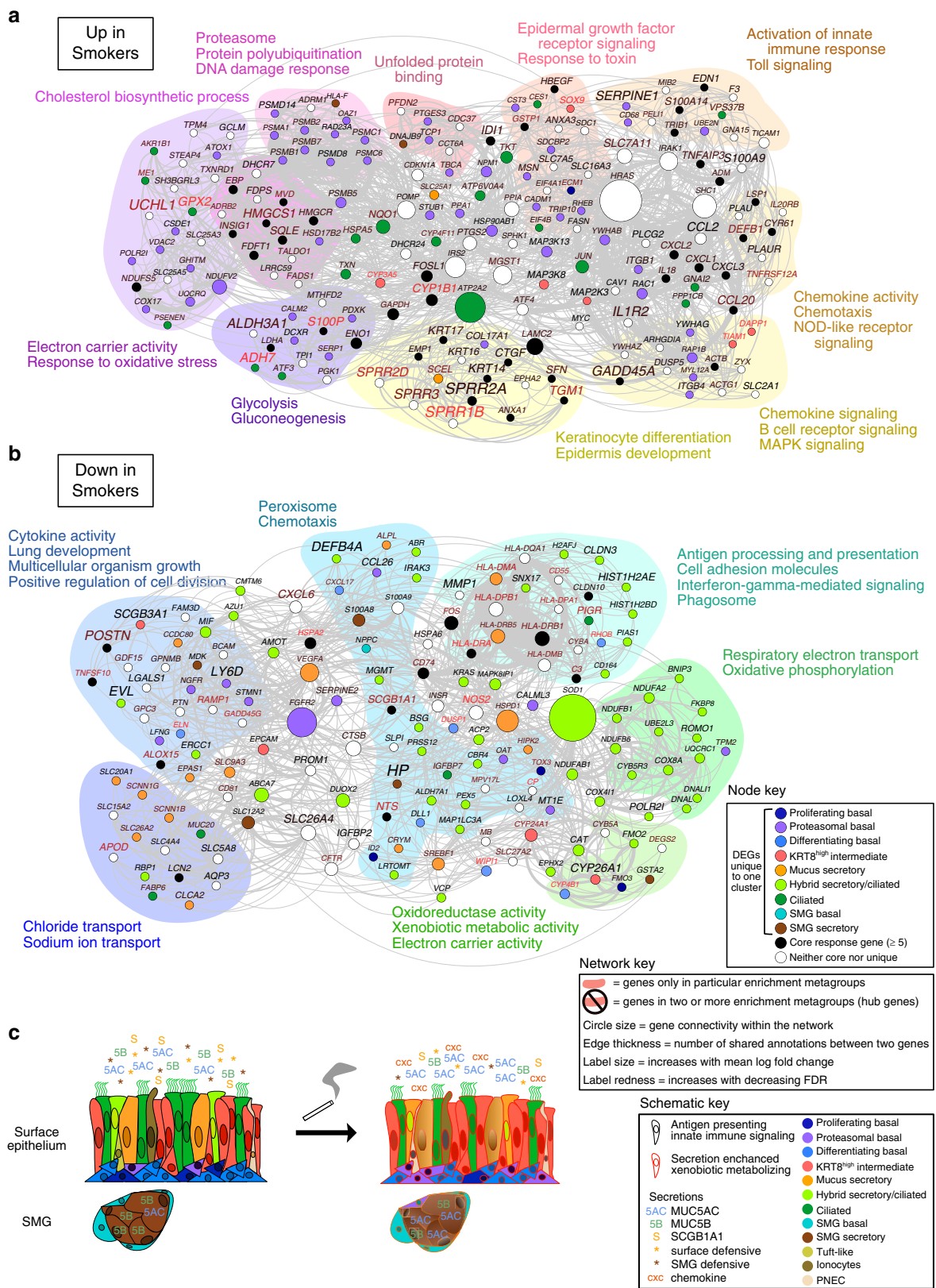

reducing the percent mitochondrial reads cutoff to 30%), which left us with 36,248 epithelial cells for further analysis.

To account for differences in coverage across cells, before downstream analysis we normalized and variance stabilized UMI counts for each donor using SCTransform[69], which yields Pearson residuals from a generalized linear model that includes sequence depth as a covariate. The Seurat R package[70] was used to carry out all data normalization and scaling as well as downstream batch

correction, dimensionality reduction, clustering, visualization, and differential expression.

**Pre-processing of in vitro scRNA-seq data.** We trimmed and culled raw demultiplexed cDNA reads in FASTQ files using Cutadapt[71], trimming poly A tails and 5′ and 3′ ends with $q < 20$ and removing any reads shorter than 25 base pairs.

**Fig. 9 Whole epithelium smoking responses reconstructed from cell-specific scRNA-seq analyses. a** Functional Gene Network (FGN) based on all genes upregulated with smoking (excluding rare cells) shows how genes that respond to smoking in distinct cell types of the airway epithelium may collaborate in carrying out dysregulated function. Node (i.e. gene) colors in the node key refer to the cell type in which a gene was differentially expressed if unique; nodes for semi-unique and core DEGs are white and black, respectively. Edges connect genes annotated for the same enriched term. Exemplar enriched functions are given next to each functional metagroup (or category), which are indicated by the underlay colors that encompass all genes annotated only for the terms within the metagroup. Nodes without colored underlay represent genes in multiple metagroups. Other properties of the network, including node size, connecting edge thickness, and label size/redness are defined in the network key. **b** FGN as in **a**, but for all genes downregulated with smoking in the airway epithelium. Legend serves for both **a** and **b**. **c** Schematic summarizes the smoking response of the whole epithelium.

Trimmed reads were then aligned to the hg38 human genome with GSNAP[72], setting "max-mismatches=0.05" and accounting for both known Ensembl splice sites and SNPs. Gene expression was quantified using HTSeq[73] with "stranded=yes", "mode=intersection-nonempty", and "t=gene" and then summed the number of unique molecular identifiers (UMIs) for each gene across runs for each cell to obtain a UMI count matrix used for all downstream analysis.

We carried out quality-control filtering using a similar approach to that used for the in vivo dataset, removing 100 cells for which the percentage of reads mapping to genes was <50%, 2262 cells with >25% of mapped reads being mitochondrial, and 384 cells with UMI counts outside the 3rd and 97th percentiles. We filtered genes using the procedure described above for the in vivo dataset. The final quality-controlled dataset consisted of 5976 cells, and 23,825 genes.

**Pre-processing of in vitro bulk AmpliSeq data.** Reads sequenced on the Ion Torrent Proton sequencer were mapped to AmpliSeq transcriptome target regions with the torrent mapping alignment program (TMAP) and gene count tables were generated for uniquely mapped reads using the Ion Torrent ampliSeqRNA plugin. After removing duplicated sequences our dataset contained 20,869 genes for 20 sampled timepoints. We normalized expression based on size factors calculated using DESeq2[74].

**Dimensionality reduction, clustering, and visualization.** Before clustering and visualizing each of the two scRNA-seq datasets (in vitro and in vivo), we reduced the dimensionality of variation in a way that accounted for batch-based shifts in expression among donors. To do this in the in vivo dataset, we used single-cell integration[75] implemented in Seurat v3, which identifies mutual nearest neighbor (MNN) cells across pairwise donors to use as "anchors" by which a batch correction can be calculated and then applied in order to bring expression across datasets into a common subspace. We carried out the integration analysis using the top 30 dimensions from a canonical correlation analysis (CCA) based on SCTransform normalized expression of the top 3000 most informative genes across donor datasets, where "informativeness" was defined by gene dispersion (i.e., the log of the ratio of expression variance to its mean) across cells, calculated after accounting for its relationship with mean expression (using the SelectIntegrationFeatures function). With integrated expression values in hand, we carried out principle component analysis (PCA) and then clustered and visualized the integrated data using the top 30 dimensions. For visualization, we reduced variation to two dimensions using Uniform Manifold Approximation and Projection (UMAP; n.neighbors = 10, min.dist = 0.35). Furthermore, we carried out unsupervised clustering by constructing a shared nearest neighbor (SNN) graph based on $k$-nearest neighbors ($k = 20$) and then determining the number and composition of clusters using a modularity function optimizer based on the Louvain algorithm (resolution = 0.22).

For the in vitro dataset, we used the single-cell alignment approach in Seurat v2[70] to carry out batch correction. First, CCA was used to identify the strongest components of gene correlation structure that were shared across donors (using Seurat's RunMultiCCA function) based on the union of the top 10,000 most informative genes involving two or more of the three donors (9542 genes total). Correlated expression across donors based on the top 25 CCA dimensions was then projected into a common subspace using Seurat's AlignSubspace function. We further reduced dimensionality of these 25 subspace-aligned CCA dimensions using the Barnes–Hut implementation of $t$-distributed neighborhood embedding (tSNE) and then plotted cell coordinates based on the first two dimensions (perplexity = 80). We further carried out SNN clustering as with the in vivo dataset, except using an SLM optimizer with $k = 15$ and resolution = 0.4.

**Subclustering.** We clustered subsets of cells from the in vivo dataset to further characterize cell types and states obscured by global heterogeneity. We subclustered four different subsets: (1) KRT8[high] and mucus secretory populations, (2) SMG basal and SMG secretory populations, (3) ciliated cells, and (4) rare cells. When there were a sufficient number of cells per donor (KRT8[high]/secretory and SMG cells), we reintegrated the subsetted cells prior to clustering. In the latter two cases, we simply clustered subsetted cell using the original globally integrated (albeit rescaled) expression. All integration and subclustering were done using the same approach, outlined above for the entire in vivo dataset. The specific parameters for analysis of each of these four subsets that differ from those above are as follows: (1)

KRT8[high]/secretory cells: Dataset integration was carried out with the top 3000 genes, UMAP was created with n.neighbors = 50, min.dist = 0.3, and SNN clustering was done using the Louvain algorithm, k.param = 50, and resolution = 0.54. (2) SMG cells: dataset integration was carried out with the top 3000 genes (although all integrated expression was calculated for all genes, given sufficient computational resources to do this) and with k.filter = 60, UMAP was created with n.neighbors = 50, min.dist = 0.3, and SNN clustering was done using the Louvain algorithm, k.param = 30, and resolution = 0.1. (3) Ciliated cells: The top 1500 most informative genes and 22 PCs were used for subclustering. UMAP was created with n.neighbors = 10, min.dist = 0.5 and SNN clustering was done using the Louvain algorithm, k.param = 20, and resolution = 0.1. (4) Rare cells: The top 1500 most informative genes and 30 PCs were used for subclustering. UMAP was created with n.neighbors = 10, min.dist = 0.3 and SNN clustering was done using the Louvain algorithm, k.param = 20, and resolution = 0.1.

**Plotting expression across cells.** For overlaying expression onto the UMAP/tSNE or dot plots for single genes or for the average across a panel of genes, we plotted normalized expression along a continuous color scale. All heat maps showing gene expression across cells (except Fig. 3a and Supplementary Fig. 6g, which were created using Monocle) were produced using Heatmap3[76] and also show scaled normalized expression along a continuous color scale, with break scales set as indicated.

To help minimize the influence of outliers, average or mean expression across cells was always calculated as the geometric mean, where a fixed value of 1 was added to each normalized count to avoid taking the log of zero.

**Differential expression analysis.** Differential expression (DE) for each gene between various groups specified in the text that contain a mixture of smoking habits (i.e., between cell-type populations) was tested using a logistic regression (LR) test that contained smoking habit as a latent variable. For DE comparisons between smokers and non-smokers, we used a non-parametric Wilcoxon-rank sum test. For DE comparisons between cell-type populations containing only non-smokers, we used a LR test with donor identity included as a latent variable. We limited each comparison to genes exhibiting both an estimated log fold-change > +0.25 and detectable expression in >10% of cells in one of the two groups being compared. We corrected for multiple hypothesis testing by calculating FDR-adjusted $p$-values. Genes were considered to be differentially expressed when FDR < 0.05.

**Functional enrichment analysis.** We tested for gene overrepresentation of all target lists within a panel of annotated gene databases (Gene Ontology [GO] Biological Process [BP] 2018, GO Molecular Function [MF] 2018, GO Cellular Component [CC] 2018, Kyoto Encyclopedia of Genes and Genomes [KEGG] 2019, and Reactome 2016) using hypergeometric tests implemented with Enrichr[77], as automated using the python script, EnrichrAPI [https://github.com/russell-stewart/EnrichrAPI]. We report only terms and pathways that were enriched with FDR < 0.05.

**Identification and plotting of cell-type-specific markers.** To identify cell-type-specific markers for both the in vitro and in vivo datasets, we first carried out pairwise differential expression analysis between each of the major clusters, downsampling large clusters to the median cluster size. Markers for each cluster were those genes exhibiting significant upregulation (FDR < 0.05) when compared against all other clusters. Markers for each cluster were sorted by FDR as calculated based on largest $p$-value observed for each gene across comparisons.

For plotting expression of in vivo rare cell types across differentiating AmpliSeq bulk samples in vitro, we first obtained in vivo cell-type markers for each of the three rare cells by isolating genes that were both significantly upregulated in each rare cell type relative to one another (with FDR < 1e−5) and when compared to all non-rare cells (with FDR < 1e−5). We then plotted the geometric mean of log-normalized expression in bulk across the top 25 in vivo markers for each cell type (based on FDRs in the non-rare cell comparisons), after scaling values to be between zero and one (Fig. 7f). For the expression of non-rare in vivo cluster markers shown in Fig. 8c, we took the geometric mean of log-normalized expression across the top 25 markers (or as many as available if fewer than 25) for each of the main non-rare in vivo cell clusters.

For the marker UMAP/tSNE overlay plots in Supplementary Fig. 7a, for each in vitro and in vivo cluster, we calculated average expression across the top 100 markers (or as many as were available, if fewer than 100) and then used these values to show characteristic expression of select in vitro cell types on the in vivo cells and vice versa. Because the Secretory cell 1 in vitro cluster was transitional and consequently had only a single marker, we used the top 25 most distinct genes for this population, despite these markers not being significant by FDR < 0.05. Cells on the UMAP/tSNE plots were identified as being characteristic of a given cell type if marker mean expression for that cell type was at least in the 85th percentile, whereas marker mean expression for all other cell types was below the 85th percentile. To render the in vitro secretory cell 1 and 2 populations more distinguishable, we increased stringency for expression of secretory cell 2 markers to a 95th percentile cutoff.

**Defining core and unique smoking response genes.** Core smoking response genes were defined as those significantly differentially expressed in smokers compared to non-smokers in five or more of the main in vivo populations (Fig. 1b). For this analysis, we excluded the rare cell population, which generally contained too few cells to detect significant smoking DEGs, and used a ciliated population from which a small subpopulation of hybrid secretory/early ciliating cells was removed, as the smoking response in this small population was so distinct. Unique response genes were those that were significantly differentially expressed in only one population while exhibiting either a log fold-change <0.25 and/or an FDR > 0.2 in all other populations. Genes responding to smoking in a least one cell population but not considered unique or core were defined as semi-unique. Populations assessed for uniqueness in this way were the three basal cell populations, $KRT8^{high}$, mature ciliated (with hybrid cells remove), SMG basal, SMG secretory, and surface secretory cell populations. In a separate analysis, uniqueness of smoking DEGs in the hybrid secretory/early ciliating subpopulation and three rare cell populations was also assessed using the above approach, by comparing these groups to all the main populations as well as to each other.

**Calculating correlations with MUC5AC and MUC5B.** To find genes in surface secretory cells (excluding the hybrid secretory/ciliated subpopulation) whose expression was correlated with that of *MUC5AC* or *MUC5B*, we used Spearman partial correlation analysis, which calculated gene correlations while controlling for differences in expression due to smoking. Light smoker/pediatric donor cells and cells that did not express both *MUC5AC* and *MUC5B* were excluded from this analysis.

**Lineage trajectories.** For the in vivo dataset, we constructed lineage trajectories for $KRT8^{high}$ and mucus secretory cell populations combined and for SMG cells to better understand the genes and processes that regulate and transition across these two lineages. For the $KRT8^{high}$-to-secretory cell trajectory, we first integrated expression data across donors from these two populations using the strategy outlined above, based on the top 3000 informative genes. Then, using Monocle v2.8[26], we carried out dimensionality reduction on the integrated expression using the DDRTree algorithm, and then ordered cells along a trajectory of pseudotime (using Monocle's orderCells function; see Supplementary Fig. 5b) based on expression of the 3000 genes. We then tested each gene for differential expression as a function of pseudotime, with smoking status as a covariate, after which we hierarchically clustered the significantly correlated genes (those with q-value < 1e −10), and then used the plot_pseudotime_heatmap function to plot smoothed scaled expression of genes belonging to four major phases across cells sorted by pseudotime, assuming that the most basal-like cells occupy the initial state (Fig. 3a). To view the expression of key regulators across the trajectory on a shared scale, we normalized all smoothed expression values to be between zero and one, and then plotted these normalized expression curves across pseudotime (Fig. 3b).

This same approach was followed for constructing the SMG cell trajectory. For this trajectory, we used two of three SMG basal substates (excluding myoepithelial cells due to a lack of intermediates) and SMG secretory cells inferred from subclustering (described above). To order the subclustered cells (Supplementary Fig. 6g), we used the top 1000 most informative genes. Smoothed expression across 16 clusters of genes significantly associated with pseudotime (q-value < 1e−10) were plotted across the trajectory, assuming that SMG basal A cells occupy the root state (Supplementary Fig. 6g).

In addition, we constructed lineage trajectories for the in vitro dataset, allowing us to capitalize on the known real-time appearance of cell states across differentiation of ALI cultures. Applying the previously calculated tSNE dimensions, we used Slingshot[44] to build lineages of cells that link in vitro SNN cell clusters by fitting a minimum spanning tree (MST) onto the clusters. When constructing these lineages, we only used differentiating basal, secretory, and ciliating/ciliated populations, where lineages were constrained to begin with the differentiating basal population and to end with either the mature secretory (secretory cell 2) or mature ciliated populations. We inferred two major lineages, one defining the transition from differentiating basal to secretory cell 1 then secretory cell 2, and the other defining the transition from differentiating basal to early and late ciliating cells, and then on to mature ciliated cells. Pseudotime values for cells were obtained for each lineage by projecting cells onto smoothed lineages constructed using Slingshot's simultaneous principle curves method. For the two lineages, we then plotted smoothed scaled expression (as a weighted average across a 100 cell window) of select genes that were significantly associated with pseudotime based on Monocle's differential gene test (q-value < 0.05) (Supplementary Fig. 6g).

**Isolating mature secretory cells.** The human tracheal epithelia we sequenced contained cells that transcriptionally fall along a developmental continuum, advancing from early differentiation basal-like cells to mature secretory cells (Supplementary Fig. 5a, b). To identify the terminal population of cells in this continuum, we modeled pseudotime as a mixed Gaussian, under the assumption that the distribution of cell densities along branches versus at end points will differ. After fitting the mixed Gaussian model to cells belonging to the transcriptional state inferred using Monocle that contains the terminal branch and endpoint of the trajectory (Supplementary Fig. 5b), we assigned any cells with a pseudotime greater than two standard deviations to the left of the second (endpoint) distribution as belonging to a mature secretory cell state.

**Spliced/unspliced ratios.** It has been shown that unspliced and spliced mRNA molecules capture earlier and later expression states, respectively, thus providing temporal information that is distinct from expression of combined RNA-seq data[78]. Thus, to further test the polarity of the later ciliating and mature ciliated expression states in the ALI cultured scRNA-seq dataset, we used the Velocyto pipeline[78] (applying default options) to identify unspliced and spliced mRNA reads for each gene in the original in vitro BAM files.

**Gene networks.** We constructed functional gene networks (FGNs) in order to summarize the major processes being carried out by selected gene sets in a way that shows the genes involved and their interconnectivity. FGNs were created by finding enriched terms for the given gene set (based on Gene Ontology and KEGG Pathway libraries), filtering and consolidating these enrichments into categories (i.e. metagroups) using GeneTerm Linker[79], and then constructing gene networks based on select metagroups using FGNet[80], which connects genes via edges with shared annotations that fall within a particular metagroup. Genes (i.e., nodes) uniquely involved in distinct processes (i.e., metagroups; each with a different colored border) can be distinguished from those involved in multiple processes (nodes belonging to multiple metagroups, indicated with white borders). Edges indicate at least one shared annotation.

**Reporting summary.** Further information on research design is available in the Nature Research Reporting Summary linked to this article.

## Data availability

Gene lists and other source data associated with Figure and Supplementary Figure panels can be found in Source Data. All raw and processed scRNA-seq data used in this study have been deposited in the National Center for Biotechnology Information/Gene Expression Omnibus (GEO) with accession number GSE134174.

## Code availability

The code used to produce analyses and figures in the study can be found on GitHub [github.com/seiboldlab/SingleCell_smoking].

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

## Acknowledgements

This work was supported by the National Jewish Health Regenerative Medicine and Genome Editing Program (REGEN), the Cystic Fibrosis Foundation (BRATCH16I0), the Eugene F. and Easton M. Crawford Charitable Lead Unitrust, and NIH grants R01 HL135156, R01 MD010443, R01 HL128439, P01HL132821, and P01 HL107202. We would like to thank Dr. HongWei Chu, Dr. Reem Al Mubarak, and Nicole Pavelka in the NJH Live Cell Core, M.R.G. and A.Q.G., as well as Dr. Carolyn Morris, Dr. Yingchun Li, Ari Stoner, Dr. Meghan Cromie, Dave Heinz, Katrina Diener, and Todd Woessner for assistance with tissue processing, sequencing, and useful discussion.

## Author contributions

Conceptualization, K.C.G. and M.A.S.; methodology, K.C.G., N.D.J., S.P.S., and M.A.S.; software, N.D.J., S.P.S., N.D., and K.S.L.; validation, K.C.G., N.D.J., E.K.V., and M.A.S.; formal analysis, N.D.J, S.P.S., N.D., E.G.P., and K.S.L.; investigation, K.C.G., C.L.R., M.T.M., J.L.E., P.E.B., and E.K.V.; resources, K.C.G., C.L.R., J.L.E., E.K.V., and M.A.S.; writing—original draft, K.C.G., N.D.J., and M.A.S.; writing—review and editing, K.C.G., N.D.J., E.K.V., and M.A.S.; visualization, K.C.G., N.D.J., S.P.S., N.D., K.S.L., E.G.P., E.K.V., and M.A.S.; supervision, K.C.G., N.D.J., E.K.V., and M.A.S.; funding acquisition, P.E.B. and M.A.S.

## Competing interests

The authors have no competing interests.
