## [Peer Review File · Nature Communications]

Reviewers' comments:

Reviewer #1 (Remarks to the Author):

The authors applied scRNA-seq to identify cell types and states of the tracheal airway epithelium in smokers and nonsmokers, infer the lineage relationship among these cells, and determine the influence of cigarette smoke on individual airway epithelial cell types. This study provided insights into cellular heterogeneity in human airway epithelium, lineage relationships, and cell type specific responses to smoking. The dataset and study could be a useful resource in the field.

Major comments:

1. The cell names are not clearly and consistently defined. Each figure appears to have different cells not clear how they defined and how they related (Fig1b, 2a, 3d, 4a, 5a). The authors presented several clustering analyses using different subsets of cells and summarized the clustering specifications in Supp Table 5, it is difficult to read the text and figures while map clustering results back to Supp Table 5. No complete cluster information was shown in the figures; instead, clusters were replaced with cell types/states in the different figures, some of which do not have the same number types/states as the number of clusters. It is unclear how clusters were mapped to those cell types/states and any clusters were merged or removed in different analyses. Providing tSNE/UMAP visualizations of complete cell clusters and the mapping of clusters to cell types/states in their corresponding figures will greatly help understand the analysis. The authors should be very clear about the total number of major cells and associated subtypes (in one t-SNE or several associated t-SNEs in one figure) and explain the evidence supporting the definition (e.g., early vs later vs maintenance vs mature ciliated or secretory cells, basal vs basal-like, differentiating and intermediate basal).

2. While pseudotime lineage predictions and interpretations are interesting, the figure presentation is not easy to visualize (Fig2b, 3e). It would be nice to see the pseudotemporal ordering of the single cells along each differentiation lineage as in many of the publications. The authors should provide further evidence from the literature or their own validation to support the novel predictions (e.g., tuft-like to ionocytes). Any evidence to support the transdifferentiation of mature mucous cells to mature ciliated cells? In Fig1, these two populations are not close to each other. Similarly, the rationale for why they use four gene modules to represent secretory cell lineage instead of the previous defined cell types and subtypes and how these gene modules in relationship with the cell subtypes is not clearly explained.

3. The authors stated in abstract and multiple other places in the manuscript that they generated a first comprehensive atlas of human airway epithelial cell types, states, lineages, and cell-specific responses to smoking which will be a useful resource to the field. But neither raw nor processed scRNA-seq data have yet submitted to any public repository (e.g., dbgap or GEO). It is required by the journal to make the data accessible. A public website to host and query these data will be a nice addition of the manuscript and useful tool to the field.

Specific comments:

1. Fig1i presents one of the major results of cell type specific responses to smoking. Nevertheless, the methods used to reach the conclusions were not clearly presented.

(a) Please describe the analysis. What's the gene list? How many of the genes from the bulk RNA-seq analysis were able to be mapped to the single cell data and used in the analysis? Any of those genes

differentially expressed between nonsmokers and heavy smokers in any cell types?

(b) What does the “mean normalized expression” mean? The variances of individual genes in individual cells were not shown. “Mean” operations can be greatly affected by outliers. Fisher-exact test can be used to check whether the smoking genes from bulk RNA-seq are enriched in specific airway cells.

2. Fig1j showed the common and cell type unique smoking responsive genes from single cell analysis. Are any of these genes overlapped with the bulk RNA-seq derived smoking response genes that were used in Fig1i?

3. For the lineage reconstruction for KRT8hi and mucous secretory cells, (a) Please show the clustering results, such as visualizing the 9 clusters in tSNE or UMAP in FigS2. The visualization of 8 clusters in FigS2a is difficult to read the clusters. (b) How were the 9 clusters in FigS2ab mapped to the four modules in Fig2a?

4. The MUC5AC and MUC5B correlation analysis.

Fig2c showed 78% of mucous secretory cells co-expressing MUC5AC and MUC5B, “consistent with the transcriptome-wide homogeneity observes”. However, Fig2e showed that there is little overlap in genes correlated with MUC5AC and MUC5B. Seems to suggest a heterogeneity within this population of cells. Are these two results contradicted to each other? Is the difference due to smoking?

5. Fig3c is insufficient to support that “MUC5AC was induced and MUC5B was suppressed by heavy smoking”. No statistics were shown to support the change in the expression levels or in the expression frequencies, the % change could due to sampling.

6. The name and analysis of “proliferating basal cells” in Fig1b, Fig3d and FigS3d/e are confusing.

In Fig1a and FigS1, authors named the cluster as “proliferating basal”, in Fig3d, there are “proliferating basal SMG cells” and “surface proliferating basal”, are those are same cluster or sub-cluster cells. Why was this original “surface” basal cell population considered in the analysis of SMG cells? If “proliferating SMG basal cells” were considered as SMG basal cells, why were these cells not involved in the lineage reconstruction analysis? In FigS3d, a subset of “proliferating SMG basal cells” are closed to “myoepithelial” cells and away from other cells. Are they in a same cluster in the sub-clustering analysis? Please show the clusters of sub-clustering analysis.

7. Doublet analysis cannot be purely based on the gene count. Several new doublet detection tools can be used to detect potential doublet. E.g., Are the SMG myoepithelial cells (epithelial/mesenchymal) doublets? Are the hybrid secretory/ciliated cells doublets?

8. The analyses in Fig3 and FigS3 are insufficient to support the stem function of the “SMG myoepithelial cells”.

(a) The “SMG myoepithelial cells” are not proliferative, no SOX9 expression, the pseudo time analysis manually specified myoepithelial cells as the starting point, please provide rationale for doing that. It is obvious that the expression of muscle related genes will reduce and basal related genes will increase when moving from myoepithelial cells to basal-like cells, as these are the markers that were used to define the two cell populations, this is not be enough to support that “SMG myoepithelial cell” is the progenitor population.

(b) For the lineage construction analysis in FigS3h, “myoepithelial cells” appears away from other SMG cells and not in the backbone of the inferred lineage. Should the green cells, located around “2.5 of component 2” at the end point of the backbone of the inferred lineage, be more appropriate to be the

start point?

(c) In addition, there is another end point of the backbone that consists of cells with dark green color in the far right of "Component 1" (SMG basal 3?). From FigS3e, dark green cells were close to "proliferating SMG basal cells". Would this point be more appropriate to be the start point?

(d) Why were the "proliferating SMG basal cells" not considered in the lineage reconstruction analysis? I think it will be helpful to include this population, it likely to be the starting point.

Minor

1. Fig2b and 3e are difficult to read. The variances of individual genes in individual cells were not shown in this type of graph presentation.

2. The network presentations are hard to read and impossible to see the relationships among the functional terms. Suggest to move these hairball networks to supplementary figures and summarize key model schema of the network in the main figure.

Reviewer #2 (Remarks to the Author):

The manuscript by Goldfarbmuren et al., describes a series of single-cell RNA seq experiments, designed to generate an atlas of tracheal epithelial cell states from smokers and non-smokers, and to ultimately define the changes in states and/or lineage trajectories driven by cigarette smoke. The authors reach several conclusions based on their data, most of which lack substantial follow-up and instead read more as intriguing hypotheses based on bioinformatic analysis of the scRNA-seq data. Overall, while the single-cell RNA-seq data presented is interesting, there are several factors that limit the impact of the work: 1) the number of samples (3 non-smokers, 2 'light' smokers, and 2 'heavy' smokers), as well as the number of cells (13,840 in total, note that it isn't clear how many cells are from each donor) is quite limited, making the claim in the abstract 'we generated a comprehensive atlas' questionable; 2) many of the observations reported (e.g., FOXN4+ population, a continuum of secretory states from club through goblet with markers of both cell types co-expressed) have been described elsewhere (see Specific Comments, below); 3) many other observations described throughout the manuscript are based solely on the scRNA-seq data and lack proper validation. The most interesting, and indeed novel aspect of this study is the overlaying of the smoking data onto the non-smoking atlas. If the authors focused more of the study on the changes induced by smoking, provide rigorous validation of the sequencing data, and identified new, potential therapeutic strategies for restoring airway homeostasis in smokers, this would undoubtedly increase the impact and broad interest of the work.

Specific comments:

1. Figure 1: While this figure lays the foundation for much of the work in the paper, there are several challenges to the experimental design: 1) the number of samples for each 'condition' (3 non-smokers, 2 'light' smokers, and 2 'heavy' smokers), as well as the number of cells in total (13,840), is quite modest; 2) the characteristics of each donor group vary broadly, particularly for the 'heavy' smokers, where one donor has 15 total pack years and the other has 90. Additionally, the heavy smokers were in their late 50s, while the light and non-smokers were in their mid-30s or younger; 3) It's unclear exactly how many cells were sequenced for each donor (if it's an equal number per donor then it would be less than 2000 cells for each). This is extremely important, given the small number of donors per group, and the fact that the authors only observe transcriptomic differences between the heavy smokers and the non-smokers. Altogether this raises questions about the conclusions that the authors make about the effects of smoking throughout the manuscript. This study would greatly

benefit from many more donors (and more cells per donor), particularly for the smoking groups.

2. The relationship between club cells and mucus-producing goblet cells has not only been described in mice (as the authors reference), but the co-expression of club cell markers (SCGB1A1) and mucins (MUC5B and MUC5AC) in surface epithelial cells has been recently described in humans (Okada et al., *AJRCCM* 199: 715-727. 2019.). In fact, the staining in Fig. 2D essentially reproduces the staining in Fig. 6 of the Okada paper, which the authors do not cite. While the transcriptional data, and the gene modules identified from the pseudotime trajectory analysis are interesting, the previously published work decreases the novelty of the 'developmental lineage of human secretory cells' from intermediate basal cells through a club cell state and into a mucus-producing goblet cell the authors report.

Functional validation of one or more of the genes present in any of the gene modules identified (e.g., determining whether any of the transcription factors identified are necessary and/or sufficient for secretory cell formation/maturation) would increase the impact of the authors' findings.

3. The transcriptional changes induced by smoking are largely not validated by in situ hybridization (e.g., RNAscope) or at the protein level by IF (apart from a single panel image in Figure 3g). The smoking data is the most novel aspect of this work, and a more thorough, careful validation of the changes observed in the scRNA-seq data would increase the impact of the work.

4. The ALI scRNA-seq timecourse data set has the potential to be quite impactful, but (similar to the primary human data) the number of cells analyzed (5,976) is really quite low. It's unclear how many cells were sequenced at each time point, but assuming an equal number per timepoint (20), that would mean about 300 cells per timepoint. Moreover, it's unclear whether all of the timepoints were from multiple donors (which would make the number of cells per timepoint per donor extremely small) or whether a subset of timepoints were from donor 1, a subset from donor 2, etc., and whether all of the cells used in the ALI experiments were from non-smokers (I'm assuming this is the case). Overall, the limited number of cells analyzed, and the lack of clarity about the samples used, limit the impact of this part of the study, which could be a valuable resource for the community.

5. Although the ALI data set is limited with respect to the number of cells analyzed, this is the part of the paper with the most validation, with the authors focusing on FOXP4 and performing a CRISPR experiment in the ALI model to test whether it is required for ciliogenesis. However, there are at least two factors that limit the impact of this finding: 1) the observation that FOXP4 is expressed during MCC differentiation was described recently (Plasschaert et al., *Nature* 560: 377-381. 2008). In fact, the staining shown in Fig. 4c is very similar to the data shown in Extended Data 4B in Plasschaert et al.; 2) the CRISPR experiment lacks sufficient information about controls, e.g., how the percent editing was determined, and whether the percent editing (indel) correlates with the effect on ciliogenesis.

6. The secretory/ciliated 'hybrid' population described by the authors has been previously observed in primary human tissue samples (Tyner et al., *JCI* 116: 309-321. 2006).

7. The authors description of the rare cell types (PNEC, tuft, and ionocytes), and in particular the markers that are expressed in each one, as well as their originating from basal cells, has been described recently (Montoro et al., *Nature* 560: 319-324. 2018; Plasschaert et al., *Nature* 560: 377-381. 2008). While the lineage relationship among the rare populations is potentially intriguing, the data supporting this speculation is limited to the RNA-seq analysis without proper validation with immunofluorescence, or testing whether depleting tuft cells (for example, by knocking out POU2F3) results in fewer ionocytes, which would support the authors statement that 'tuft-like cells may be a precursor to ionocytes, and possibly PNECs'. In addition, using the clustering of the rare populations in the tSNE plot in Figure 1 as evidence that they have 'a shared origin as well as phenotype' is not a particularly strong argument, and could simply be the result of these cells being represented by a small fraction of the data. It's worth noting that in the Plasschaert et al paper that the ionocytes clustered separately from the 'Brush + PNEC' cluster in the human ALI cultures, and each of the three rare populations clustered independently in the in vivo mouse data, although the authors cite the Plasschaert paper as having seen the cell types cluster together as they report. Finally, the previously published work by Montoro et al., and Plasschaert et al., define ionocytes as FOXI1+; V-ATPase+

based on the expression of these factors in ionocytes in other organisms/tissues. The expression of CFTR is a feature of pulmonary ionocytes, but should not be used to define whether a cell is an ionocyte or not.

Reviewer #3 (Remarks to the Author):

General comments:

In this study Goldfarbmuren and colleagues performed sc-RNAseq on tracheal epithelial cells isolated from 3 lifelong nonsmokers, 2 "light smokers", and 2 "heavy smokers". This work: 1) confirms and extends lineage specification trajectories established in mice to human tracheas; and 2) investigates how cigarette exposure alters gene expression profiles in a cell-type-specific manner. On the first point, the authors validate the existing paradigm that secretory cells arise from basal cells and provide evidence that goblet cells pass through a Club-cell-like intermediate stage and importantly they define specific transcription factors involved in this progression. The authors also identified a population of cells that express markers of mature MCCs and mucus secretory cells, and thus suggest that mucus secretory cells may transdifferentiate into MCCs rather than both arising from a shared Club-cell-like progenitor. Additionally, the authors: 1) characterized differences between submucosal gland (SMG) and superficial secretory cells, suggesting that myoepithelial cells may be an early progenitor for SMG basal cells; 2) provide evidence that rare cells such as tuft and ionocytes are phylogenetically related; and 3) newly identify FOXP4 as a MCC-specific transcription factor required for proper MCC development in vitro. Overall, the data presentation and visualization are excellent, the methods described in exacting detail, and many of the individual observations highly novel.

However, data regarding cell-type specific influence of cigarette smoke are hard to interpret due to several factors. The sample size is extremely small and not age-matched. The authors acknowledge that the 2 samples they termed "light smokers" did not differ from nonsmokers in measured parameters, which is not surprising given they had minimal smoking history and were from relatively young individuals. Many of the crucial findings of the paper such as the impact of CS on MUC5B and MUC5AC are thus generated from three young nonsmokers versus 2 older patients with an extensive smoking history. Neither PFT data or imaging data are provided to ensure these patients did not have lung disease. In-group, as opposed to between-group comparisons are minimal. These issues substantially limit interpretation of smoking related effects.

Specific comments:

1. The authors should increase the sample size of nonsmokers and smokers and use more standard descriptions (not light and heavy smokers). It would be preferable for the groups to be better age-matched. Also, it is necessary to know whether the long-term smokers had COPD. Demographic characteristics should be included in the body of the manuscript.
2. While the data supports the conclusion that Club cells serve as a progenitor population for mucus-expressing cells, care must be taken to not imply that progression through each module is inevitable "culminates... in a mucus cell." These data are not incompatible with the hypothesis that only a subset of Club cells make the final transition to mucus cells depending on external cues.
3. While the authors correctly state that smoking-induced alterations in lineage specification are associated with diseases such as COPD and asthma, they do not mention that the bulk of pathology in these diseases arise from small airways. The types and proportions of cells differ along the tracheobronchial tree, yet the authors only sampled large airways. This should be acknowledged as a limitation of the study.
4. Please clarify the source of tracheal tissue that was used to identify rare cell types (i.e. Fig. 1g).

5. Core smoking-related genes were defined as present as in >4 cell types. The authors should separately provide information on how this number varies when more stringent criteria are used (i.e. 5, 6, 7, all cell types).
6. The authors should provide a second method of pseudotime trajectory analysis to validate findings in 2b and 3e.
7. The authors should provide the EM image currently in the supplement for Figure 4d.
8. Since protein-level data is not provided, differences in CFTR expression between ionocytes and other cell types don't provide much biologically relevant information. It seems very surprising that Krt8+ transitional cells would be a major source of CFTR since these may have minimal contact with the airway lumen. This should be better explained/discussed in the manuscript.
9. PNECs are clustered at airway branch points and present sporadically throughout the epithelium. If sufficient cells are available, the authors should provide subclustering information on PNECs and determine whether different clusters correspond to these 2 groups.

Reviewer #1

The authors applied scRNA-seq to identify cell types and states of the tracheal airway epithelium in smokers and nonsmokers, infer the lineage relationship among these cells, and determine the influence of cigarette smoke on individual airway epithelial cell types. This study provided insights into cellular heterogeneity in human airway epithelium, lineage relationships, and cell type specific responses to smoking. The dataset and study could be a useful resource in the field.

Thank you for your thoughtful review. We believe our revisions, based on your suggestions and those of the other reviewers, have raised the quality of our work to a level that it would indeed be a useful resource to the field, as you suggest.

Major comments:

1. The cell names are not clearly and consistently defined. Each figure appears to have different cells not clear how they defined and how they related (Fig1b, 2a, 3d, 4a, 5a). The authors presented several clustering analyses using different subsets of cells and summarized the clustering specifications in Supp Table 5, it is difficult to read the text and figures while map clustering results back to Supp Table 5. No complete cluster information was shown in the figures; instead, clusters were replaced with cell types/states in the different figures, some of which do not have the same number types/states as the number of clusters. It is unclear how clusters were mapped to those cell types/states and any clusters were merged or removed in different analyses. Providing tSNE/UMAP visualizations of complete cell clusters and the mapping of clusters to cell types/states in their corresponding figures will greatly help understand the analysis. The authors should be very clear about the total number of major cells and associated subtypes (in one t-SNE or several associated t-SNEs in one figure) and explain the evidence supporting the definition (e.g, early vs later vs maintenance vs mature ciliated or secretory cells, basal vs basal-like, differentiating and intermediate basal).

We agree that we did not adequately communicate the identity of cell populations being analyzed and discussed throughout the various sections of the paper. We have heeded the reviewer's advice by adding tSNE/UMAP context for subsets of cells being discussed throughout, overlaying these subsetted cells onto either the global *in vivo* UMAP (see Figures 3a, 4a, 4e, 6a, and Supplementary Figures 5a, 5c, 6g, and 9cf) or the global *in vitro* tSNE plot (see Figure 5a). Furthermore, we now keep cell group names and colors sufficiently distinctive and consistent throughout the manuscript. Finally, all genes that define each cell type/subtype discussed in the paper are included in Supplementary Table 4.

2. While pseudotime lineage predictions and interpretations are interesting, the figure presentation is not easy to visualize (Fig2b, 3e). It would be nice to see the pseudotemporal ordering of the single cells along each differentiation lineage as in many of the publications.

The goal of the visualizations in those figures (now Figures 3b for the early-to-mature secretory lineage and 4f for the SMG lineage) is to show the reader which transcription factors are likely to be most critical for each functional stage of pseudotime (labeled above the plots). These figures are valuable, even to the non-computational biologist as they make testable predictions about the transcriptional regulation of the lineages. However, we also agree that the pseudotemporal ordering of cells provides valuable information, and thus these plots are now located in Supplementary Figure 5b (for the early-to-mature secretory lineage) and Supplementary Figure 6g (for the SMG lineage).

The authors should provide further evidence from the literature or their own validation to support the novel predictions (e.g., tuft-like to ionocytes).

The potential lineage relationship between the rare cell types predicted by our single cell analysis and *in vitro* differentiation data is novel and unexpected, so there is no literature reference to support this. Therefore, in the past 5 months, we have been performing gene targeting experiments in our human primary mucociliary airway epithelial cultures to investigate this putative lineage relationship. Specifically, we used CRISPR-Cas9 technology to knockout the *POU2F3* and *FOXI1* genes in basal airway epithelial cells, which control specification of tuft cells and ionocytes, respectively. These KO cells were then differentiated into mucociliary epithelia in air-liquid interface cultures. Analysis of these cultures revealed that *POU2F3* KO blocked the differentiation of all rare cell types, confirming that *POU2F3* is indeed required for tuft-like cell specification and that tuft-like cells are upstream of both ionocytes and PNECs in the rare cell lineage. In contrast, loss of *FOXI1* and ionocytes increased the prevalence of PNECs, suggesting a branched lineage where tuft-like cells can differentiate into either ionocytes or PNECs and the relative proportion of these cell types is further regulated. These results are detailed in a new section (“A novel lineage relationship among rare epithelial cell types”, Figures 6 and Supplementary Figure 10). We believe these human *in vitro* experiments strongly support the single cell analyses and make a compelling case for the rare cell lineage we propose.

Any evidence to support the transdifferentiation of mature mucous cells to mature ciliated cells? In Fig1, these two populations are not close to each other.

We would not expect mature ciliated cells to cluster with mature mucus secretory cells, since these cell types have dramatically different expression profiles. However, we would expect the small subpopulation of mucus secretory cells that we propose are becoming ciliated cells to cluster with ciliated cells, which is exactly what we see in our dataset (see the strand of ciliated cells connecting to secretory cells in Figure 1a, which correspond to the hybrid secretory/early ciliating subtype). Moreover, if this occurs as we propose (mucus secretory cell → ciliated cell) we would only expect to see this transition at its earliest stage, when these cells are starting to gain ciliated cell expression while not yet having lost mucus secretory expression. That is exactly what we see in this population: mucus secretory cells with the earliest ciliated cell expression pattern (e.g. *FOXN4*).

To provide more evidence for the mucus secretory-to-ciliated cell transition, we again turned to our *in vitro* human mucociliary epithelial cell culture system. Specifically, we stimulated mature mucociliary airway epithelial cultures with a gamma-secretase inhibitor (1 uM DAPT), which is known to induce ciliated cell formation via blocking Notch signaling¹. Using confocal imaging of these cultures, we were able to identify cells that were positive for both a marker of mature mucus secretory cells (*MUC5B*) and an early ciliating cell phenotype (*TUBG1* foci)², as was observed for the hybrid cells *in vivo* (see Figures 5f and Supplementary Figure 9d-e).

These data show that mucus secretory cells can transition into ciliated cells given the proper stimulus. Moreover, we note that only mucus secretory cells exhibited this early ciliating pattern in our *in vivo* dataset, suggesting that the mature human airway epithelium may replenish ciliated cells primarily through mucus secretory cells rather than from basal cells.

Similarly, the rationale for why they use four gene modules to represent secretory cell lineage instead of the previous defined cell types and subtypes and how these gene modules in relationship with the cell subtypes is not clearly explained.

The cell types we defined in Figure 1 are broad, and the goal of our pseudotime analysis was to characterize in more detail the major expression programs cells employ to differentiate into a mature mucus secretory cell. We are not aware of any previously defined cell types or subtypes that capture this process at baseline in humans. As detailed in the section of our paper entitled, “Secretory cells form a continuous lineage that culminates in mucus secretory cells,” we predict that differentiation of mucus secretory cells proceeds through many continuous intermediates, which are captured in our pseudotime trajectory. The most distinct intervals (groups of cells) in this trajectory are well-described by the co-expressed gene signatures or modules associated with these parts of pseudotime. We believe these gene modules give an accurate description of the functional steps by which mucus secretory cells develop. The modules and cells that correspond to known secretory cell subtypes (e.g., club secretory and mucus secretory) are labeled as such, to put our results in context of known secretory cell types as much as possible. Furthermore, as mentioned above, we have now added a global UMAP to the figure in question (now Figure 3a), making clear which two populations of cells were included in the pseudotime trajectory. Additionally, we elaborate extensively on our subclustering of these 2 broad populations and the pseudotime generation/analysis in Supplementary Figure 5a-c.

3. The authors stated in abstract and multiple other places in the manuscript that they generated a first comprehensive atlas of human airway epithelial cell types, states, lineages, and cell-specific responses to smoking which will be a useful resource to the field. But neither raw nor processed scRNA-seq data have yet submitted to any public repository (e.g., dbgap or GEO). It is required by the journal to make the data accessible. A public website to host and query these data will be a nice addition of the manuscript and useful tool to the field.

We agree that these data should be accessible to all. Both the raw data and processed count matrices from the original dataset are housed at GEO under accession number GSE134174 (to gain access, go to <https://www.ncbi.nlm.nih.gov/geo/query/acc.cgi?acc=GSE134174> and then use the reviewer access token, sbqdcocvhabdwd). We are currently uploading the latest *in vivo* data to this repository and we anticipate it will be available by Dec 15th. Moreover, we have included comprehensive lists of cell type markers, smoking DEGs, pseudotime associated genes, and other data associated with each figure panel in Supplementary Table 4.

Specific comments:

1. Fig1i presents one of the major results of cell type specific responses to smoking. Nevertheless, the methods used to reach the conclusions were not clearly presented.

(a) Please describe the analysis.

The main goal of this figure was to show that previously observed transcriptional effects of smoking from bulk data were generally also reflected in our single cell data. As such, we wouldn't really consider this a “major” portion of the paper. Furthermore, in our revised manuscript, we have expanded our analysis of single cell smoking effects to such an extent (per reviewer requests) that we have now moved this figure to the supplement (Supplementary Figure 3a).

As for how we obtained this figure, we simply plotted mean expression of smoking response genes from the bulk study across each of our main single cell populations. We provide the details asked for below and have added these details to the legend of Supplementary Figure 3a.

What's the gene list?

120 genes expressed in our dataset were present in a list of 130 genes that Beane et al.³ found to be upregulated in current smokers compared to never-smokers using bulk RNA-seq from bronchial brushings (top plot in Supplementary Figure 3a). For the plot of downregulated smoking genes from the bulk study (bottom plot in Supplementary Figure 3a), we used 48 genes expressed in our dataset among the 55 total in Beane et al. that were downregulated by smoking in bulk. This is now clearly indicated in the legend of Supplementary Figure 3a.

How many of the genes from the bulk RNA-seq analysis were able to be mapped to the single cell data and used in the analysis?

120 (of 130) for upregulated genes; 48 (of 55) for downregulated genes. See above and Supplementary Figure 3a legend.

Any of those genes differentially expressed between nonsmokers and heavy smokers in any cell types?

For the upregulated genes, 53 (of 120) were upregulated by smoking in our dataset in at least 1 population (including rare cells). For the downregulated genes, 9 (of 48) were downregulated by smoking in our dataset in at least 1 population (see Supplementary Figure 3a legend).

(b) What does the "mean normalized expression" mean?

For our current dataset, we normalized our single cell expression data using scTransform (see the third paragraph in the "Pre-processing of in vivo scRNA-seq data" section of the Methods). For each cell, we simply calculated the mean of this normalized expression across the 120 (for up) or 48 (for down) genes. Previously, we used arithmetic mean, but in the new plot this has been changed to the geometric mean in order to lend outliers less influence (see also the response below to the comment about arithmetic means being affected by outliers).

The variances of individual genes in individual cells were not shown.

As stated above, the goal for this figure was simply to show that we broadly observe smoking effects reported in previous bulk studies. Assessing the variability around each bulk RNA-seq gene individually was beyond the scope of what we wanted to do and moreover, would be redundant with the full, agnostic single gene differential expression analyses we have subsequently performed. Raw data will be available for readers (housed at GEO under accession number GSE134174) who would like to inspect variance around any particular genes.

"Mean" operations can be greatly affected by outliers. Fisher-exact test can be used to check whether the smoking genes from bulk RNA-seq are enriched in specific airway cells.

In order to minimize the influence of outliers, we now use geometric mean rather than arithmetic mean to summarize across expression of genes here and throughout the paper (see the second paragraph

under the “Plotting expression across cells” section of the Methods). Up and downregulated smoking response genes in all main cell populations were significantly enriched for the up and downregulated bulk smoking response genes, respectively (Fisher exact test FDR < 0.05), except in the case of upregulated genes for the SMG secretory population. This information has been added to the legend of Supplementary Figure 3a.

2. Fig1j showed the common and cell type unique smoking responsive genes from single cell analysis. Are any of these genes overlapped with the bulk RNA-seq derived smoking response genes that were used in Fig1i?

Yes, 11 bulk smoking DEGs overlapped with “core” smoking genes in Figure 1i (now in Figure 2a) and 26 bulk smoking DEGs overlapped with unique smoking genes.

3. For the lineage reconstruction for KRT8hi and mucous secretory cells, (a) Please show the clustering results, such as visualizing the 9 clusters in tSNE or UMAP in FigS2. The visualization of 8 clusters in FigS2a is difficult to read the clusters. (b) How were the 9 clusters in FigS2ab mapped to the four modules in Fig2a?

A UMAP of these subclusters are now shown in Supplementary Figure 5a. Also, we have now enlarged the trajectory and used more distinctive colors such that the lineage plot is easier to read (see Supplementary Figure 5b). Finally, we have added an inset trajectory plot with a pseudotime overlay (see Supplementary Figure 5b) such that the progression of these substates across pseudotime can be visualized. The modules in the pseudotime heat map (previous Fig2a, now Figure 3a) were determined by the blocks of co-expressed genes, and their borders are now indicated in Supplementary Figure 5b for additional clarity.

4. The MUC5AC and MUC5B correlation analysis.

Fig2c showed 78% of mucous secretory cells co-expressing MUC5AC and MUC5B, “consistent with the transcriptome-wide homogeneity observes”. However, Fig2e showed that there is little overlap in genes correlated with MUC5AC and MUC5B. Seems to suggest a heterogeneity within this population of cells. Are these two results contradicted to each other?

Most mature mucus secretory cells express both *MUC5AC* and *MUC5B* as we describe in the paper. However, that does not mean the expression levels of *MUC5AC* and *MUC5B*-correlated genes need be significantly correlated with each other *within* mucus secretory cells. The point of the correlation analysis is to show that *MUC5AC* and *MUC5B*, while co-expressed in mucus secretory cells, are associated with remarkably separate transcriptional programs. These results are fitting with much prior data suggesting that different stimuli are responsible for induction of these two mucins.

Is the difference due to smoking?

We controlled for smoking habit when identifying the genes co-expressed with *MUC5AC* and *MUC5B*, specifically to avoid the detection of co-expression patterns solely driven by smoking (see the “Calculating correlations with *MUC5AC* and *MUC5B*” section in the Methods). We do however find that the *MUC5B* and *MUC5AC* co-expression programs are modulated (in opposite directions) by smoking (see Figure 3f).

5. Fig3c is insufficient to support that “MUC5AC was induced and MUC5B was suppressed by heavy

smoking". No statistics were shown to support the change in the expression levels or in the expression frequencies, the % change could be due to sampling.

We agree. We have now added to this plot p-values from tests of mean differences between smokers and never smokers (see Supplementary Figure 6c). Furthermore, we clarify the statement, now stating that "the proportion of *MUC5AC*-only cells increases," rather than "*MUC5AC* was induced," more precisely fitting the data. We found that *MUC5B* expression was significantly decreased in heavy smokers compared to never smokers (FDR = 0.0095) but that, while the proportion of *MUC5AC*-only cells increased with smoking, average expression of the gene was not significantly modified.

6. The name and analysis of "proliferating basal cells" in Fig1b, Fig3d and FigS3d/e are confusing. In Fig1a and FigS1, authors named the cluster as "proliferating basal", in Fig3d, there are "proliferating basal SMG cells" and "surface proliferating basal", are those the same cluster or sub-cluster cells. Why was this original "surface" basal cell population considered in the analysis of SMG cells? If "proliferating SMG basal cells" were considered as SMG basal cells, why were these cells not involved in the lineage reconstruction analysis? In FigS3d, a subset of "proliferating SMG basal cells" are closed to "myoepithelial" cells and away from other cells. Are they in the same cluster in the sub-clustering analysis? Please show the clusters of sub-clustering analysis.

In the original dataset, we identified a small subpopulation of cells within the broad "proliferating basal" population that exhibited SMG character. Thus, we described this as a "proliferating basal SMG cell" population. We agree with the reviewer that the origin of this population was unclear in our original write-up. As we were unable to identify an analogous subgroup when analyzing our new expanded dataset, describing this population is no longer an issue. But we took the broader point that we needed to more clearly define the various subgroups we discuss in the paper. See our response under Major Comment #1 above for how we have addressed this. Also note that we now describe four main basal cell populations: proliferating, proteasomal, differentiating, and SMG basal populations (see Figure 1a).

7. Doublet analysis cannot be purely based on the gene count. Several new doublet detection tools can be used to detect potential doublets. E.g., Are the SMG myoepithelial cells (epithelial/mesenchymal) doublets? Are the hybrid secretory/ciliated cells doublets?

We argue that calculating outlier gene counts is one plausible method for detecting doublets, one that is used in the literature (e.g., ⁴) and performs reasonably well when compared to other more complex *in silico* doublet prediction methods⁵. While more elaborate methods to detect doublet cells do tend to be more sensitive, we believe that they may be inappropriate when transdifferentiating cells are present, as in our dataset, as these methods tend to work by categorizing cells containing transcripts from distinct cell types as doublets.

In the case of the SMG myoepithelial and hybrid secretory/ciliated cells, there are several lines of evidence that lead us to believe these are real populations and not doublet artifacts. First, from the recent literature, we know that both SMG myoepithelial^{6,7} and hybrid secretory/ciliated cells⁸ exist in the mouse and human airways, respectively, and thus are two populations we can reasonably expect to detect by single cell sequencing, albeit in small quantities. That we do detect populations that fit these profiles would seem to be more parsimoniously explained by their actual correspondence to these known cell types, than to the chance that the only doublet artifact populations we observe just happened to correspond to these two rare, but expected cell types, rather than to myriad other potential hybrid cell populations that we would not expect (e.g., epithelial basal-macrophages or

multiciliated-endothelial cells) and indeed do not detect. This is particularly true given that the putative “parent” populations of these two small cell groups do not include the most common populations in our dataset (e.g., basal and *KRT8*^{high} cells), which are the populations that would be most readily available to form doublets.

Secondly, while doublet cells should, if anything, contain a larger number of UMIs than either of the putative parent populations, we largely found the opposite pattern. The SMG myoepithelial cells, which would seem to be doublets composed of SMG basal and mesenchymal cell populations, exhibited a smaller median library size than either of those two potential parent populations (parent-to-child ratios of 1.42 and 1.41 for SMG basal and mesenchymal populations, respectively). Hybrid secretory/ciliated cells also exhibited a smaller median library size than mucus secretory cells (parent-to-child ratio = 1.39) or SMG secretory cells (parent-to-child ratio = 1.49), although not compared to ciliated cells (parent-to-child ratio = 0.92).

Third, our own experimental and histological data support the existence of these two populations. For the hybrid secretory/ciliated cells, we show that this phenotype can indeed be experimentally induced in culture (see Figure 5f and Supplementary Figure 9bc). And with IF labeling, we observed co-localization of ACTA2+ and KRT5+ cells (see Figure 4g), supporting the existence of cells with this myoepithelial phenotype in human airway epithelia *in vivo*.

Finally, these two cell populations contain unique gene expression signatures that are not well explained by those of either sets of putative parent populations. To systematically look at this, we used the program DoubletCluster implemented using the scran package⁹, which aims to identify doublet-based populations (rather than doublet cells) by assaying the number of genes with expression patterns that go against the hypothesis of a doublet origin. Within a cluster of doublet cells, most genes should exhibit intermediate expression relative to the two potential parent populations. Thus, the method tests for genes that are consistently up- or downregulated in a given query (i.e., putative doublet) cluster relative to *both* potential source (i.e., parent) populations, which would suggest that the query population is not simply a composite of the two sources, but exhibits a genuinely unique signature. For the myoepithelial cells, there were 669 such genes, when SMG basal and mesenchymal cell populations were assumed to be the source, whereas for hybrid secretory/ciliated cells, there were 2,169 such genes when compared to mature ciliated and mucus secretory populations. These results constitute strong evidence that these populations exhibit a unique character that is not well explained by combining other populations in the dataset.

8. The analyses in Fig3 and FigS3 are insufficient to support the stem function of the “SMG myoepithelial cells”.

(a) The “SMG myoepithelial cells” are not proliferative, no SOX9 expression, the pseudo time analysis manually specified myoepithelial cells as the starting point, please provide rationale for doing that. It is obvious that the expression of muscle related genes will reduce and basal related genes will increase when moving from myoepithelial cells to basal-like cells, as these are the markers that were used to define the two cell populations, this is not be enough to support that “SMG myoepithelial cell” is the progenitor population.

We agree with the reviewer that not much is learned by including the myoepithelial cells in the SMG pseudotime trajectory, as there are no intermediate cells by which to connect them to other SMG basal cells. Thus, we have removed myoepithelial cells from the pseudotime analysis (see new Figures 4f and Supplementary Figure 6g).

We also agree that our scRNA-seq results don't provide evidence for the stem function of this population. That said, stem function of these cells has previously been shown in mice^{6,7}, and thus in our view, this constitutes the null hypothesis for their function in humans. We have modified the discussion to make clear that the stem nature of this population remains a hypothesis in need of further testing.

(b) For the lineage construction analysis in FigS3h, "myoepithelial cells" appears away from other SMG cells and not in the backbone of the inferred lineage. Should the green cells, located around "2.5 of component 2" at the end point of the backbone of the inferred lineage, be more appropriate to be the start point?

We agree. See comment above.

(c) In addition, there is another end point of the backbone that consists of cells with dark green color in the far right of "Component 1" (SMG basal 3?). From FigS3e, dark green cells were close to "proliferating SMG basal cells". Would this point be more appropriate to be the start point?

See above.

(d) Why were the "proliferating SMG basal cells" not considered in the lineage reconstruction analysis? I think it will be helpful to include this population, it likely to be the starting point.

In the new dataset, we no longer have a proliferating SMG basal cell population. We now include all SMG populations in the trajectory, excluding myoepithelial cells, per reasons discussed above.

Minor

1. Fig2b and 3e are difficult to read. The variances of individual genes in individual cells were not shown in this type of graph presentation.

We respectfully disagree. Our goal with these plots is not to show the nuances of cell-by-cell expression of these genes, but rather to show 1) which transcription factors are likely important during the process of differentiation and 2) their relative order of peak expression along this differentiation trajectory. These figures are valuable, even to the non-computational biologist as they make testable predictions about the transcriptional regulation of the lineages. We believe the best way to show these two pieces of information is to plot these trajectories on the same plot and scaled in the same way.

2. The network presentations are hard to read and impossible to see the relationships among the functional terms. Suggest to move these hairball networks to supplementary figures and summarize key model schema of the network in the main figure.

While we agree that the network connections themselves are difficult to visualize in these graphs, we believe that these plots contain a lot of novel information that *can* be gleaned, such as which functional gene groups are modified by smoking across the epithelium as a whole, how these functional responses differ based on the nature or number of cell types involved and the strength of response, and which genes act as functional hubs. Therefore, although these plots can be moved to the supplement if necessary, we would prefer to keep them in the main figure, as we think that they provide a nice synthetic summary of the smoking response. Figure 8 also includes a key model schematic to help summarize the dense information in the networks.

Reviewer #2 (Remarks to the Author):

The manuscript by Goldfarbmuren et al., describes a series of single-cell RNA seq experiments, designed to generate an atlas of tracheal epithelial cell states from smokers and non-smokers, and to ultimately define the changes in states and/or lineage trajectories driven by cigarette smoke. The authors reach several conclusions based on their data, most of which lack substantial follow-up and instead read more as intriguing hypotheses based on bioinformatic analysis of the scRNA-seq data.

Overall, while the single-cell RNA-seq data presented is interesting, there are several factors that limit the impact of the work: 1) the number of samples (3 non-smokers, 2 'light' smokers, and 2 'heavy' smokers), as well as the number of cells (13,840 in total, note that it isn't clear how many cells are from each donor) is quite limited, making the claim in the abstract 'we generated a comprehensive atlas' questionable; 2) many of the observations reported (e.g., FOXN4+ population, a continuum of secretory states from club through goblet with markers of both cell types co-expressed) have been described elsewhere (see Specific Comments, below); 3) many other observations described throughout the manuscript are based solely on the scRNA-seq data and lack proper validation.

The most interesting, and indeed novel aspect of this study is the overlaying of the smoking data onto the non-smoking atlas. If the authors focused more of the study on the changes induced by smoking, provide rigorous validation of the sequencing data, and identified new, potential therapeutic strategies for restoring airway homeostasis in smokers, this would undoubtedly increase the impact and broad interest of the work.

We thank the reviewer for their critical assessment of our work. We have spent the past 5 months addressing these insightful critiques.

Specifically, we have:

- (1) Increased dramatically the number of donors and cells involved in our analyses;
- (2) Addressed concerns regarding the novelty of our work below, and also performed additional experiments to increase novelty;
- (3) Validated several of the key, most controversial, and novel aspects of the paper;
- (4) Strengthened our smoking analyses by the significant increase in both donors and cells, new analyses, and follow-up validation by immunofluorescence labeling of donor tissue.

See our specific responses to comments below.

Specific comments:

1. Figure 1: While this figure lays the foundation for much of the work in the paper, there are several challenges to the experimental design: 1) the number of samples for each 'condition' (3 non-smokers, 2 'light' smokers, and 2 'heavy' smokers), as well as the number of cells in total (13,840), is quite modest; 2) the characteristics of each donor group vary broadly, particularly for the 'heavy' smokers, where one donor has 15 total pack years and the other has 90. Additionally, the heavy smokers were in their late 50s, while the light and non-smokers were in their mid-30s or younger; 3) It's unclear exactly how many cells were sequenced for each donor (if it's an equal number per donor then it would be less than 2000 cells for each). This is extremely important, given the small number of donors per group, and the fact that the authors only observe transcriptomic differences between the heavy smokers and the non-smokers. Altogether this raises questions about the conclusions that the authors make about the effects of smoking throughout the manuscript. This study would greatly benefit from many more donors (and more cells per donor), particularly for the smoking groups.

As discussed above, we agreed that the paper would be greatly strengthened by inclusion of more donors, more sequenced cells, and closer age-matching of the comparison groups. In the revised manuscript we have addressed all of these critiques by (1) nearly tripling the number of examined cells to 36,248; (2) over doubling both the overall number of donors (n=15) as well as the donors used in the smoking comparison groups (6 vs. 6); and (3) carefully selecting new donors such that the age distribution between smokers and non-smokers is now much more closely matched (57 ± 3.02 , 50 ± 7.47 , respectively, no significant difference, $p = 0.4$). The number of cells, donors, and donors in comparison groups are now in line with or higher than state-of-the-art single cell characterization papers currently being published in *Nature* journals (e.g., Plasschaert *et al*, 2018¹⁰, Braga *et al*, 2019⁸).

2. The relationship between club cells and mucus-producing goblet cells has not only been described in mice (as the authors reference),

To be clear regarding what has previously been described:

- (1) The mouse airway lacks substantial goblet cells outside of IL-13 stimulation¹¹.
- (2) In mice, IL-13 stimulation clearly induces club secretory cells to differentiate into goblet cells¹².
- (3) In the human airway, many mucus secretory “goblet” cells exist in the absence of IL-13 or other stimulus unlike in laboratory mice^{13,14}.

Therefore, the mouse airway state does not parallel the human airway state with regard to mucus secretory cells. Thus, our data supporting human club secretory cells differentiating into mucus secretory cells in the absence of IL-13 is novel and presents a strong hypothesis that many researchers in the field can investigate.

but the co-expression of club cell markers (SCGB1A1) and mucins (MUC5B and MUC5AC) in surface epithelial cells has been recently described in humans (Okada *et al.*, AJRCCM 199: 715-727. 2019.). In fact, the staining in Fig. 2D essentially reproduces the staining in Fig. 6 of the Okada paper, which the authors do not cite.

We now cite the Okuda paper, however our panel showing co-labeling of club secretory markers and mucus markers is simply meant to validate the scRNA-seq-based co-expression of these markers. Beyond demonstrating that co-expression of SCGB1A1, MUC5AC, and MUC5B can occur, part of the novelty in our secretory section is that we reveal the propensity of this co-expression across mature mucus secretory cells *in vivo* - something that can only be defined by assaying the expression profile of numerous single cells. This is a very significant advance over the Okuda paper.

While the transcriptional data, and the gene modules identified from the pseudotime trajectory analysis are interesting, the previously published work decreases the novelty of the ‘developmental lineage of human secretory cells’ from intermediate basal cells through a club cell state and into a mucus-producing goblet cell the authors report. Functional validation of one or more of the genes present in any of the gene modules identified (e.g., determining whether any of the transcription factors identified are necessary and/or sufficient for secretory cell formation/maturation) would increase the impact of the authors’ findings.

We respectfully disagree that the novelty of our pseudotime trajectory analysis is diminished by the published work mentioned for the reasons listed above and also:

1. Our work represents the first strong empirical data that human secretory cells of different types (club and mucus secretory) and different degrees of secretory expression are likely part of one continuous lineage.
2. Not only does our lineage connect these cell types and states but it defines the transcriptional progression of this lineage including transcription factors that likely drive this progression. This is a very novel area where very little progress has been made in the field. Even with regard to the most studied of mucus secretory transcription factors, SPDEF, only human *in vitro* data exists to suggest its role in IL-13-induced mucus metaplasia. SPDEF's role in the development of human mucus secretory cells outside of IL-13 stimulation is unknown and our study gives some of the first data supporting its role in this context. Moreover, we present highly novel transcription factors likely involved in human mucus secretory cell development, a notoriously slow-moving field that has thus far yielded poor understanding of this cell lineage.
3. We agree that validation of these transcription factors would be highly valuable to the field. However, we are limited in how many findings we can functionally validate in one paper, and have focused on other areas for functional validation. Our paper presents new testable hypotheses about known and novel transcription factors that the entire field can follow up on.

3. The transcriptional changes induced by smoking are largely not validated by *in situ* hybridization (e.g., RNAscope) or at the protein level by IF (apart from a single panel image in Figure 3g). The smoking data is the most novel aspect of this work, and a more thorough, careful validation of the changes observed in the scRNA-seq data would increase the impact of the work.

We wholly agree that further validation of our transcriptome-level smoking effects would be valuable. We now include validation through analysis of both histological tissue and cytopins of select smoking related genes in this revised manuscript. The images shown in Supplementary Figures 2a and 4 are representative of the intact, non-metaplastic epithelium in a single histological cross-section of human trachea from 2 never smokers and 2 heavy smokers, and display expression trends akin to our single cell analyses for *BPIFA1*, *SCGB1A1*, *HLA-DRB1*, and *MUC5AC* mRNAs and *SCGB1A1* protein.

We believe that more comprehensive IF/ISH validation is beyond the scope of this study. Our single cell transcriptomic analyses represent cells sampled from a large surface area of human trachea. As such, these samples describe cellular states independently of local variation due to anatomical position, irritation etc. In contrast, imaging histological sections of the trachea with FISH/IF is extremely dependent on position, and thus local variation within each donor renders quantification between smoking groups near impossible due to the sheer number of images required to account for this spatial heterogeneity. An alternative approach utilizes cytopins as the randomly sampled single cell suspension, but these dissociated cells lose all localization context within the tissue and thus require a robust panel of cell type labels to re-establish cell type in the sample. Further, our scRNA-seq smoking analysis harnesses differential expression rather than differential cell counts, yet the latter is much more straightforward to assess from cytopins. Thus ultimately, quantification of these highly varied sections and expression level quantification of cytopins both require method development that is outside the purview of the paper.

Therefore, we strongly believe the best way to establish confidence in our results generated from single cell RNA-seq is to make the single cell experiment more robust. We have done this through the dramatic increase in both cells and donors analyzed in this revised manuscript. This manuscript represents a tremendous amount of careful work which we feel sufficient confidence in to present to the scientific community, allowing the entire field to validate specific aspects of this work. Additionally, we have

expanded our analysis of this new, more robust dataset, enabling multiple lines of evidence to support our conclusions (see new Figure 2cd).

4. The ALI scRNA-seq timecourse data set has the potential to be quite impactful, but (similar to the primary human data) the number of cells analyzed (5,976) is really quite low. It's unclear how many cells were sequenced at each time point, but assuming an equal number per timepoint (20), that would mean about 300 cells per timepoint. Moreover, it's unclear whether all of the timepoints were from multiple donors (which would make the number of cells per timepoint per donor extremely small) or whether a subset of timepoints were from donor 1, a subset from donor 2, etc., and whether all of the cells used in the ALI experiments were from non-smokers (I'm assuming this is the case). Overall, the limited number of cells analyzed, and the lack of clarity about the samples used, limit the impact of this part of the study, which could be a valuable resource for the community.

To clarify, each single cell time point (n=20) included cells from 3 donors with the cells from those 60 donor/time points harvested from 2-3 replicate inserts. The donors included a never smoker, light smoker, and heavy smoker also included in the *in vivo* portion of the study (this information is now included in the "In vitro ALI culture (time course)" section of the methods, as well as Supplementary Table 1). The experiment captured differentiation intermediates and transcriptional transitions underlying mucociliary differentiation in both real-time and pseudotime with the number of cells analyzed. This represents a tour de force of an experiment, unlikely to be repeated by others. The value of this dataset lies in the many time points that collectively, densely cover differentiation. Moreover, we don't believe there is any objective basis by which to deem the number of cells we've analyzed as "quite low." In fact, empirically, we found this number to be sufficient to derive both ciliated and secretory cell lineages. Both the timing of cell type appearances and associated gene expression patterns aligned well with known information regarding mucociliary differentiation, strongly supporting the validity of the results generated from analysis of this dataset. Furthermore, the number of cells was adequate for us to discover a rare cell population (FOXN4+ early ciliated) acting as an early ciliating cell intermediate, a group we would expect to be most confounded by insufficient cell numbers. This population was validated in a functional experiment and has been observed in another study as well¹⁰. If we compare our dataset to a study recently published in *Nature* using 2,970 cells from a single human ALI time point¹⁰, we believe our dataset represents a significant advance that will be highly valuable to the field.

5. Although the ALI data set is limited with respect to the number of cells analyzed, this is the part of the paper with the most validation, with the authors focusing on FOXN4 and performing a CRISPR experiment in the ALI model to test whether it is required for ciliogenesis. However, there are at least two factors that limit the impact of this finding: 1) the observation that FOXN4 is expressed during MCC differentiation was described recently (Plasschaert et al., *Nature* 560: 377-381. 2008). In fact, the staining shown in Fig. 4c is very similar to the data shown in Extended Data 4B in Plasschaert et al.;

Again, our identification of the FOXN4 population and its appearance in ALI differentiation time is empirical evidence for the value of our ALI data set. Regarding the novelty/impact of our work in light of the Plasschaert et al paper:

- (1) We now cite the paper at this point in the manuscript.
- (2) Plasschaert et al noted a FOXN4+ population in human ALI day 14 data and speculated that it might be involved in early ciliogenesis due co-expression with FOXJ1 and lack of co-expression with other late ciliated markers. Their "staining" shows FOXN4 in 1 of several FOXJ1+ cells, and

not in AcTub+ cells. There are 6 cells shown compared with our wholemount images which are far more robust.

- (3) Most importantly, using both an actual time course and pseudotime analysis, we placed FOXN4 expression in the context of a specific ciliated cell developmental state and classified all genes associated with this state, providing much stronger evidence for FOXN4+ cells in early ciliated cell development. Moreover, through extremely challenging KO studies we generated FOXN4 KO epithelial cultures. FOXN4 KO resulted in blockade of ciliated cell development, trapping cells in an immature ciliated state. We also performed extensive protein localization studies to confirm that this blockade occurred at the early ciliating cell step, which our trajectory analyses and real-time ALI studies suggested is where FOXN4 is active. Our analysis also revealed that basal body machinery was still generated with the loss of FOXN4, but that basal body docking and deuterosome disassembly was affected, further pinpointing the role of FOXN4 in ciliated cell development. In total these experiments have generated a wealth of novel information regarding ciliated cell development and the function of FOXN4 in this process that goes beyond linking it to ciliogenesis.

2) the CRISPR experiment lacks sufficient information about controls, e.g., how the percent editing was determined, and whether the percent editing (indel) correlates with the effect on ciliogenesis.

We developed a dual-exonic guide (2 guides) CRISPR-RNP method for generating KOs in human airway epithelial cells (subject of a manuscript in preparation). In this system, editing can occur either at only a single site or at both sites, with the latter possibly leading to the deletion of the sequence between the two sites. Therefore, quantitation of editing efficiency at the sequence level is difficult. Rather, as is normal protocol for us and other groups, we performed high resolution melt curve analysis of PCR products covering the two cut sites¹⁵. See HRM curves below for both guides in the edited vs. control edited culture PCR products. This analysis yielded large differences between the melting curves of the cut site PCR products from the CRISPR-RNP with non-targeting guide (control) vs. FOXN4 guides, characteristic of a high degree of editing at those sites.

Beyond this, editing efficiency can, but does not always, correlate with protein loss, as many edits are silent with respect to protein expression. The goal of KO is stop protein production. Assessing this is difficult given that 1) the available FOXN4 antibody did not work for western blot, 2) FOXN4+ cells are very transient, and 3) FOXN4 cells make up a small percentage of cultures. Despite this, wholemount IF labeling of day 9 cultures, like Figure 5c shown in the paper, rarely exhibited a FOXN4+ cell in the KO's (we estimate 80-90% reduction compared to control cultures) and when present in the KO, FOXN4 was expressed at a lower intensity (see figure below). This loss of FOXN4 protein strongly correlated with

blockade of ciliated cell differentiation, trapping cells in an immature ciliated state, consistent with its expression in the ciliated cell lineage as determined by single cell sequencing.

6. The secretory/ciliated 'hybrid' population described by the authors has been previously observed in primary human tissue samples (Tyner et al., JCI 116: 309-321. 2006).

[Redacted]

To clarify, the secretory/ciliated cell population described in Tyner et al. contained cilia (stained positive for beta-tubulin marking the axonemes of cilia) and contained MUC5AC mucus granules. Therefore, this population appears to have characteristics of both mature ciliated and mature mucus secretory cells. In experiments conducted throughout the Tyner paper, they determined that this population forms from mature ciliated cells that gain mucus secretory properties through IL-13 stimulation. This is altogether different from the hybrid population we describe. We describe a population with mature mucus secretory

characteristics that also have early ciliating cell differentiation gene expression. The implication here is that mature mucus secretory cells are transdifferentiating into ciliated cells. Moreover, we find that this is the only population in the *in vivo* human single cell data containing the FOXN4+ early ciliating cell expression pattern, suggesting that transdifferentiation of mucus secretory cells may be a major source of new ciliated cells in adult humans. As such, this work is highly novel and entirely distinct from the Tyner work.

7. The authors description of the rare cell types (PNEC, tuft, and ionocytes), and in particular the markers that are expressed in each one, as well as their originating from basal cells, has been described recently (Montoro et al., Nature 560: 319-324. 2018; Plasschaert et al., Nature 560: 377-381. 2008).

To clarify, the Montoro paper describes these cells in mice not humans⁴. The Plasschaert paper describes these cells in human airway epithelial cell cultures¹⁰. We describe these cells *in vivo* from humans, as well as in a detailed time course of human epithelial cell cultures.

While the lineage relationship among the rare populations is potentially intriguing, the data supporting this speculation is limited to the RNA-seq analysis without proper validation with immunofluorescence, or testing whether depleting tuft cells (for example, by knocking out POU2F3) results in fewer ionocytes, which would support the authors statement that 'tuft-like cells may be a precursor to ionocytes, and possibly PNECs'. In addition, using the clustering of the rare populations in the tSNE plot in Figure 1 as evidence that they have 'a shared origin as well as phenotype' is not a particularly strong argument, and could simply be the result of these cells being represented by a small fraction of the data. It's worth noting that in the Plasschaert et al paper that the ionocytes clustered separately from the 'Brush + PNEC' cluster in the human ALI cultures, and each of the three rare populations clustered independently in the *in vivo* mouse data, although the authors cite the Plasschaert paper as having seen the cell types cluster together as they report.

We agree that alone, the clustering of rare cell populations together is not sufficient evidence to prove a lineage relationship between rare cells. In addition to the overlapping differential expression analyses and co-labeling of cells with *POU2F3* and *FOXI1* mRNAs (Figure 6), we have now generated and analyzed

both *POU2F3* and *FOXI1* KO human ALI cultures. As described in the text (related to Figures 6 and 7) these experiments strongly support a novel rare cell lineage where tuft-like cells are progenitors of both ionocytes and PNECs, in a branched lineage.

Finally, the previously published work by Montoro et al., and Plasschaert et al., define ionocytes as FOXI1+; V-ATPase+ based on the expression of these factors in ionocytes in other organisms/tissues. The expression of CFTR is a feature of pulmonary ionocytes, but should not be used to define whether a cell is an ionocyte or not.

We agree. We do not define ionocytes by CFTR expression, but rather by their full expression pattern in the single cell data or (in IF labeling) by FOXI1 nuclear localization. In fact, a main point in our original manuscript version and definitely in the revised manuscript is that other cells in the epithelium provide a large majority of epithelial CFTR expression. We now conduct Ussing chamber analysis of the KO cultures in the revised manuscript, providing a wealth of novel information regarding ionocyte regulation of transepithelial conductance and ion channel activity. Finally, to address the implicit reference to Figure 1c, we have added voltage-gated chloride channel CLCNKB as an additional marker of ionocytes.

Reviewer #3 (Remarks to the Author):

General comments:

In this study Goldfarbmuren and colleagues performed sc-RNAseq on tracheal epithelial cells isolated from 3 lifelong nonsmokers, 2 "light smokers", and 2 "heavy smokers". This work: 1) confirms and extends lineage specification trajectories established in mice to human tracheas; and 2) investigates how cigarette exposure alters gene expression profiles in a cell-type-specific manner. On the first point, the authors validate the existing paradigm that secretory cells arise from basal cells and provide evidence that goblet cells pass through a Club-cell-like intermediate stage and importantly they define specific transcription factors involved in this progression. The authors also identified a population of cells that express markers of mature MCCs and mucus secretory cells, and thus suggest that mucus secretory cells may transdifferentiate into MCCs rather than both arising from a shared Club-cell-like progenitor. Additionally, the authors: 1) characterized differences between submucosal gland (SMG) and superficial secretory cells, suggesting that myoepithelial cells may be an early progenitor for SMG basal cells; 2) provide evidence that rare cells such as tuft and ionocytes are phylogenetically related; and 3) newly identify FOXN4 as a MCC-specific transcription factor required for proper MCC development in vitro. Overall, the data presentation and visualization are excellent, the methods described in exacting detail, and many of the individual observations highly novel.

We greatly appreciate your positive perception of our work. Thank you.

However, data regarding cell-type specific influence of cigarette smoke are hard to interpret due to several factors. The sample size is extremely small and not aged matched. The authors acknowledge that the 2 samples they termed "light smokers" did not differ from nonsmokers in measured parameters, which is not surprising given they had minimal smoking history and were from relatively young individuals. Many of the crucial findings of the paper such as the impact of CS on MUC5B and MUC5AC are thus generated from three young nonsmokers versus 2 older patients with an extensive smoking history. Neither PFT data or imaging data are provided to ensure these patients did not have lung

disease. In-group, as opposed to between-group comparisons are minimal. These issues substantially limit interpretation of smoking related effects.

As discussed above, we agreed that the paper would be greatly strengthened by inclusion of more donors, more sequenced cells, and closer age-matching of the comparison groups. In the revised manuscript, we have addressed all of these critiques by (1) nearly tripling the number of examined cells to 36,248, (2) doubling both the overall number of donors (n=15) as well as the donors in the smoking comparison groups (6 vs. 6), and (3) carefully selecting new donors such that the age distribution between smokers and never smokers is now much more closely matched (57 ± 3.02 , 50 ± 7.47 , respectively, no significant difference, $p = 0.4$). We also supply all the additional metadata that is available to us for these donors in Supplementary Table 1. To address in-group comparisons, we report the variation among individual donors within each smoking habit where applicable (e.g., Supplementary Figures 1ab, 3c, and 5f).

Specific comments:

1. The authors should increase the sample size of nonsmokers and smokers and use more standard descriptions (not light and heavy smokers). It would be preferable for the groups to be better age-matched. Also, it is necessary to know whether the long-term smokers had COPD. Demographic characteristics should be included in the body of the manuscript.

We have increased the sample sizes and implemented age-matching as described above. Our thinking was that “light” and “heavy” well represented the two discrete groups of smokers in our dataset (≤ 3 versus ≥ 15 pack years), which we now explicitly define in the first paragraph of the Results. Nevertheless, as we focus only on “heavy smokers” and “never smokers” in analysis of smoking effects, throughout the paper we now simply refer to our “treatment groups” as “smokers” and “nonsmokers”. We have included a table (Supplementary Table 1) with all demographic and clinical data available on the donors. No donors used for this analysis reported a history of COPD.

2. While the data supports the conclusion that Club cells serve as a progenitor population for mucus-expressing cells, care must be taken to not imply that progression through each module is inevitable “culminates... in a mucus cell.” These data are not incompatible with the hypothesis that only a subset of Club cells make the final transition to mucus cells depending on external cues.

We completely agree with this interpretation. We have modified the text in this section of the results to read “...these data support a single developmental lineage of human secretory cells, driven by sequentially activated TFs, which transitions through functional intermediates (club cells) to culminate in a multi-functional mucus secretory cell, although all cells in this lineage need not reach this mucus secretory endpoint, depending on internal or external differentiation cues.”

3. While the authors correctly state that smoking-induced alterations in lineage specification are associated with diseases such as COPD and asthma, they do not mention that the bulk of pathology in these disease arise from small airways. The types and proportions of cells differ along the tracheobronchial tree, yet the authors only sampled large airways. This should be acknowledged as a limitation of the study.

Again, we agree. We have added the following statement to the Discussion: “We note these effects are based on sampling the tracheal airway, and future work will be necessary to extend our single cell

understanding of smoking effects to the small airways, which exhibit extensive pathology in smoking-related lung disease.”

4. Please clarify the source of tracheal tissue that was used to identify rare cell types (i.e. Fig. 1g).

The tracheal tissue used to identify the rare cell types by histological labeling was the same as used for the common cell types. We looked at tissue from both heavy and never smokers, and some tissue was from a subset of the same donors used for single cell sequencing. The images in Figure 1 are representative of labeling patterns observed in both heavy and never smokers.

5. Core smoking-related genes were defined as present as in >4 cell types. The authors should separately provide information on how this number varies when more stringent criteria are used (i.e. 5, 6, 7, all cell types).

We provide all smoking response genes for all cell-types in the paper in Supplementary Table 4. With this information, a reader can determine any possible set of overlapping genes. But to answer your question, please see the table below in which the number of upregulated and downregulated core DEGs are given for all numbers of main clusters that may share a given DEG.

No. clusters	N Up core DEGs	N Down core DEGs
8	12	3
7	24	8
6	47	10
5	86	29
4	128	40
3	200	58
2	335	99
1	803	227

6. The authors should provide a second method of pseudotime trajectory analysis to validate findings in 2b and 3e.

Uniform Manifold Approximation and Projection (UMAP)¹⁶ is a relatively new machine learning method for dimensionality reduction that has been shown, compared to tSNE, to better preserve global structure of scRNA-seq data, thus better capturing lineage relationships among cell types¹⁷. Thus, for the two Monocle-derived trajectories indicated by the reviewer (now depicted in Supplementary Figures 5b and 6g), we have now included UMAP plots (Supplementary Figure 5a and Figure 4e) that overlay the same cell subclusters as are overlaid onto the Monocle trajectory plots, and demonstrate remarkable concordance in the two-dimensional ordering of these cells between the two methods. Thus, although UMAP isn't an explicit pseudotime trajectory method, we argue that this concordance represents a strong technical validation of our reconstructed trajectories.

7. The authors should provide the EM image currently in the supplement for Figure 4d.

The image referred to in the supplement (now Supplementary Figure 9b) depicts confocal microscopy of acetylated alpha-tubulin immunofluorescence labeling as stated in the legend, and is now clarified. Since images like this representative example were used to produce the summary bar graph in the main figure (now Figure 5d), we have kept the confocal IF image in the supplement due to space limitations.

8. Since protein-level data is not provided, differences in CFTR expression between ionocytes and other cell types don't provide much biologically relevant information. It seems very surprising that Krt8+ transitional cells would be a major source of CFTR since these may have minimal contact with the airway lumen. This should be better explained/discussed in the manuscript.

Our revision includes protein-level IF labeling of CFTR (Figure 7d, Supplementary Figure 11c) demonstrating apical hotspots in FOXI1⁺ cells (ionocytes) and additional, mostly apical, signal in FOXI1⁻ cells (non-ionocytes), that is consistent with our scRNA-seq expression data. Further, as we state in the text and demonstrate in Figure 1f, our *KRT8*^{high} population does contact the airway lumen often and thus may contain apical CFTR. Finally, our revised manuscript includes Ussing chamber analysis that demonstrates that CFTR activity is robust in ionocyte-depleted human epithelial cultures, indicating that this non-ionocyte CFTR signal at both the mRNA and protein level does in fact represent active channels.

9. PNECs are clustered at airway branch points and present sporadically throughout the epithelium. If sufficient cells are available, the authors should provide subclustering information on PNECs and determine whether different clusters correspond to these 2 groups.

This is a good idea. We did try to subcluster the 83 PNEC cells in our *in vivo* dataset, but were unable to identify apparently meaningful subtypes that might correspond to clustering in tissue. This is definitely something we will be interested in pursuing but will likely require many more PNECs to perform this subclustering in a powerful way.

Literature cited

- 1 Stubbs, J. L., Vladar, E. K., Axelrod, J. D. & Kintner, C. Multicilin promotes centriole assembly and ciliogenesis during multiciliate cell differentiation. *Nat Cell Biol* **14**, 140-147, doi:10.1038/ncb2406 (2012).
- 2 Vladar, E. K. & Stearns, T. Molecular characterization of centriole assembly in ciliated epithelial cells. *J Cell Biol* **178**, 31-42, doi:10.1083/jcb.200703064 (2007).
- 3 Beane, J. *et al.* Reversible and permanent effects of tobacco smoke exposure on airway epithelial gene expression. *Genome Biol* **8** (2007).
- 4 Montoro, D. T. *et al.* A revised airway epithelial hierarchy includes CFTR-expressing ionocytes. *Nature* **560**, 319-324 (2018).
- 5 Bais, A. S. & Kostka, D. scds: computational annotation of doublets in single-cell RNA sequencing data. *Bioinformatics* **2019**, 1-9, doi:10.1093/bioinformatics/btz698 (2019).
- 6 Lynch, T. J. *et al.* Submucosal gland myoepithelial cells are reserve stem cells that can regenerate mouse tracheal epithelium. *Cell Stem Cell* **22**, 653-667 (2018).
- 7 Tata, A. *et al.* Myoepithelial cells of submucosal glands can function as reserve stem cells to regenerate airways after injury. *Cell Stem Cell* **22**, 668-683 (2018).
- 8 Braga, F. A. V. *et al.* A cellular census of human lungs identifies novel cell states in health and in asthma. *Nature Medicine* **25**, 1153-1163 (2019).

- 9 Lun, L. T. L., McCarthy, D. J. & Marioni, J. C. A step-by-step workflow for low-level analysis of single-cell RNA-seq data with Bioconductor. *F1000Research* **5**, 2122, doi:10.12688/f1000research.9501.2 (2016).
- 10 Plasschaert, L. W. *et al.* A single-cell atlas of the airway epithelium reveals the CFTR-rich pulmonary ionocyte. *Nature* **560**, 377-381 (2018).
- 11 Evans, C. M. *et al.* Mucin is produced by Clara cells in the proximal airways of antigen-challenged mice. *Am J Respir Cell Mol Biol* **31**, 382-394, doi:10.1165/rcmb.2004-0060OC (2004).
- 12 Chen, G. *et al.* SPDEF is required for mouse pulmonary goblet cell differentiation and regulates a network of genes associated with mucus production. *Journal of Clinical Investigation* **119**, 2914-2924, doi:10.1172/jci39731 (2009).
- 13 Okuda, K. *et al.* Localization of secretory mucins MUC5AC and MUC5B in normal/healthy human airways. *American Journal of Respiratory and Critical Care Medicine* **199**, 715-727, doi:10.1164/rccm.201804-0734OC (2019).
- 14 Seibold, M. A. *et al.* The idiopathic pulmonary fibrosis honeycomb cyst contains a mucociliary pseudostratified epithelium. *PLoS ONE* **8**, doi:10.1371/journal.pone.0058658 (2013).
- 15 Everman, J. E., Rios, C. & Seibold, M. A. in *Disease Gene Identification: Methods and Protocols, Methods In Molecular Biology* Vol. 1706 (ed Johanna K. DiStefano) Ch. 15, 267-292 (Springer, 2018).
- 16 McInnes, L., Healy, J. & Melville, J. UMAP: Uniform Manifold Approximation and Projection for Dimension Reduction. *arXiv:1802.03426* (2018).
- 17 Becht, E. *et al.* Dimensionality reduction for visualizing single-cell data using UMAP. *Nature Biotechnology* **37**, 38-44 (2019).

REVIEWERS' COMMENTS:

Reviewer #1 (Remarks to the Author):

In their revised manuscript, authors have adequately addressed most of my concerns raised with the original submission. They added CRISPR-Cas9 knockout studies to validate the single cell prediction that tuft-like cells are the likely progenitor of both pulmonary neuroendocrine cells and CFTR-rich ionocytes. They have improved the cell classification and presentation by including the global umap and keep cell group names and colors consistent throughout the manuscript. Overall, the manuscript is much improved. Several remaining questions are listed below:

1. In multiple places, authors assessed smoking effects on cell populations via cell proportional changes (Fig2, Fig3, FigS5,). We know that estimating compositional changes in single-cell data requires sufficient cell numbers to robustly assess proportions changes, and sufficient sample numbers to evaluate expected background variation. The estimation of rare cell type proportion is even more difficult (Fig7), usually requires specific statistical models. While these results can be informative of changes in compositions between non-smokers and heavy smokers, however, should be interpreted with caution.
2. Line 243-244: "In all, these data support a single developmental lineage of human secretory cells, driven by sequentially activated TFs, which transitions through functional intermediates (club cells) to culminate in a multi-functional mucus secretory cell," The statement is too strong without lineage tracing or evidence of the TF driving this process.

Reviewer #2 (Remarks to the Author):

The revised manuscript by Goldfarbmuren et al., has nicely addressed many of the points raised after the initial submission. Specifically, the authors have: 1) added additional donors and increased the number of cells analyzed from their single-cell RNA-seq studies of in vivo samples; 2) more extensively validated the scRNA-seq results; 3) performed additional experiments to follow-up some of the findings from their in vivo and in vitro scRNA-seq profiling studies. This work, particularly the data defining smoke-induced transcriptional changes, will be a valuable resource for the broader respiratory community.

Specific comments:

1) While there are certainly novel aspects of this work – e.g., transcriptional changes induced by smoke, some that span cell states and others that are state-specific, evidence of a lineage hierarchy among tuft cells, PNECs, and ionocytes – some of the other follow-up work presented are validation and/or extensions of previously reported work. As such, the authors should cite the relevant previously studies to contextualize their findings for the reader.

a. The relationship between club cells and mucus-producing goblet cells. While the authors raise a good point about the findings from mice that suggest that club cells can differentiate into goblet cells in response to IL-13, the Okuda paper (Okuda et al., *AJRCCM* 199: 715-727. 2019) quantifies the number of cells that express one or more of SCGB1A1, MUC5AC, and MUC5B in different regions of the airway (i.e., the 'propensity' of co-expression), and in the discussion, cite work from the Crystal lab (Zuo et al., *AJRCCM* 198: 1375 – 1388. 2018) that, together with their data lead them to propose a 'possible lineage relationship between CCSP+ secretory cells and mucin-secreting cells in human airways'. In addition, since the initial submission of this manuscript, the Teichmann lab (Braga et al., *Nature Medicine* 25: 1153-1563. 2019) and the Barbry/Zaragosi labs (Garcia et al., *Development* (2019) 146, dev177428. doi:10.1242/dev.177428) have published complementary data that also identified a similar lineage relationship between club and goblet cells from scRNA-seq data from in

vivo and in vitro (ALI cultures), respectively. These papers should be cited.

b. The secretory/ciliated hybrid population. Braga et al., and Garcia et al., each describe a goblet/ciliated population, similar to the one defined in this work. Zuo et al., also proposed an 'ontologic link between club cells and ciliated cells' based on a small number of SCGB1A1+ DNAI1+ cells in their single cell data. Each of these papers should be cited in this context as well.

c. FOXN4+ population. While the characterization of the FOXN4+ population is certainly more extensive in this manuscript, the fact that it was described in Plasschaert et al., (albeit briefly, and in the supplement) should be acknowledged by referencing the work. The authors say that they cite the paper in the revised manuscript, but I don't see the citation in the body of the results section. Moreover, the FOXN4+ population is described quite extensively as the 'deuterosomal cell' by Garcia et al., which should be cited as well.

d. Rare cell populations. The authors are correct that the majority of the work in Montoro et al., was in mice, but the manuscript does include scRNA-seq data from primary human samples, which they use to identify human pulmonary ionocytes (see Fig. 5 of Montoro et al.). Moreover, Goldfarbmuren et al., state that 'these three rare populations tended to cluster together when the entire epithelium was analyzed (in vitro and in vivo, and by others) potentially indicates a shared origin as well as function' citing Plasschaert et al. However, in both Montoro et al., and Plasschaert et al., all three rare populations clustered separately in the mouse data, and the ionocytes formed a unique cluster in the primary human data (Montoro et al.) and ALI data (Plasschaert et al.) so referencing Plasschaert et al., in this instance is not correct and potentially misleading for the reader. It should be removed from this particular sentence in the manuscript.

e. ALI timecourse. The authors have done an excellent job collecting data from the in vitro ALI model of airway epithelial differentiation at a very high temporal resolution, and the additional follow-up presented in the revised manuscript makes this section feel a bit like a paper within the paper. However, the number of cells collected per timepoint/donor leaves a bit to be desired. Based on the author's response ('each single cell time point (n=20) included cells from 3 donors with the cells from those 60 donor/time points harvested from 2-3 replicate inserts'), 5,976 total cells would mean that ~100 cells were analyzed from each donor/timepoint. By comparison, although there was only a single timepoint analyzed in Plasschaert et al., that study analyzed 10x the number of cells per donor/timepoint. In addition, the recent work from Garcia et al., analyzed 9 timepoints during ALI differentiation, each with at least 10x the number of cells per timepoint. While the number of cells analyzed by Goldfarbmuren et al., was 'sufficient to derive both ciliated and secretory cell lineages', I wonder if they would have identified additional trajectories or substates if they had analyzed more cells/timepoint.

2) For the UMAP projections presented in Fig. 1 and Supplementary Fig. 1 it would be interesting to know whether different groups (smokers, non-smokers) contribute differently to the cell states. The authors should split the data out into different panels by groups in the supplement (perhaps after Supplementary Figure 1b, which is very difficult to look at because of the density of the plot).

3) For the rare states, which the authors define as 0.8% of the total population, it would be informative to report the percentage of each rare cell type for each donor to indicate the donor to donor variability (Figure 1). Note that in 1g there aren't any CHGA+ cells, and without a control for the antibody there's no reason to show the channel (i.e., it's impossible to know whether there aren't any cells in the field or the antibody simply didn't work).

4) The x-axis in Supplementary Figure 11b seems to be mis-labelled. Should the 0.25 and 0.5 labels be reversed?

Reviewer #3 (Remarks to the Author):

The authors have responded appropriately to my prior critiques. This revised version of the manuscript

represents an important contribution to the field.

Reviewer #5 (Remarks to the Author):

This manuscript has a wealth of useful information on cellular phenotypes in the airway and their responses to smoking. I was asked to just review the electrophysiology in the manuscript regarding POU2F3 and FOXI1 KO human airway cultures. It is my understanding reviewers requested this new data and while I can appreciate the reviewer's request and the importance of defining functions of the ionocyte, it really is a minor part of this massive and well-executed study. I also appreciate the author's perspective on how detailed they should dig into the ionocyte functional studies in an already very lengthy and detailed manuscript. Nonetheless, I have some concerns about making the wrong conclusion that would set back the field in an area that is strongly debated. Here are my concerns, but I would like to go on record that even without the electrophysiologic functional data, I would say this is an outstanding manuscript and should be published.

1) The approach to knockout POU2F3 and FOXI1 assumes that biallelic indels at the target occurs at a very high frequency. I would imagine that one would need at least 95% biallelic targeting to classify these are true knockout cultures. But the efficiency of gene targeting at the DNA level in these cultures is not given and this makes it difficult to conclude they are truly knockout for each gene based solely on protein or mRNA FISH expression profile of the gene target. I will simply use FOXI1 KO as an example here. Fig. 6g shows a 2-fold decline in FOXI1 mRNA expression relative to control and no change in CFTR expression. The conclusion being that ionocytes don't contribute to the majority of CFTR expression. But what if the indel frequency in these cultures was about 50%, that would mean less than 25% of the basal cells used to seed these cultures were actually nulls. I realize they also stain for FOXI1 protein in KO cultures (Fig. 6h) and the decline is larger than in the RNAseq, but in this scenario the sensitivity for detecting expression in cells with a single allele indel will also decline and could give a false impression of the KO efficiency. Without knowing the actual indel frequency in the genomic DNA of these populations it is hard to clearly interpret the functional data. How is the indel frequency best demonstrated? The information on the crRNAs used is unclear, but they stated they used two crRNAs to each gene, so TIDE analysis may not work if the indels are large. The target efficiency would need to be done by DNaseq of the locus or some PCR standard cloning of the target and Sanger sequencing. I really don't mean to be nitpicking, but if the indel frequency in the populations is say 50%, I would find the electrophysiology hard to interpret from a CFTR cell biology standpoint. If the target indel efficiency is lower than they Fig. 6h might suggest, I think these limitations need to be pointed out in the text and conclusions toned down. I would also include the crRNA target sites, I may have missed them in the on line supplemental information.

2) My major concern about the electrophysiology data is the manner in which almost all the data is presented as normalized to WT controls. This is also a concern for morphometry and cell counts for expression of the KO targets. How data is normalized can also affect the statistics (i.e., differences with normalization might be significant between groups, where as the raw delta data is not). I would much prefer to see the average raw delta data for all groups (including WT controls) with errors and some example timed traces that are representative the average deltas for each condition. This would provide unit measurements on the y-axis (i.e., uA/cm²) rather than a delta from WT controls. They two keep what is in the figures now, and put the additional data in a supplement. Why do I think this important? Ionocyte abundance, and maybe tuft cells as well, can vary greatly in human cultures. What is the cause of this variability, likely regional differences from where cells were collected in the airways (proximal to distal). Time in culture also impact whether CFTR current persist following differentiation and this could be related to ionocyte precursors in the cultures. This variability in ionocyte abundance also has a relationship to the amount of CFTR-mediated current in ALI cultures

under stimulatory (forskolin) and inhibitor (CFTRinh172) conditions. In my opinion it important to know what the absolute current is under various stimulatory and inhibited conditions. An example of how this could affect the data is as follows. If you have a culture that makes few ionocytes and you knockout FOXI1, you would expect very little electrophysiologic difference from the WT control, but the absolute CFTR-dependent current (forskolin inducible or CFTRinh172 inhibited) in the WT cultures could be very low. This variability could be used as a strength, if the correlations between current changes were linked to both indel frequency and abundance of ionocytes in each donor culture. Right now, my sense is, without seeing the actually values for electrophysiology data rather than reference to WT controls, that there likely is this variability between donors and this could be misinterpreted if the data is only reported as normalized to WT controls. Well-differentiated human ALI cultures typically give CFTR-mediated currents in the range of 20-25 $\mu\text{A}/\text{cm}^2$, but there is a very high range between donors. Poor human ALI currents can be in the range of 5 $\mu\text{A}/\text{cm}^2$, but can still have high resistances. All of this variability is lost when data is reported as normalized to WT controls.

3)CFTR antibodies have fooled the field for decades, having demonstrated very high levels at the apical membrane of ciliated cells, so be careful about interpretation (Fig. 7d). As this group and others have shows, ciliated cells have near the lowest levels of mRNA by scRNAseq despite the apical CFTR staining with highly sited antibodies. mRNA FISH is the gold standard for CFTR detection. While this is not an electrophysiologic concern, the CFTR protein localization is used to make the case around the electrophysiology story.

RESPONSE TO REVIEWERS

Reviewer #1 (Remarks to the Author):

In their revised manuscript, authors have adequately addressed most of my concerns raised with the original submission. They added CRISPR-Cas9 knockout studies to validate the single cell prediction that tuft-like cells are the likely progenitor of both pulmonary neuroendocrine cells and CFTR-rich ionocytes. They have improved the cell classification and presentation by including the global umap and keep cell group names and colors consistent throughout the manuscript. Overall, the manuscript is much improved. Several remaining questions are listed below:

1. In multiple places, authors assessed smoking effects on cell populations via cell proportional changes (Fig2, Fig3, FigS5,). We know that estimating compositional changes in single-cell data requires sufficient cell numbers to robustly assess proportions changes, and sufficient sample numbers to evaluate expected background variation. The estimation of rare cell type proportion is even more difficult (Fig7), usually requires specific statistical models. While these results can be informative of changes in compositions between non-smokers and heavy smokers, however, should be interpreted with caution.

We entirely agree, and accordingly we refer to our cell proportion changes as trends (Figures 2 and now 8), emphasizing our differential expression data as more robust and interpretable. Figure 3 does not contain any cell proportion change data. P-values and donor distributions are provided in Supplementary Figure 5ef.

2. Line 243-244: "In all, these data support a single developmental lineage of human secretory cells, driven by sequentially activated TFs, which transitions through functional intermediates (club cells) to culminate in a multi-functional mucus secretory cell," The statement is too strong without lineage tracing or evidence of the TF driving this process.

Here we are simply providing the reader with a plausible explanation for the in-silico trajectory analysis results, and our wording makes clear "i.e. supportive" that we are not making a statement of fact. However, to ensure the reader is not confused we have restated the sentence below and followed this with a statement of limitation.

"In summary, our in-silico trajectory analysis is suggestive of a developmental lineage of human secretory cells, driven by sequentially activated TFs, which transitions through functional intermediates (club cells) to culminate in a multi-functional mucus secretory cell, although all cells in this lineage need not reach this mucus secretory endpoint, depending on internal or external differentiation cues. However, lineage tracing studies will be needed to confirm this hypothesis."

Reviewer #2 (Remarks to the Author):

The revised manuscript by Goldfarbmuren et al., has nicely addressed many of the points raised after the initial submission. Specifically, the authors have: 1) added additional donors and increased the number of cells analyzed from their single-cell RNA-seq studies of in vivo samples; 2) more extensively validated the scRNA-seq results; 3) performed additional experiments to follow-up some of the findings from their in vivo and in vitro scRNA-seq profiling studies. This work, particularly the data defining

smoke-induced transcriptional changes, will be a valuable resource for the broader respiratory community.

Specific comments:

1) While there are certainly novel aspects of this work – e.g., transcriptional changes induced by smoke, some that span cell states and others that are state-specific, evidence of a lineage hierarchy among tuft cells, PNECs, and ionocytes – some of the other follow-up work presented are validation and/or extensions of previously reported work. As such, the authors should cite the relevant previously studies to contextualize their findings for the reader.

a. The relationship between club cells and mucus-producing goblet cells. While the authors raise a good point about the findings from mice that suggest that club cells can differentiate into goblet cells in response to IL-13, the Okuda paper (Okuda et al., AJRCCM 199: 715-727. 2019) quantifies the number of cells that express one or more of SCGB1A1, MUC5AC, and MUC5B in different regions of the airway (i.e., the ‘propensity’ of co-expression), and in the discussion, cite work from the Crystal lab (Zuo et al., AJRCCM 198: 1375 – 1388. 2018) that, together with their data lead them to propose a ‘possible lineage relationship between CCSP+ secretory cells and mucin-secreting cells in human airways’. In addition, since the initial submission of this manuscript, the Teichmann lab (Braga et al., Nature Medicine 25: 1153-1563. 2019) and the Barbry/Zaragosi labs (Garcia et al., Development (2019) 146, dev177428. doi:10.1242/dev.177428) have published complementary data that also identified a similar lineage relationship between club and goblet cells from scRNA-seq data from in vivo and in vitro (ALI cultures), respectively. These papers should be cited.

Citations have been added.

b. The secretory/ciliated hybrid population. Braga et al., and Garcia et al., each describe a goblet/ciliated population, similar to the one defined in this work. Zuo et al., also proposed an ‘ontologic link between club cells and ciliated cells’ based on a small number of SCGB1A1+ DNAI1+ cells in their single cell data. Each of these papers should be cited in this context as well.

While we have added the Garcia et al. citation to our manuscript, Braga et al. describe a mucous ciliated population which is formed by asthma-induced mucous cell hyperplasia, thus suggesting ciliated -> mucous cell transdifferentiation in the context of type 2 inflammation. This is in contrast to our hybrid population that transitions from a mature mucus secretory cell to an early ciliating cell in the homeostatic, non-type 2 inflamed airway. Since our work is demonstrating the mucus secretory to ciliated cell transition, the Zuo et al. proposal of a club to ciliated transition is also not relevant.

c. FOXN4+ population. While the characterization of the FOXN4+ population is certainly more extensive in this manuscript, the fact that it was described in Plasschaert et al., (albeit briefly, and in the supplement) should be acknowledged by referencing the work. The authors say that they cite the paper in the revised manuscript, but I don’t see the citation in the body of the results section. Moreover, the FOXN4+ population is described quite extensively as the ‘deuterosomal cell’ by Garcia et al., which should be cited as well.

The citations have been added.

d. Rare cell populations. The authors are correct that the majority of the work in Montoro et al., was in mice, but the manuscript does include scRNA-seq data from primary human samples, which they use to identify human pulmonary ionocytes (see Fig. 5 of Montoro et al.).

We agree.

Moreover, Goldfarbmuren et al., state that ‘these three rare populations tended to cluster together when the entire epithelium was analyzed (in vitro and in vivo, and by others) potentially indicates a shared origin as well as function’ citing Plasschaert et al. However, in both Montoro et al., and Plasschaert et al., all three rare populations clustered separately in the mouse data, and the ionocytes formed a unique cluster in the primary human data (Montoro et al.) and ALI data (Plasschaert et al.) so referencing Plasschaert et al., in this instance is not correct and potentially misleading for the reader. It should be removed from this particular sentence in the manuscript.

While ionocytes form a unique cluster, Brush & PNECs cluster together and nearby in Plasschaert et al. Figure 1d (HBEC ALIs). However, to avoid confusion we have modified the text accordingly and removed this reference.

e. ALI timecourse. The authors have done an excellent job collecting data from the in vitro ALI model of airway epithelial differentiation at a very high temporal resolution, and the additional follow-up presented in the revised manuscript makes this section feel a bit like a paper within the paper. However, the number of cells collected per timepoint/donor leaves a bit to be desired. Based on the author’s response (‘each single cell time point (n=20) included cells from 3 donors with the cells from those 60 donor/time points harvested from 2-3 replicate inserts’), 5,976 total cells would mean that ~100 cells were analyzed from each donor/timepoint. By comparison, although there was only a single timepoint analyzed in Plasschaert et al., that study analyzed 10x the number of cells per donor/timepoint. In addition, the recent work from Garcia et al., analyzed 9 timepoints during ALI differentiation, each with at least 10x the number of cells per timepoint. While the number of cells analyzed by Goldfarbmuren et al., was ‘sufficient to derive both ciliated and secretory cell lineages’, I wonder if they would have identified additional trajectories or substates if they had analyzed more cells/timepoint.

There undoubtedly exists more understanding to be found when more cells are analyzed. It is unfortunate that our experiment preceded 10x sequencing technology so acutely, but since we were able to generate useful information from our experiment, we are reporting the data that we have.

2) For the UMAP projections presented in Fig. 1 and Supplementary Fig. 1 it would be interesting to know whether different groups (smokers, non-smokers) contribute differently to the cell states. The authors should split the data out into different panels by groups in the supplement (perhaps after Supplementary Figure 1b, which is very difficult to look at because of the density of the plot).

We have replaced the panel in Supplementary Figure 1b with a new set of panels where each UMAP contains only one smoker status.

3) For the rare states, which the authors define as 0.8% of the total population, it would be informative to report the percentage of each rare cell type for each donor to indicate the donor to donor variability (Figure 1).

A new Supplementary Figure panel (new 1d) has been added with this information.

Note that in 1g there aren't any CHGA+ cells, and without a control for the antibody there's no reason to show the channel (i.e., it's impossible to know whether there aren't any cells in the field or the antibody simply didn't work).

Please note that this is RNAScope, not IF in this panel (as indicated in the legend and by the italic text). But to your point, we have modified the panel to address your comment.

4) The x-axis in Supplementary Figure 11b seems to be mis-labelled. Should the 0.25 and 0.5 labels be reversed?

Thank you for catching this typo, we have corrected it.

Reviewer #3 (Remarks to the Author):

The authors have responded appropriately to my prior critiques. This revised version of the manuscript represents an important contribution to the field.

We thank the reviewer for their previous assessment which we believe after addressing has greatly improved the manuscript. We are glad the revisions have satisfied your concerns.

Reviewer #5 (Remarks to the Author):

This manuscript has a wealth of useful information on cellular phenotypes in the airway and their responses to smoking. I was asked to just review the electrophysiology in the manuscript regarding POU2F3 and FOXI1 KO human airway cultures. It is my understanding reviewers requested this new data and while I can appreciate the reviewer's request and the importance of defining functions of the ionocyte, it really is a minor part of this massive and well-executed study. I also appreciate the author's perspective on how detailed they should dig into the ionocyte functional studies in an already very lengthy and detailed manuscript. Nonetheless, I have some concerns about making the wrong conclusion that would set back the field in an area that is strongly debated. Here are my concerns, but I would like to go on record that even without the electrophysiologic functional data, I would say this is an outstanding manuscript and should be published.

Thank you for your positive assessment, please see below for our detailed responses.

1) The approach to knockout POU2F3 and FOXI1 assumes that biallelic indels at the target occurs at a very high frequency. I would imagine that one would need at least 95% biallelic targeting to classify these are true knockout cultures. But the efficiency of gene targeting at the DNA level in these cultures is not given and this makes it difficult to conclude they are truly knockout for each gene based solely on protein or mRNA FISH expression profile of the gene target. I will simply use FOXI1 KO as an example here. Fig. 6g shows a 2-fold decline in FOXI1 mRNA expression relative to control and no change in CFTR expression. The conclusion being that ionocytes don't contribute to the majority of CFTR expression. But what if the indel frequency in these cultures was about 50%, that would mean less than 25% of the basal cells used to seed these cultures were actually nulls. I realize they also stain for FOXI1 protein in KO cultures (Fig. 6h) and the decline is larger than in the RNAseq, but in this scenario the sensitivity for

detecting expression in cells with a single allele indel will also decline and could give a false impression of the KO efficiency. Without knowing the actual indel frequency in the genomic DNA of these populations it is hard to clearly interpret the functional data. How is the indel frequency best demonstrated? The information on the crRNAs used is unclear, but they stated they used two crRNAs to each gene, so TIDE analysis may not work if the indels are large. The target efficiency would need to be done by DNAseq of the locus or some PCR standard cloning of the target and Sanger sequencing. I really don't mean to be nitpicking, but if the indel frequency in the populations is say 50%, I would find the electrophysiology hard to interpret from a CFTR cell biology standpoint. If the target indel efficiency is lower than they Fig. 6h might suggest, I think these limitations need to be pointed out in the text and conclusions toned down. I would also include the crRNA target sites, I may have missed them in the online supplemental information.

We developed a dual-exonic guide (2 guides) CRISPR-RNP method for generating KOs in human airway epithelial cells (subject of a manuscript in preparation). In this system, editing can occur either at only a single site or at both sites, with the latter possibly leading to the deletion of the sequence between the two sites. Therefore, quantitation of editing efficiency at the sequence level is difficult. Rather, as is normal protocol for us and other groups, before performing experiments we test the guides by high resolution melt curve analysis of PCR products covering the two cut sites¹⁵. See HRM curves below for both guides in the edited vs. control edited culture PCR products. This analysis yielded large (Ex1 guide) and moderate (Ex2 guide) differences between the melting curves of the cut site PCR products from the CRISPR-RNP with non-targeting guide (control) vs. FOXI1 guides, characteristic of editing occurring at both sites. Moreover, we attached an HRM curve for a pcr product that spans both cutting sites, which due to pcr conditions will only amplify if the region in between is deleted. As shown in these curves the deletion PCR product was detected in the FOXI1 targeted cells but not the controls.

Beyond this, editing efficiency can, but does not always, correlate with mRNA or protein loss, as many edits are silent with respect to mRNA and protein expression. Some edits may be null because they cause mRNA instability and degradation as appears to be the case with FOXI1 KO cells which exhibit a 4-fold loss of mRNA, suggesting at least half of FOXI1 alleles are edited. However, some edits can create stops on translation (null alleles) without loss of mRNA, so the mRNA change gives an incomplete view of KO efficiency, or a readout specific for this type of null allele. The ultimate goal of the experiment is to stop protein production. As you suggest, assessing this is difficult given that FOXI1 cells make up a small percentage of cultures, despite this, our analysis suggests 400% reduction in FOXI1 positive cells.

Regarding the use of the term knockout, it is standard in the field for this type of experiment and separates it from a knockdown experiment (siRNA) where alleles are not lost in any cells but rather a percent decrease in mRNA and protein expression is achieved across all cells. Here, we have a mosaic of cells mixed between homozygous, heterozygous, and wildtype KO. However, we will add clarification to the manuscript to ensure the reader knows KO does not mean a complete KO in all cells.

Regarding the amount of KO needed to provide useful information, by all measures assessed (DNA, mRNA, protein) we are achieving a significant amount of editing and KO with our strategy. We believe enough to draw valid conclusions, especially when considering we are performing the editing in 5 donors' cells, with readouts from multiple assays, and finding the data across these donors and assays is consistent and coherent with the conclusions we put forth.

As an example, you voiced concern over our conclusion that loss of ionocytes does not result in either a dramatic loss in CFTR expression or CFTR activity. To outline the justification for this in the paper, CRISPR-Cas9-related and independent data we find:

1. (Non-CRISPR-Cas9-related) Our in vivo tracheal airway epithelial single cell data generated on 15 donors and over 36,000 cells reveal that although ionocytes express per cell, much more CFTR than other epithelial cell types, their low frequency means that they only contribute 11% of total epithelial CFTR. Whereas 89% of epithelial CFTR expression is contributed in aggregate by the other 10 non-ionocyte epithelial cell types composing the airway epithelium (Figure 8b).
2. (Non-CRISPR-Cas9-related) Secondly, during in vitro airway epithelial mucociliary differentiation we find CFTR expression reaches an initial peak at ~day 5, when the expression of ionocyte gene markers is still very low and long before the peak of ionocyte marker gene expression (Figure 8c).
3. (CRISPR-Cas9 data) We can detect no change in CFTR expression with FOXI1 loss. We note that if CFTR was only expressed by ionocytes we should have the power with our sample size and the amount of targeting achieved to see an effect on CFTR expression, but rather if these cells only contribute 11% of total epithelial CFTR then that change would be difficult to detect. Moreover, we find no loss in CFTR activity with FOXI1 KO cells. Of course, this raises the possibility as you suggest that the level of FOXI1 KO is insufficient to see effects. However, this is directly contradicted by the fact that we observe a very significant reduction in the levels of ASCL3, a gene that has been shown by us and others to be restricted to ionocytes and by Rajagopal to be downstream of FOXI1 in mice ionocytes. Additionally, we do see a significant change in epithelial electrophysiology with FOXI1 KO, as might be expected when the epithelium loses significant numbers of a cell type important to epithelial ion transport.

We also submit that experiments in humans and with human cells are necessary to understand and confirm human biology. Although these experiments usually cannot be as controlled as with model organisms, we believe we have leveraged the latest tools, applied carefully, to derive novel and important human data. With all this being said we agree that our results are best viewed as strong support for the conclusions we reach rather than definitive proof. If you rereview our actual words we believe you will find that we consistently avoid any dogmatic factual statements, but rather say our data is “suggestive” or “supports”. Indeed, instead we urge caution against the rapidly proliferating view that ionocytes are the only source of CFTR and only cell type important for CF, in that this story may be incomplete (Lines 765-774).

2) My major concern about the electrophysiology data is the manner in which almost all the data is presented as normalized to WT controls. This is also a concern for morphometry and cell counts for expression of the KO targets. How data is normalized can also affect the statistics (i.e., differences with normalization might be significant between groups, where as the raw delta data is not).

All statistics/modeling was performed on the non-normalized raw delta data. The normalization to WT control cultures in the figures enables non-experts to digest the directions of changes observed from this complex data, and facilitates comparison between the expression and functional data.

I would much prefer to see the average raw delta data for all groups (including WT controls) with errors and some example timed traces that are representative the average deltas for each condition. This would provide unit measurements on the y-axis (i.e., $\mu\text{A}/\text{cm}^2$) rather than a delta from WT controls. They two keep what is in the figures now, and put the additional data in a supplement.

This data is in Supplementary Table 4, and example traces are in Supplementary Figure 10a. We have now added a new Supplementary Figure 11 with graphs of all the non-normalized data (error bars to represent the standard error of electrophysiology readings of 4 replicate cultures per donor per KO treatment). Further we’ve added a clarified reference in the text to these supplementary figures.

Why do I think this important? Ionocyte abundance, and maybe tuft cells as well, can vary greatly in human cultures. What is the cause of this variability, likely regional differences from where cells were collected in the airways (proximal to distal).

We have also observed differences in ionocyte abundance across un-edited human cultures between an average of 8.9 – 29.4 FOXI1+ cells per field (Supplementary Table 4, tab 42_FigS7hi).

Time in culture also impact whether CFTR current persist following differentiation and this could be related to ionocyte precursors in the cultures.

We also observe that ionocyte marker gene abundances increase over time in culture (See Figures now labeled 7f, 8c), however our electrophysiology was all performed at day 32 of ALI culture to maximize the presence of these rare cells.

This variability in ionocyte abundance also has a relationship to the amount of CFTR-mediated current in ALI cultures under stimulatory (forskolin) and inhibitor (CFTRinh172) conditions. In my opinion it important to know what the absolute current is under various stimulatory and inhibited conditions. An example of how this could affect the data is as follows. If you have a culture that makes few ionocytes and you knockout FOXI1, you would expect very little electrophysiologic difference from the WT control,

but the absolute CFTR-dependent current (forskolin inducible or CFTRinh172 inhibited) in the WT cultures could be very low.

In our hands, we do not observe a positive correlation between ionocyte abundance (as measured by IF microscopy or qPCR) and CFTR-mediated current response (i.e. change in current with forskolin/IBMX or CFTRinh172 treatment). In fact, we observe a weak negative correlation indicating that the smaller the ionocyte population in a given culture (due to donor variation or either KO condition) the larger magnitude current response. Further, in the donor with the fewest ionocytes in the wildtype cultures (T133), we still observe the dramatic electrophysiologic difference with FOXI1 KO observed with the more ionocyte-rich donors. This indicates that none of our cultures make sufficiently few ionocytes to fall into the hypothetical category you describe.

This variability could be used as a strength, if the correlations between current changes were linked to both indel frequency and abundance of ionocytes in each donor culture. Right now, my sense is, without seeing the actual values for electrophysiology data rather than reference to WT controls, that there likely is this variability between donors and this could be misinterpreted if the data is only reported as normalized to WT controls.

New Supplementary Figure 11 depicts the data in Supplementary Table 4 (now tabs 53, 54, 55), demonstrating the robustness of both wildtype and KO cultures, across 4 donors, both at baseline and with each treatment. As you can see, we do not observe a relationship between ionocyte abundance and current at any step in the series.

Well-differentiated human ALI cultures typically give CFTR-mediated currents in the range of 20-25 $\mu\text{A}/\text{cm}^2$, but there is a very high range between donors. Poor human ALI currents can be in the range of 5 $\mu\text{A}/\text{cm}^2$, but can still have high resistances. All of this variability is lost when data is reported as normalized to WT controls.

Please see new Supplementary Figure 11 which demonstrates how all our donors/KO treatments exhibit the current responses expected for high-quality ALI cultures.

3) CFTR antibodies have fooled the field for decades, having demonstrated very high levels at the apical membrane of ciliated cells, so be careful about interpretation (Fig. 7d). As this group and others have shown, ciliated cells have near the lowest levels of mRNA by scRNAseq despite the apical CFTR staining with highly sited antibodies. mRNA FISH is the gold standard for CFTR detection. While this is not an electrophysiologic concern, the CFTR protein localization is used to make the case around the electrophysiology story.

Thank you for your thoughtful comment. We share your skepticism about CFTR antibodies, and have further validated this antibody using cultures from a CF patient with the F508del mutation (see figure from a recent grant proposal below).

[Redacted]

Since this mutation is not a protein null, but rather leads to misfolding and degradation, we observe the lower, but not absent signal in the bottom middle panel. Corrector treatments known to help the protein fold and get to the cell surface increase the signal in the treated panel (right). Even with this extra confidence from an unrelated study, we have only included the IF labeling as an additional datapoint that is consistent with our multiple lines of CFTR mRNA evidence (in vivo sc-RNAseq in Figure 8b, bulk in vitro RNAseq of the dense time-course in Figure 8c and qPCR of the KO human cultures in Figure 7g). Further, although not explicitly mentioned in the paper, we do see ionocyte hotspots alongside additional low density signal by CFTR mRNA FISH as well (see Figure 7d). Thus, we have multiple mRNA-based lines of evidence to support the electrophysiology data depicting robust CFTR-mediated ion transport in the KOs, thus we do not rely solely or primarily on the protein data.